# LILRB1-HLA-G axis defines a checkpoint driving natural killer cell exhaustion in tuberculosis

Jing Wang[1,5], Qiyao Chai[1,5], Zehui Lei [1,2,5], Yiru Wang[1,2,5], Jiehua He[1,2,5], Pupu Ge[1,2], Zhe Lu [1,2], Lihua Qiang[1,2], Dongdong Zhao[1,2], Shanshan Yu[3], Changgen Qiu[1,2], Yanzhao Zhong[1,2], Bing-Xi Li[1], Lingqiang Zhang[4], Yu Pang[3 ✉], George Fu Gao [1,2 ✉] & Cui Hua Liu [1,2 ✉]

## Abstract

**Chronic infections, including *Mycobacterium tuberculosis* (Mtb)-caused tuberculosis (TB), can induce host immune exhaustion. However, the key checkpoint molecules involved in this process and the underlying regulatory mechanisms remain largely undefined, which impede the application of checkpoint-based immunotherapy in infectious diseases. Here, through adopting time-of-flight mass cytometry and transcriptional profiling to systematically analyze natural killer (NK) cell surface receptors, we identify leukocyte immunoglobulin like receptor B1 (LILRB1) as a critical checkpoint receptor that defines a TB-associated cell subset (LILRB1+ NK cells) and drives NK cell exhaustion in TB. Mechanistically, Mtb-infected macrophages display high expression of human leukocyte antigen-G (HLA-G), which upregulates and activates LILRB1 on NK cells to impair their functions by inhibiting mitogen-activated protein kinase (MAPK) signaling via tyrosine phosphatases SHP1/2. Furthermore, LILRB1 blockade restores NK cell-dependent anti-Mtb immunity in immuno-humanized mice. Thus, LILRB1-HLA-G axis constitutes a NK cell immune checkpoint in TB and serves as a promising immunotherapy target.**

**Keywords** *Mycobacterium tuberculosis* (Mtb); Checkpoint Molecule; Immunotherapy; Natural Killer Cell; Leukocyte Immunoglobulin Like Receptor B1 (LILRB1)
**Subject Categories** Immunology; Microbiology, Virology & Host Pathogen Interaction

## Introduction

In chronic infections or tumors, immune cells exposed to continuous antigenic or inflammatory stimuli acquire a dysfunctional status known as exhaustion, which leads to compromised host immune protective responses and impaired efficacy of vaccination against tumors or pathogens (Abers et al, 2019; Wykes and Lewin, 2018). Immune checkpoint signaling controls the duration and magnitude of immune cell responses, and its dysregulation leads to immune cell exhaustion. While immunotherapy based on checkpoint blockade has revolutionized cancer therapeutics and provided potential therapeutic approaches against chronic viral infections (Kubli et al, 2021), whether it can be effective for treating chronic bacterial infections remain largely unexplored. Persistent infection caused by *Mycobacterium tuberculosis* (Mtb) has been linked to systemic exhaustion of host immune system, which exacerbates the progression of tuberculosis (TB), a chronic infectious disease that remains a major public-health threat worldwide (World Health Organization, 2023). Moreover, the continuing spread of drug-resistant Mtb and limited protective efficacy of Bacillus Calmette–Guérin (BCG), the only vaccine currently available against TB, further pose difficult challenges to TB control. Against this backdrop, host-based immunotherapy, such as targeting checkpoint signaling to provide alternative anti-TB strategies, has attracted growing interest. However, the mechanisms underlying immune checkpoint-mediated functional exhaustion during TB infection remain largely unclear.

Previous attempts to block T cell-based checkpoint molecules, such as programmed death protein-1 (PD-1), for anti-Mtb treatment have led to undesirable effects on the host including hyper-inflammation and exacerbation of TB progression (Barber et al, 2011; Barber et al, 2019; Kauffman et al, 2021; Langan et al, 2020; Tezera et al, 2020). Thus, there might be alternative critical checkpoint molecules involved in regulating host anti-Mtb immunity that remain to be identified for developing anti-Mtb immunotherapy with good efficacy and safety. Innate immunity is the first line of host defense against invading pathogens including Mtb, and it has been indicated to not only assist adaptive immunity, but also control the bacterial growth at the initial phase of infection (Khan et al, 2016). Natural killer (NK) cells constitute an important part of the innate immune system, which exert prompt immune response towards pathogen-infected cells, thus they have emerged as attractive targets for developing immune checkpoint-based therapy against infectious diseases including TB.

[1]CAS Key Laboratory of Pathogenic Microbiology and Immunology, Institute of Microbiology, Chinese Academy of Sciences, Beijing, China. [2]Savaid Medical School, University of Chinese Academy of Sciences, Beijing, China. [3]Beijing Tuberculosis and Thoracic Tumor Research Institute, Beijing Chest Hospital, Capital Medical University, Beijing, China. [4]State Key Laboratory of Proteomics, National Center for Protein Sciences, Beijing Institute of Lifeomics, Beijing, China. [5]These authors contributed equally: Jing Wang, Qiyao Chai, Zehui Lei, Yiru Wang, Jiehua He. ✉E-mail: py@bjxkyy.cn; gaof@im.ac.cn; liucuihua@im.ac.cn

Unlike T lymphocytes, NK cells mediate cytotoxic functions independent of major histocompatibility complex (MHC)-mediated antigen presentation, and they either directly kill pathogen-infected cells, or secrete various cytokines, such as interferon-γ (IFN-γ) and tumor necrosis factor-α (TNF-α), to eliminate target cells. NK cells possess a variety of activating or inhibitory receptors, whose interactions with their ligands determine the killing activities of NK cells (Sivori et al, 2019). Active TB (ATB) patients display a decreased frequency of NK cells as compared to asymptomatic individuals with latent TB infection (LTBI) (Roy Chowdhury et al, 2018). Furthermore, NK cells from ATB patients are featured with decreased expression of activating receptors (e.g., NKp30 and NKp46) (Bozzano et al, 2009), indicating the functional change of NK cells during Mtb infection. However, the phenotypic and functional properties of NK cells in TB patients remain largely unclear. In this study, we combine transcriptional profiling with time-of-flight mass cytometry (CyTOF) to systematically analyze the functional characteristics and immune receptor repertoire of NK cells in individuals at different status of Mtb infection, with an aim to identify the critical NK checkpoint molecules and their signaling network that may serve as new targets for improving host anti-Mtb immunity. Our findings reveal that leukocyte immunoglobulin like receptor B1 (LILRB1) and its ligand human leukocyte antigen-G (HLA-G) constitute a critical immune checkpoint axis to drive NK cell exhaustion in TB patients, thereby providing a promising target for TB immunotherapy.

## Results

### NK cell functions are compromised in TB

To characterize the immune status of Mtb-infected individuals at different status of infection, we began with analysis of different immune cell populations in the peripheral blood from a Han Chinese cohort who were assigned into three groups including healthy control (HC), LTBI, or ATB (Fig. EV1A and Dataset EV1). As compared to HC or LTBI groups, ATB group displayed decreased percentage of NK cells and T cells but increased percentage of monocytes in peripheral blood (Figs. 1A and EV1B). In addition, the percentage of B cell population in ATB group was lower than that in LTBI group (Figs. 1A and EV1B). We then determined the absolute counts of these peripheral blood cell populations to confirm their variations among tested groups, and consistently found that ATB group showed decreased count of NK cells and T cells but increased count of monocytes in peripheral blood (Fig. EV1C). Considering that ATB patients showed particularly low peripheral blood levels of NK cells, whose immune dysfunction has been correlated to TB progression (Bozzano et al, 2009; Roy Chowdhury et al, 2018), we then characterized transcriptional profiling to investigate functional changes of NK cells during the course of TB. Gene set enrichment analysis (GSEA) revealed that genes involved in multiple inflammatory pathways [such as Toll-like receptor (TLR), NOD-like receptor (NLR), and TB-related signaling pathways] and apoptosis were upregulated, while genes related to phosphoinositide-3 kinase (PI3K)-AKT and RAP1 signaling pathways were downregulated, in NK cells from ATB group as compared to that from HC or LTBI group (Fig. 1B).

Accordingly, as compared to HC or LTBI group, a large amount of inflammatory response- and apoptosis-associated genes were upregulated, while many NK cell cytotoxic activity-associated genes were downregulated in ATB group (Fig. 1C; Appendix Fig. S1). These data indicate an altered immune status of NK cells featured with increased activation of inflammatory and apoptotic pathways and decreased activation of cytotoxic activity-related pathways in ATB patients.

To determine the anti-Mtb immune activity of NK cells at different status of TB, we established a co-culture system containing collected peripheral blood NK cells and Mtb-infected monocyte-derived macrophages (MDMs) (Yoneda and Ellner, 1998). ATB group exhibited higher frequency of NK cells expressing inflammatory cytokines, including IFN-γ and TNF-α, than HC and LTBI groups when NK cells were cultured alone, while showing comparable frequency of NK cells expressing IFN-γ and TNF-α to the other two groups in MDM-NK cell co-culture system (Fig. 1D,E; Appendix Fig. S2A,B). In contrast, ATB group exhibited lower frequency of NK cells expressing CD107a, a marker of NK cell degranulation that indicates cytotoxic activity (Krzewski et al, 2013), than the other two groups in MDM-NK cell co-culture system (Fig. 1F; Appendix Fig. S2C). Moreover, a higher percentage of apoptotic NK cells was detected in ATB group as compared to HC and LTBI groups when co-cultured either with or without Mtb-infected MDMs (Fig. 1G; Appendix Fig. S2D). Finally, NK cells from ATB group were less able to reduce macrophage viability and Mtb survival in MDM-NK cell co-culture system compared to that from HC or LTBI group (Fig. 1H; Appendix Fig. S2E,F), indicating their attenuated ability to kill Mtb-infected MDMs and to control Mtb infection. Together, these results suggest that NK cells in ATB patients display an exhausted phenotype characterized by altered transcriptional profiling, impaired cytotoxic activity, increased apoptosis, as well as compromised anti-Mtb capacity.

### LILRB1 defines a TB-associated NK cell subset

The effector functions of NK cells in response to infection are governed by the integration of activating and inhibitory signals from cell surface receptors (Sivori et al, 2019). To investigate the key immune factors that determine the exhausted phenotype of NK cells in ATB patients, we performed CyTOF to quantify the expression of 39 immune markers on single NK cells from HC, LTBI, and ATB groups (Appendix Fig. S3 and Appendix Table S1; Dataset EV2). t-distributed stochastic neighbor embedding (t-SNE)-based clustering of CyTOF single-cell data revealed 19 separate clusters of NK cells, among which the cluster 5 (C5) subset was found to be primarily present in ATB group (Fig. 2A–C). Notably, C5 NK cell subset did not express the majority of the activating receptors such as NKG2C, NKG2D, NKp30, NKp44, and NKp46 (Fig. 2C). In contrast, C5 subset showed the highest expression of LILRB1, an inhibitory receptor whose activation dampens NK cell immune responses (Favier et al, 2010; Harrison et al, 2020), as compared to the other subsets (Fig. 2C,D). Accordingly, ATB group showed a distinctly higher frequency of NK cells expressing LILRB1 in the C5 subset as compared to that in HC and LTBI groups (Fig. 2E). Moreover, t-SNE visualization further revealed a distinctive immune signature of this LILRB1-expressing C5 NK cell subset, which was characterized by rare expression of granzyme B (GZMB) and perforin (PRF1) but high expression of IFN-γ and

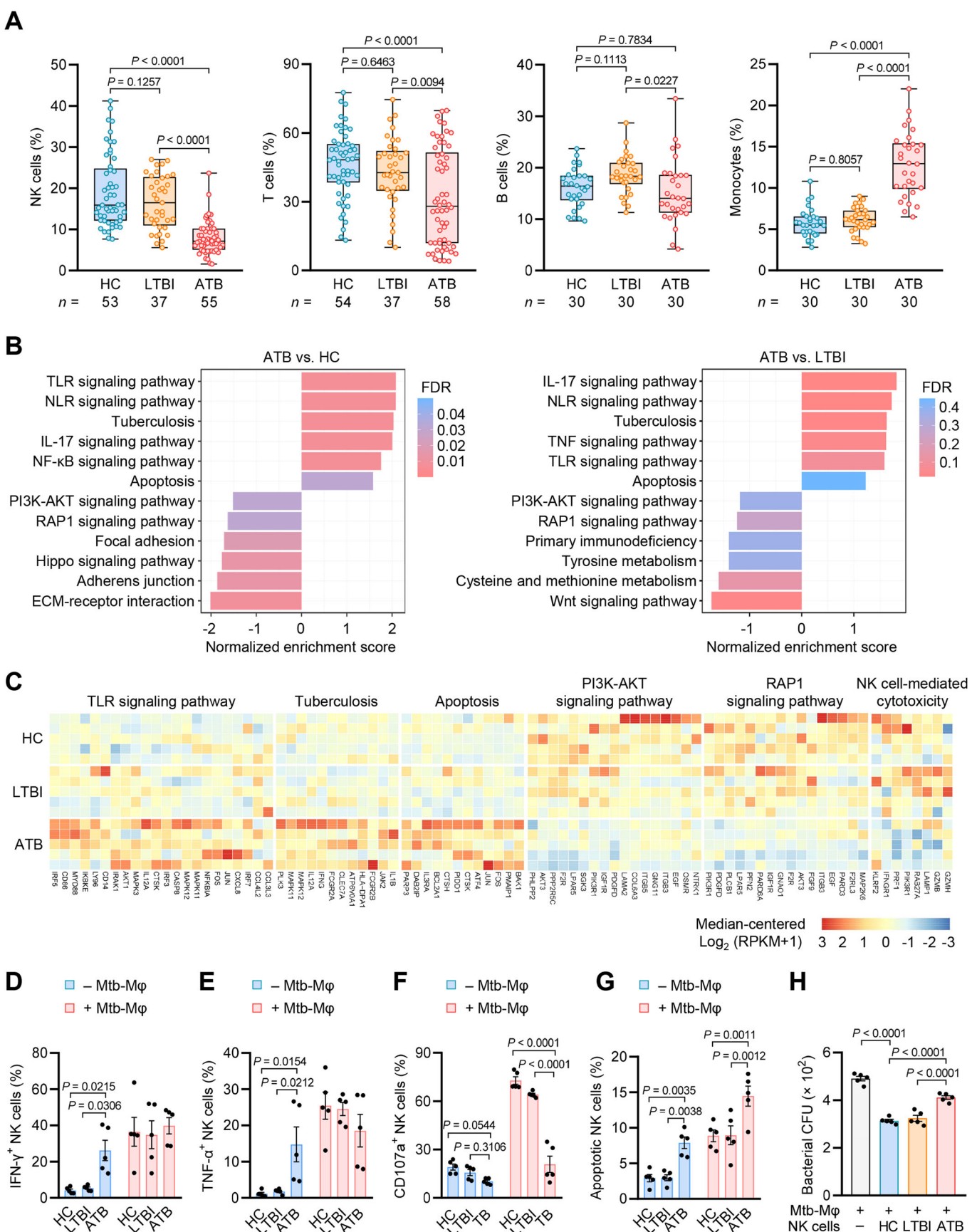

◀ **Figure 1.   NK cells from ATB patients display compromised anti-Mtb immunity.**

(A) Frequencies of immune cell subsets in peripheral blood of individuals (*n*) in HC, LTBI, and ATB groups as determined by FACS. Box-whisker plot indicates the interquartile range (box), the median value (line within the box), and the maximum and minimum value (whiskers). (B) Gene set enrichment analysis (GSEA) of peripheral blood NK cell transcriptional profiles in ATB group as compared to that in HC (left) or LTBI (right) group. (C) Heatmap depicting the differential expression of genes involved in the indicated signaling pathways based on Kyoto encyclopedia of genes and genomes (KEGG) enrichment among HC, LTBI, and ATB groups. (D–F) Percentages of NK cells expressing IFN-γ (D), TNF-α (E), or CD107a (F) as determined by FACS (see also Appendix Fig. S2A–C). (G) Percentages of apoptotic NK cells as determined by FACS (see also Appendix Fig. S2D). (H) Mtb survival in MDM-NK cell co-cultures as determined by colony forming unit (CFU) counting. For (D–H), NK cells from individuals in HC, LTBI, or ATB groups were co-cultured with (+) or without (−) Mtb-infected MDMs (Mtb-Mφ) for 24 h. See "Methods" for details. Data are mean ± SEM [*n* = 5 donors per group in (D–H)]. Statistical significance was determined using one-way ANOVA (A, H) or two-way ANOVA (D–G) with Tukey's post-hoc test. Results are representative of three independent experiments. Source data are available online for this figure.

TNF-α (Fig. 2F), consistent with transcriptome-derived immune signatures of NK cells in ATB patients (Fig. 1C). These data suggest that LILRB1 serves as an indicator for a unique NK cell subset (i.e., C5 subset) in ATB patients.

## LILRB1 drives NK cell exhaustion in TB

Next, we examined the correlation between LILRB1 expression and NK cell effector functions upon Mtb infection. The CyTOF single-cell data-based analysis implied that the expression level of LILRB1 was negatively correlated with that of GZMB and PRF1, but positively correlated with that of IFN-γ and TNF-α, in NK cells (Fig. 3A). We then overexpressed LILRB1 in NK cells to determine its effect on NK cell response to Mtb infection using a retroviral transduction system (Chai et al, 2022) (Appendix Fig. S4). NK cells with overexpression of LILRB1 showed decreased percentage in expressing CD107a but little change of that in expressing IFN-γ or TNF-α, reduced production of GZMB and PRF1, increased apoptosis, and attenuated anti-Mtb capacity when co-cultured with Mtb-infected MDMs (Fig. 3B–H). We then further examined the expression of LILRB1 in the lungs of ATB patients. By using CD69 and CD103 as markers of NK cell tissue-residency (Hervier et al, 2019), we found that both circulating (CD56⁺CD69⁻ or CD56⁺CD103⁻) and lung-resident (CD56⁺CD69⁺ or CD56⁺CD103⁺) NK cells showed upregulated LILRB1 expression within granulomatous lesions, and we also noticed that these NK cells displayed relatively higher LILRB1 expression in necrotic granulomas than that in non-necrotic granulomas (Fig. 3I–K). Together, these data suggest that during Mtb infection, LILRB1 is a potential key checkpoint receptor whose upregulation is responsible for the exhaustion phenotype of NK cells including the decreased expression of CD107a, reduced production of GZMB and PRF1, increased apoptosis, and attenuated anti-Mtb capacity.

## HLA-G induces LILRB1-dependent NK cell exhaustion in TB

Next, we sought to determine whether the expression of LILRB1 is exclusively changed on NK cells upon TB infection. ATB group showed significantly increased frequency of NK cells expressing LILRB1 as compared to that in HC and LTBI groups (Fig. 4A; Appendix Fig. S5A,B), confirming the results obtained from CyTOF. Contrary to LILRB1⁺ NK cells, the frequency of NK cells expressing T-cell immunoreceptor with Ig and ITIM domains (TIGIT), a critical inhibitory receptor correlated with NK cell exhaustion in host anti-tumor immunity (Zhang et al, 2018), was decreased in ATB patients (Fig. 4B; Appendix Fig. S5A,C).

Moreover, LILRB1-positive (LILRB1⁺) T cells were decreased in ATB patients, concordant with previous findings suggestive of host weak inhibitory immune signals within T cells in response to Mtb infection (Appendix Fig. S5D,E) (Gideon et al, 2022; McCaffrey et al, 2022; Wong et al, 2018). In addition, there were no significant difference in frequencies of LILRB1⁺ B cells and LILRB1⁺ monocytes among three groups (Appendix Fig. S5D,F,G). Together, these results support a unique immunoregulatory role of LILRB1 in NK cell response to Mtb infection.

We then further explored how the expression and activation of LILRB1 on NK cells are regulated during Mtb infection. The nonclassical MHC class I molecule HLA-G is the most widely expressed ligand of LILRB1 that has been linked to TB pathogenesis (Saurabh et al, 2016). Accordingly, in the serum of ATB patients, an increased level of soluble HLA-G was detected (Fig. 4C). Treatment with HLA-G increased expression of LILRB1 on NK cells from HC subjects in a dose-dependent manner (Fig. 4D), consistent with the previous observation (LeMaoult et al, 2005). Furthermore, MDMs infected with Mtb showed increased amount of transmembraneous HLA-G on the cell surface and soluble HLA-G in the supernatant (Fig. EV2A–C), both of which drive the immune inhibitory signaling (Park et al, 2004; Saurabh et al, 2016). When co-cultured with Mtb-infected MDMs, NK cells from HC subjects showed increased expression of LILRB1, and this effect was markedly reduced upon treatment with an anti-HLA-G neutralizing monoclonal antibody (87G mAb), which recognizes both transmembraneous and soluble HLA-G (Feger et al, 2007) (Fig. 4E). Thus, Mtb infection promotes the expression of HLA-G in macrophages to induce upregulation of LILRB1 on NK cells. By comparison, the frequencies of NK cells expressing KIR2DL4, another receptor for HLA-G (Rajagopalan and Long, 1999), were similarly at a low level among HC, LTBI, and TB groups (Fig. EV2D–F), and Mtb-infected MDMs showed little effect on inducing the upregulation of KIR2DL4 on NK cells (Fig. EV2G). These results further support a specific role of HLA-G-LILRB1 signaling axis in NK cell regulation during Mtb infection. To further investigate which component from Mtb induces upregulation of HLA-G expression in macrophages, we treated Mtb with proteinase K or sodium periodate (NaIO₄) to degrade bacterial cell wall proteins and carbohydrate residues, respectively, and found that lack of carbohydrate residues abolished the ability of Mtb to promote HLA-G transcription in MDMs (Fig. EV2H). Then, we removed main carbohydrate components in Mtb cell wall by using mutanolysin (which digests peptidoglycan) or α-mannosidase (which transform lipomannan and lipoarabinomannan into mannoglycolipids), and found that peptidoglycan is the critical Mtb component that increases HLA-G production in MDMs during infection (Fig. EV2I,J).

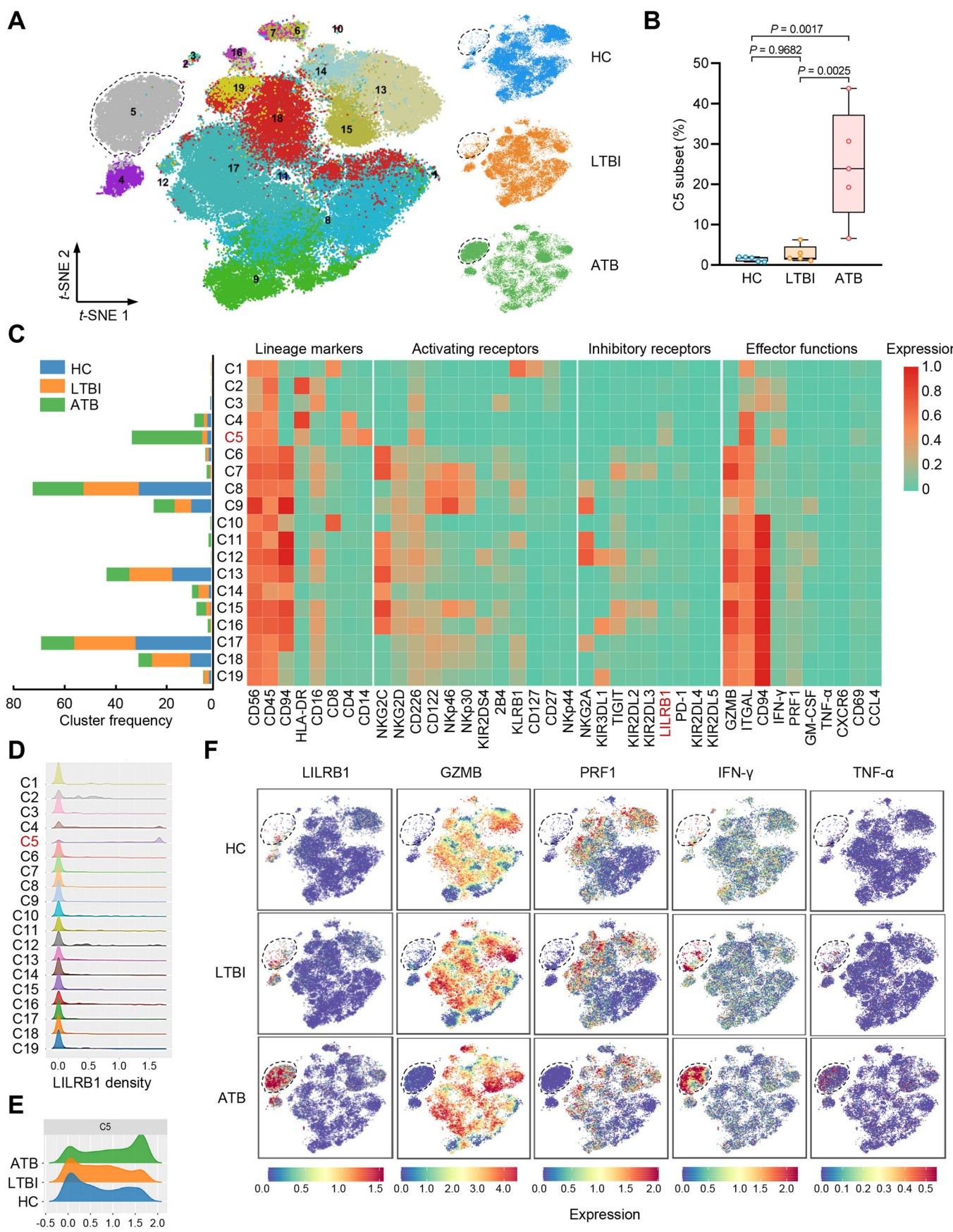

◄ **Figure 2. LILRB1 defines an TB-associated NK cell subset.**

(A) *t*-SNE plot of CyTOF single-cell data revealing 19 separate clusters (C1–C19) of NK cells from individuals in HC, LTBI, and ATB groups. The C5 subset is indicated by dotted circles. (B) Percentages of C5 subset in total NK cells among HC, LTBI, and ATB groups. Box-whisker plot indicates the interquartile range (box), the median value (line within the box), and the maximum and minimum value (whiskers). (C) Frequencies and repertoires of each NK cell subset among HC, LTBI, and ATB groups. The bar graph (left) shows the frequencies of C1–C19 subsets in HC (blue), LTBI (orange), or ATB (green) groups. The heatmap (right) shows expression of each indicated markers in C1–C19 subsets. The C5 subset and inhibitory receptor LILRB1 are highlighted in red. (D) LILRB1 expression density in each NK cell subset. The C5 subset is indicated in red. (E) LILRB1 expression density in C5 subset of NK cells among HC, LTBI, and ATB groups. (F) *t*-SNE plots for expression of indicated markers among HC, LTBI, and ATB groups. The C5 subset is indicated by dotted circles. Data are mean ± SEM (*n* = 5 donors per group) in (B). Statistical significance was determined using one-way ANOVA with Tukey's post-hoc test.

Next, we sought to determine whether HLA-G induces the LILRB1-mediated immune inhibition of NK cells. Treatment with anti-HLA-G neutralizing antibody markedly increased the expression of CD107a and the production of GZMB and PRF1, but not IFN-γ or TNF-α, concurrent with reduced apoptosis in LILRB1-overexpressed NK cells co-cultured with Mtb-infected MDMs (Figs. 4F–I and EV2K,L). Accordingly, neutralizing HLA-G led to enhanced ability of LILRB1-overexpressed NK cells in controlling Mtb survival within MDMs (Fig. 4J). However, these effects from HLA-G blockade were slightly observed in control group of NK cells without overexpression of LILRB1 (Fig. 4F–J). Furthermore, when the LILRB1-HLA-G interaction was disrupted by an anti-LILRB1 blocking monoclonal antibody, GHI/75 mAb (Dorling et al, 2000), the expression of CD107a and the production of GZMB and PRF1, but not IFN-γ or TNF-α, were increased in NK cells from ATB patients, concurrent with reduced apoptosis and enhanced anti-Mtb capacity (Figs. 4K–P and EV2M,N). By comparison, blocking LILRB1 showed little effect on NK cells from HC or LTBI group (Fig. 4K–P), probably due to their rare expression of LILRB1. Together, these data support an inhibitory role of LILRB1 in NK cell anti-Mtb immunity by acting as a checkpoint receptor that can be activated by HLA-G.

## LILRB1-HLA-G interaction drives NK cell exhaustion by inhibiting MAPK signaling via SHP1/2

Next, we investigated the mechanisms underlying LILRB1-HLA-G interaction-mediated inhibition of NK cell functions. By integrative analysis of transcriptional data, several enriched transcriptional factors were predicted to be potentially activated [normalized enrichment score (NES) > 0] or suppressed (NES < 0) in NK cells from ATB patients, among which protein C-ets-1 (ETS1), a critical regulator involved in NK cell apoptosis and cytolytic activity (Taveirne et al, 2020), showed the strongest negative correlation (NES = −1.39) with ATB (Fig. 5A). Furthermore, the transcriptional signature of ATB patient-derived NK cells was featured with suppressed RAP1 and PI3K signaling pathways (Fig. 1C), both of which mediate the activation of downstream MAPKs (including MEK1/2 and ERK1/2) that ultimately lead to ETS1 activation (Awasthi et al, 2010; Jiang et al, 2000; Nelson et al, 2010). Regulation of RAP1 and PI3K signaling pathways involves the Src homology domain 2-containing protein 1/2 (SHP1/2), which could be recruited by immunoreceptor tyrosine-based inhibitory motif (ITIM)-containing receptors including LILRB1 and then phosphorylated to exert the inhibitory functions in NK cells (Azoulay-Alfaguter et al, 2015; Krotz et al, 2005; Sayos et al, 2004). Thus, we speculated that the LILRB1-HLA-G interaction suppresses MEK1/

2-ERK1/2-ETS1 MAPK signaling via SHP1/2 to inhibit cytotoxic activity and promote apoptosis of NK cells from ATB patients (Fig. 5B). Supportively, NK cells from ATB group showed lower levels of phospho-B-raf (an effector of RAP1 driving phosphorylation of MEK1/2), phospho-MEK1/2, phospho-ERK1/2, phospho-ETS1, and ETS1, but higher levels of LILRB1, phospho-SHP1/2, and cleaved caspase-3 (a pro-apoptotic caspase) than that from HC and LTBI groups when co-cultured either with or without Mtb-infected MDMs (Fig. 5C,D). Moreover, neutralizing HLA-G by anti-HLA-G (87G) mAb increased levels of phospho-B-raf, phospho-MEK1/2, phospho-ERK1/2, phospho-ETS1, and ETS1, and decreased levels of phospho-SHP1/2 and cleaved caspase-3 in LILRB1-overexpressed NK cells, but not in control group of NK cells, when co-cultured with Mtb-infected MDMs (Fig. EV3A,B). These results suggest that the MEK1/2-ERK1/2-ETS1 signaling is inhibited by the LILRB1-HLA-G interaction. We then tested whether this signaling pathway is required for LILRB1-HLA-G interaction-mediated inhibition of cytotoxic activity and promotion of apoptosis in NK cells. To this end, we used dabrafenib mesylate and SCH772984 to specifically inhibit B-raf and ERK1/2, respectively (Figs. 5B and EV3C,D), and we found that both of them markedly reduced LILRB1-blockade-induced increase of CD107a expression and decrease of apoptosis in ATB patient-derived NK cells co-cultured with Mtb-infected MDMs (Fig. EV3E,F). Thus, the MEK1/2-ERK1/2-ETS1 axis is required for the LILRB1-HLA-G interaction-mediated inhibition of cytotoxic activity and promotion of apoptosis in NK cells during Mtb infection. In addition, SCH772984 was relatively more effective than dabrafenib mesylate in diminishing LILRB1-blockade-induced increase of CD107a expression and decrease of apoptosis (Figs. 5B and EV3E,F). This phenomenon might be explained by the fact that dabrafenib mesylate only blocks RAP1 signaling-, but not PI3K signaling-, mediated activation of the MEK1/2-ERK1/2-ETS1 axis in NK cells upon Mtb infection, while SCH772984 can inhibit both signaling pathways (Awasthi et al, 2010; Jiang et al, 2000). Likewise, we also used TPI-1 and SHP099 to specifically inhibit SHP1 and SHP2, respectively, or used NSC87877 to inhibit both of them, to confirm whether they are involved in mediating the LILRB1-HLA-G interaction-dependent inhibitory signals in NK cells. Strikingly, treatment of TPI-1, SHP099, or NSC87877 increased the activation of the MEK1/2-ERK1/2-ETS1 signaling pathway in ATB patient-derived NK cells during Mtb infection (Fig. EV3G–I), and accordingly increased CD107a expression and decreased apoptosis of NK cells either by themselves or by combined use with anti-LILRB1 blocking antibody (Fig. EV3J,K). Therefore, both SHP1 and SHP2 are involved in mediating the inhibition of NK cell cytotoxic activity upon HLA-G and LILRB1

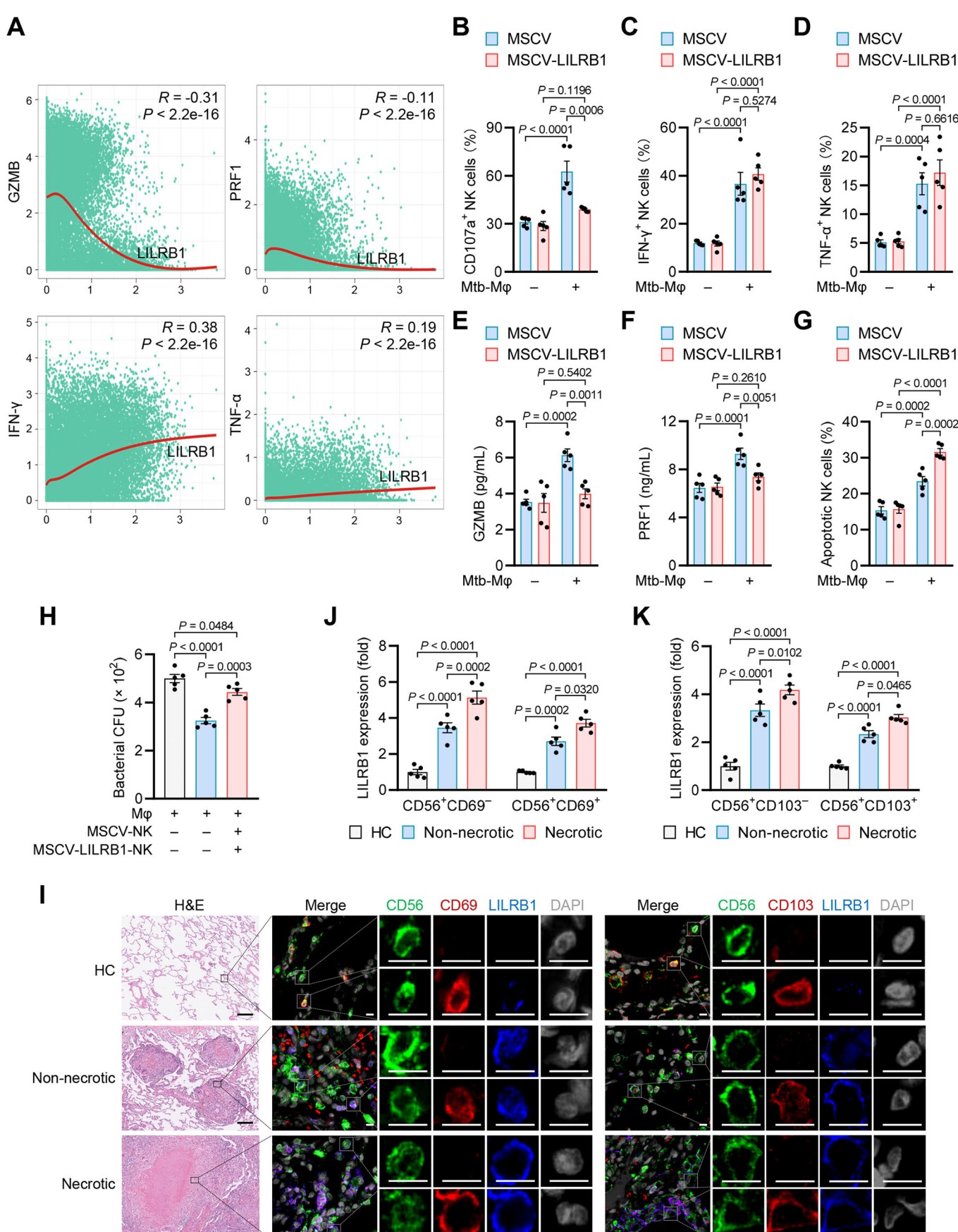

**Figure 3. Overexpression of LILRB1 dampens anti-Mtb immune functions of NK cells.**

(A) Correlation analysis of expression between LILRB1 and other markers including GZMB, PRF1, IFN-γ, and TNF-α based on CyTOF single-cell data ($n = 35,000$ cells). (B–D) Percentages of CD107a$^+$ (B), IFN-γ$^+$ (C), or TNF-α$^+$ (D) cells within total NK cells. (E, F) Supernatant levels of GZMB (E) and PRF1 (F) in MDM-NK cell co-cultures. (G) Percentages of apoptotic cells within total NK cells. (H) Mtb survival in MDM-NK cell co-cultures. For (B–H), HC donor-derived NK cells with overexpression of MSCV or MSCV-LILRB1 were co-cultured with or without Mtb-infected MDMs (Mtb-Mφ) for 24 h. (I) Histological and immunofluorescence analysis of human lung sections containing non-necrotic or necrotic TB granulomas or healthy control tissues adjacent to granulomatous lesions. Left, representative images (scale bars, 200 μm) for hematoxylin and eosin (H&E) staining. Right, representative images (scale bars, 10 μm) for cells stained with antibodies against LILRB1 (blue), CD56 (green), and CD69 or CD103 (red). Nuclei were stained with DAPI (gray). (J, K) Quantitation of LILRB1 expression in CD56$^+$CD69$^+$ or CD56$^+$CD69$^-$ NK cells (J) and CD56$^+$CD103$^+$ or CD56$^+$CD103$^-$ NK cells (K). For (I–K), lung sections from 5 ATB patients were examined. Data are mean ± SEM ($n = 5$ donors per group) in (B–H, J, K). Statistical significance was determined using two-way ANOVA (B–G, J, K) or one-way ANOVA (H) with Tukey's post-hoc test. Results are representative of three independent experiments. Source data are available online for this figure.

interaction. To further confirm the necessity of MAPK signaling in LILRB1-HLA-G interaction-mediated suppression of NK cell anti-Mtb function, we then examined the effects of SCH772984 and NSC87877 on NK cells from individuals at different status of Mtb infection. Consistently, LILRB1-blockade induced increase of CD107a$^+$ cell proportion, reduction of apoptosis, and enhancement of anti-Mtb capacity in NK cells from ATB patients when co-cultured with Mtb-infected MDMs, and ERK1/2 inhibition efficiently abolished these effects (Fig. 5E–G). Also, ERK1/2 inhibition led to comparably reduced CD107a expression, increased apoptosis, and attenuated ability of NK cells to control Mtb survival within MDMs among HC, LTBI, and ATB groups (Fig. 5E–G). Furthermore, SHP1/2 inhibition by itself or by combined treatment with LILRB1 blockade markedly increased CD107a expression, decreased apoptosis, and enhanced anti-Mtb capacity of NK cells from ATB patients (Fig. 5E–G). To avoid the potential off-target effect of inhibitors, we further employed siRNA to knock down *ERK1/2* or *SHP1/2* genes in NK cells to confirm the role of SHP1/2-MAPK signaling in LILRB1-HLA-G interaction-mediated suppression of NK cell anti-Mtb function (Appendix Fig. S6A). We found that *ERK1/2* knockdown eliminated the effects of LILRB1 blockade on ATB patient-derived NK cells including the increase of CD107a expression, decrease of apoptosis, and enhancement of anti-Mtb capacity, while *SHP1/2* knockdown by itself or by combined treatment with LILRB1 blockade markedly increased CD107a expression, decreased apoptosis, and enhanced anti-Mtb capacity of NK cells from ATB patients (Appendix Fig. S6B–D). Together, these data suggest that the LILRB1-HLA-G interaction suppresses MEK1/2-ERK1/2-ETS1 MAPK signaling via SHP1/2, thus leading to reduced cytotoxic activity and increased apoptosis of NK cells.

## Disrupting the HLA-G-LILRB1 interaction enhances anti-Mtb function of the TB-associated exhausted NK subset

Considering that LILRB1 is predominantly expressed in ATB-specific C5 cell subset featured with co-expression of CD56, CD4, and CD14 (Fig. 2B,C), we then collected CD3$^-$CD56$^+$CD4$^+$CD14$^+$ cells from ATB patients to further confirm whether disruption of the HLA-G-LILRB1 interaction by LILRB1 blockade could enhance the anti-Mtb function of this cell subset (Fig. EV4A). Accordant with the CyTOF data, the majority of CD3$^-$CD56$^+$CD4$^+$CD14$^+$ cells express LILRB1, and the frequency of this cell subset was markedly increased in ATB group as compared to that in HC and LTBI groups (Fig. EV4A–C). To confirm the natural killing potential of CD3$^-$CD56$^+$CD4$^+$CD14$^+$ cells, we co-cultured these cells with Mtb-infected MDMs, and we found that when treated by

anti-LILRB1 blocking antibody, CD3$^-$CD56$^+$CD4$^+$CD14$^+$ cells showed increased granule polarization and cytotoxic activity against Mtb-infected MDMs (Fig. 6A,B). Thus, despite expressing the monocyte marker CD14, the CD3$^-$CD56$^+$CD4$^+$CD14$^+$ cells in ATB patients are likely to constitute a unique cell subset of NK cells possessing cytotoxic potential, which can be promoted by LILRB1 blockade. Supportively, time-lapse microscopy further confirmed the direct killing of Mtb-infected MDMs by CD3$^-$CD56$^+$CD4$^+$CD14$^+$ cells, an event much more frequently observed upon LILRB1 blockade (Fig. 6C and Movies EV1,2). Moreover, LILRB1 blockade also increased the expression of CD107a, GZMB, and PRF1, but not IFN-γ or TNF-α, concurrent with reduced apoptosis and enhanced anti-Mtb ability of CD3$^-$CD56$^+$CD4$^+$CD14$^+$ NK cells when co-cultured with Mtb-infected macrophages (Figs. 6D–H and EV4D,E). These data indicate that disrupting the HLA-G-LILRB1 interaction could enhance anti-Mtb function of exhausted NK cells from ATB patients.

## Blocking LILRB1 restores host anti-Mtb immunity in immuno-humanized mice

To test the therapeutic potential of LILRB1 blockade against Mtb infection in vivo, we generated an anti-LILRB1 polyclonal antibody (pAb) that efficiently blocked the binding of HLA-G to cell surface LILRB1 (Appendix Fig. S7A,B). This anti-LILRB1 pAb showed comparable effects on boosting CD107a expression and enhancing anti-Mtb ability of NK cells from ATB patients when co-cultured with Mtb-infected MDMs as compared to GHI/75 mAb (Appendix Fig. S7C,D). Considering the functional inequivalence between LILRB1 and its murine ortholog (Barkal et al, 2018), we then examined the efficacy of LILRB1 blockade in restoration of host anti-Mtb immunity in immuno-humanized mice, which were constructed by engraftment of human peripheral blood mono-nuclear cells (PBMCs) in NOD/ShiLtJGpt-Prkdc$^{em26Cd52}$Il2rg$^{em26Cd22}$/Gpt (NCG) mice (Fig. 7A). At 2-week post-transplantation, each group of human PBMC-transferred mice showed comparable frequencies of human NK cells, T cells, B cells, and monocytes in the lungs and spleens (Appendix Fig. S8A,B). In addition, we detected a higher expression of HLA-G in human macrophages (as indicated by CD68 positivity) in the lungs of mice after being transplanted with ATB patient-derived PBMCs as compared to that in other two groups of mice, and macrophage expression of HLA-G was markedly increased in each group of human PBMC-transferred mice after infection with Mtb for 4 weeks (Fig. EV5A,B). In the phosphate-buffered saline (PBS)-treated control group, mice

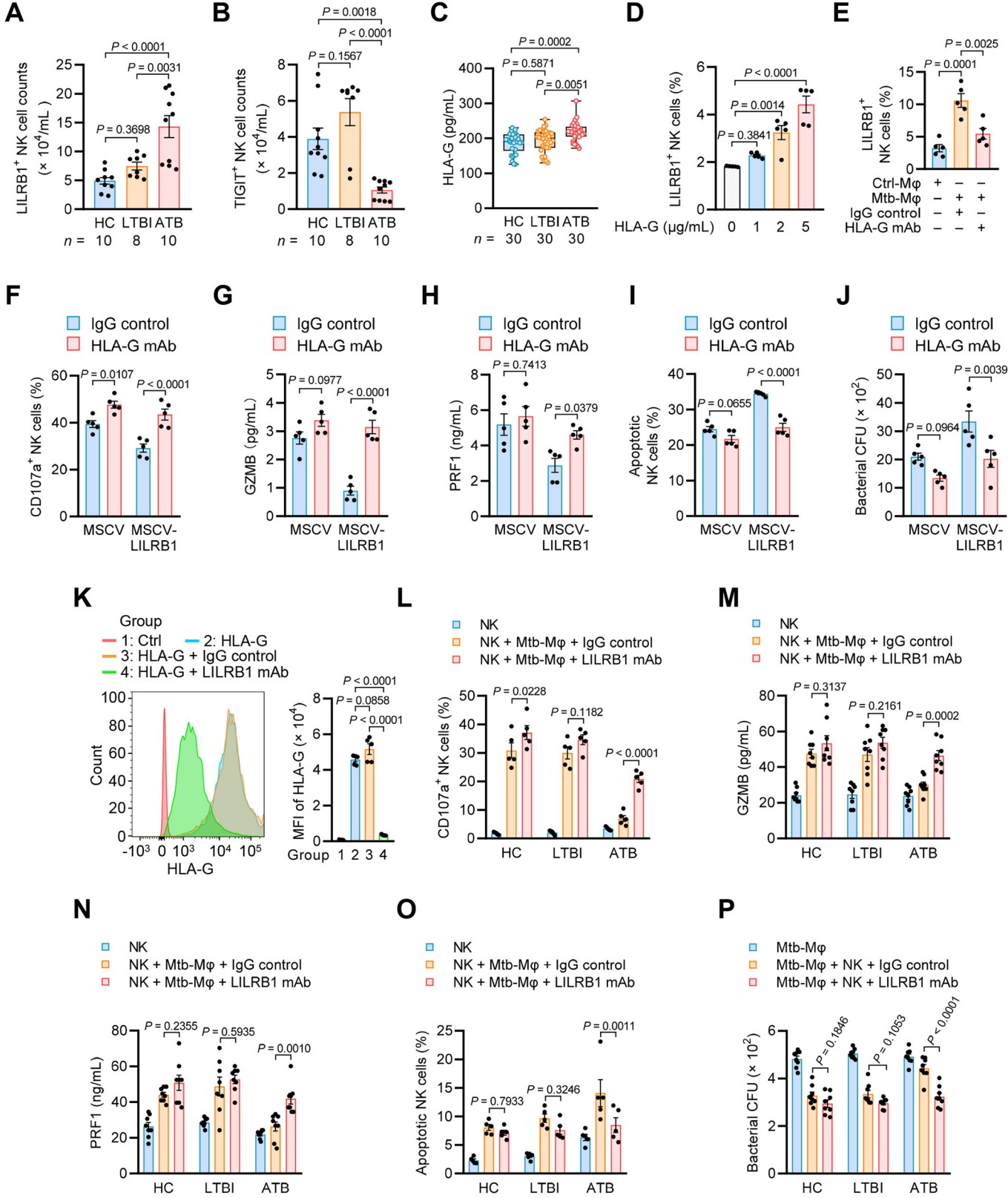

**Figure 4.  LILRB1-HLA-G interaction drives NK cell exhaustion in TB.**

(A, B) The number of LILRB1[+] (A) or TIGIT[+] (B) NK cells in the peripheral blood of individuals (*n*) from HC, LTBI, and ATB groups. (C) Serum levels of soluble HLA-G in individuals (*n*) from HC, LTBI, and ATB groups. Box-whisker plot indicates the interquartile range (box), the median value (line within the box), and the maximum and minimum value (whiskers). (D) Percentages of LILRB1[+] cells within NK cells treated with 0–5 µg/mL HLA-G for 3 days. (E) Percentages of LILRB1[+] cells within NK cells co-cultured with Mtb-Mφ or control MDMs (Ctrl-Mφ) for 3 days with treatment of anti-HLA-G (87G) mAb or IgG control. (F) Percentages of CD107a[+] cells within total NK cells. (G, H) Supernatant levels of GZMB (G) and PRF1 (H) in MDM-NK cell co-cultures. (I) Percentages of apoptotic cells within total NK cells. (J) Mtb survival in MDM-NK cell co-cultures. For (F–J), HC donor-derived NK cells with overexpression of MSCV or MSCV-LILRB1 were co-cultured with Mtb-Mφ for 24 h in the presence of anti-HLA-G (87G) antibody or IgG control. (K) Representative histograms (left) and quantitation (right) of HLA-G binding on the LILRB1-overexpressing HEK293T cells treated with anti-LILRB1 (GHI/75) mAb or IgG control. (L) Percentages of CD107a[+] cells within total NK cells. (M, N) Supernatant levels of GZMB (M) and PRF1 (N) in MDM-NK cell co-cultures. (O) Percentages of apoptotic cells within total NK cells. (P) Mtb survival in MDM-NK cell co-cultures. For (L–P), NK cells from each indicated group were co-cultured with Mtb-Mφ for 24 h in the presence of anti-LILRB1 (GHI/75) mAb or IgG control. Data are mean ± SEM [*n* = 5 donors per group in (D–K) and *n* = 8 donors per group in (L–P)]. Statistical significance was determined using one-way ANOVA (A–E, K) or two-way ANOVA (F–J, L–P) with Tukey's post-hoc test. Results are representative of three independent experiments. Source data are available online for this figure.

transferred with ATB patient-derived PBMCs showed higher inflammatory infiltration, heavier bacterial burden, and more human CD56[+] cells stained with TUNEL in the lungs at 4 weeks after infection, as compared to mice transferred with PBMCs from HC or LTBI group (Fig. 7B). Accordingly, ATB patient-derived PBMC-transferred mice displayed lower GZMB and PRF1 production but higher IFN-γ and TNF-α production in the lungs, and showed higher bacterial loads in the lungs and spleens, as compared to that in the other two groups of mice (Figs. 7C–F and EV5C,D), reflecting host attenuated protective immunity but elevated pathogenic immune responses (Orme et al, 2015). Of note, LILRB1 blockade by anti-LILRB1 pAb, but not IgG control, markedly reduced inflammatory infiltration, bacterial burden, and TUNEL[+] CD56[+] cells with increased production of GZMB and PRF1 but decreased production of IFN-γ and TNF-α in the lungs of mice transferred with ATB patient-derived PBMCs, reaching levels similar to those transferred with PBMCs from HC or LTBI group (Figs. 7B–F and EV5C,D). These data suggest that LILRB1 blockade restores host anti-Mtb immunity in immuno-humanized mice transplanted with PBMCs from ATB patients. Furthermore, inhibition of ERK1/2 by SCH772984 abolished those effects from LILRB1 blockade (Figs. 7B–F and EV5C,D), supporting an indispensable role of MAPK signaling in LILRB1-blockade-mediated restoration of host anti-Mtb immunity. Thus, these data indicate that LILRB1 blockade restores host anti-Mtb immunity of immuno-humanized mice in a MAPK signaling-dependent manner.

## LILRB1 blockade enhances host anti-Mtb immunity in a NK cell-dependent manner

To confirm the essential role of NK cells in LILRB1-blockade-driven restoration of host anti-Mtb immunity, we then established immuno-humanized mice transferred with ATB patients-derived PBMCs in which NK cells were depleted (Fig. 8A; Appendix Fig. S8C). Supportively, treatment of anti-LILRB1 pAb mitigated inflammatory infiltration and reduced TUNEL[+] CD56[+] cells in the lungs of mice infected with Mtb for 4 weeks in a manner dependent on NK cells (Fig. 8B,C). Moreover, NK cell-depletion markedly reduced the LILRB1-blockade-triggered elevation of GZMB and PRF1 production and reduction of IFN-γ and TNF-α production in the lungs of Mtb-infected mice (Figs. 8D,E and EV5E,F). Accordingly, NK cell-depletion markedly diminished the LILRB1-blockade-induced decrease of bacterial loads in both lungs and

spleens of Mtb-infected mice (Fig. 8F,G). Thus, LILRB1 blockade enhances host anti-Mtb immunity by restoring immune functions of NK cells from ATB patients. In summary, our findings identify LILRB1 as a NK cell checkpoint receptor in ATB patients, providing a promising strategy to reverse Mtb-induced exhaustion of NK cells by blocking LILRB1 to unleash host anti-Mtb immunity (Appendix Fig. S9).

## Discussion

NK cells are important effector lymphocytes of the innate immunity against tumor cells and pathogen-infected cells. With rapid progress in NK cell-based immunotherapy for cancers, increasing attention has also been paid to exploiting the therapeutic potential of NK cells for chronic infection including TB. NK cells mediate a prompt cytotoxic function against target cells independent of MHC-mediated antigen presentation. Furthermore, unlike innate phagocytic cells, NK cells do not serve as a cellular niche for Mtb survival or dissemination. These features, together with their broad distribution and trafficking across both lymphoid and non-lymphoid tissues, make NK cells an attractive candidate for anti-TB therapy (Sun et al, 2009). However, the application of NK cell-based immunotherapy for TB has been impeded by limited understanding of the mechanisms underlying Mtb-induced NK cell dysfunction. Here, we reveal that NK cells in ATB patients are distinguished from that in individuals with LTBI by exhibiting the exhaustion characteristics with ineffective anti-Mtb function. We initially find that ATB patients display reduced circulating NK cells, consistent with a recent study indicating that the NK cell frequency in peripheral blood is associated with TB progression (Roy Chowdhury et al, 2018). Further mechanistic investigation reveals that NK cell reduction is probably due to the increased apoptosis of NK cells during the progression of TB. In addition, we find that the frequency of NK cells between HC and LTBI exhibits no significant difference, while a previous study showed an increase of NK cell frequency in LTBI compared to HC (Roy Chowdhury et al, 2018). This is probably due to the difference in regions and ages of study participants between the two studies. Moreover, we show that NK cells from ATB patients exhibit a transcriptional signature characterized by downregulation of multiple signaling pathways (such as PI3K-AKT and RAP1 signaling pathways) and transcriptional factors (such as ETS1) that are important for cytotoxic potential of NK cells (Awasthi et al, 2010; Jiang et al, 2000; Taveirne et al, 2020). Accordingly, we then demonstrate that ATB

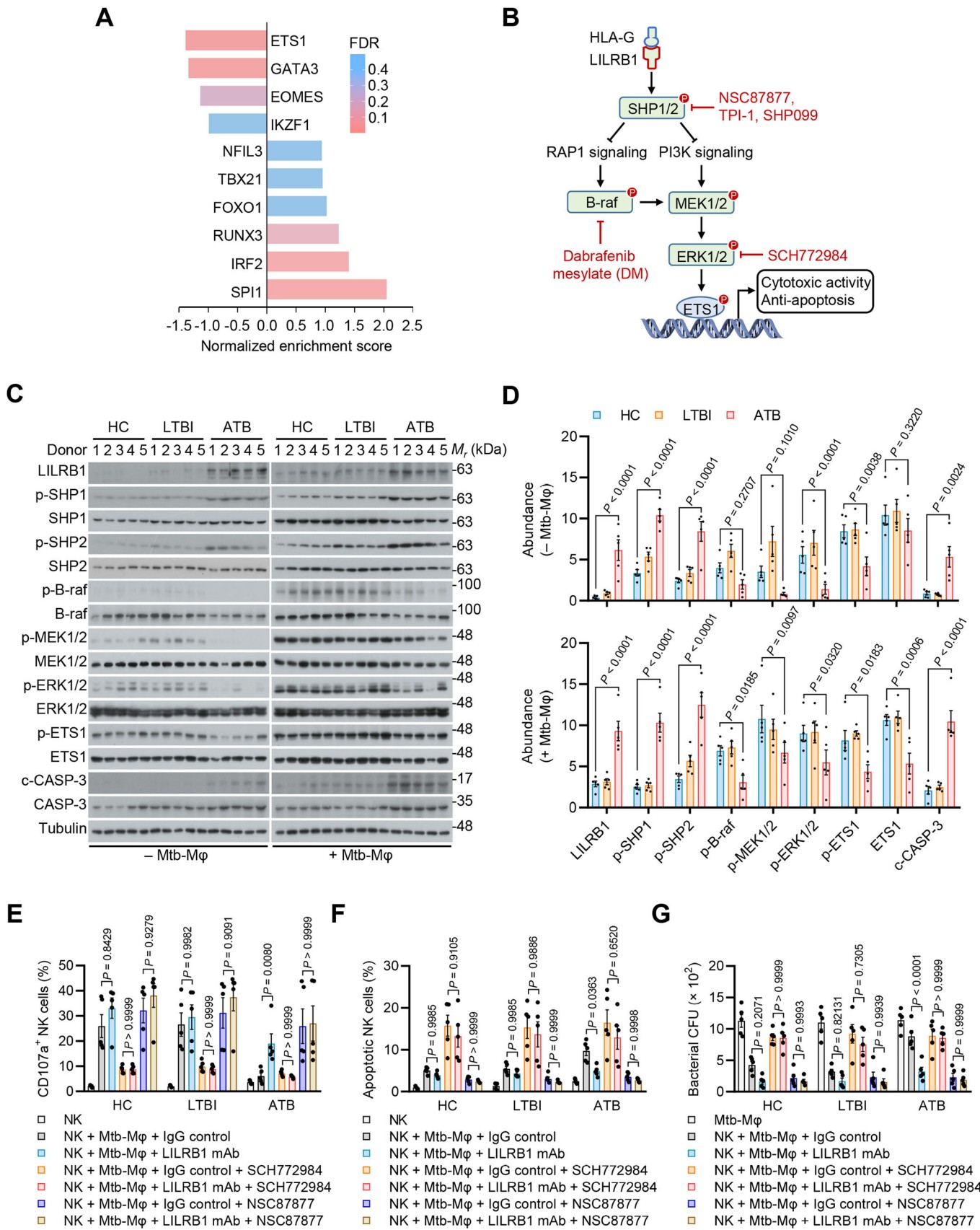

◄ **Figure 5. LILRB1-HLA-G interaction drives NK cell exhaustion by inhibiting MAPK signaling via SHP1/2.**

(A) Enrichment of transcriptional factors correlated with transcriptional signature of NK cells from ATB patients. (B) Schematic model for LILRB1-HLA-G interaction-mediated regulation of NK cell functions. The targets of indicated inhibitors are indicated in red. (C, D) Immunoblotting (C) and quantitation (D) for expression of indicated proteins in NK cells co-cultured with or without Mtb-infected MDMs for 24 h. (E, F) Percentages of CD107a$^+$ (E) or apoptotic (F) cells within total NK cells. (G) Mtb survival in MDM-NK cell co-cultures. For (E–G), NK cells were co-cultured with Mtb-infected MDMs and treated with the indicated inhibitors in the presence of anti-LILRB1 (GHI/75) mAb or IgG control for 24 h. Data are mean ± SEM ($n = 5$ donors per group) in (D–G). Statistical significance was determined using Kolmogorov–Smirnov test (A) or two-way ANOVA with Tukey's post-hoc test (D–G). Results are representative of three independent experiments. Source data are available online for this figure.

patient-derived NK cells are deficient in production of GZMB and PRF1, two cytotoxic effectors which attack infected cells and kill Mtb directly, supporting a critical role of cytotoxic potential in NK cell-mediated protection against Mtb infection (Brill et al, 2001; Esaulova et al, 2021; Lu et al, 2014). These NK cells thereby are ineffective against Mtb, even though they retain the expression of inflammatory cytokines including IFN-γ and TNF-α, strengthening the notion that hosts infected with Mtb exhibit a pathogenic immune feature with hyper-inflammatory responses but compromised protective immunity (Gagneux, 2018; Orme et al, 2015). Most importantly, we identify the inhibitory receptor LILRB1 as an exhaustion marker on NK cells in ATB patients, and reveal that LILRB1 expression indicates a TB-associated NK cell subset (i.e., C5 subset). This C5 subset distinctively co-expresses CD4 and CD14 in addition to CD56, the archetypal phenotypic marker of NK cells. CD4 is associated with cytokine production and chemotaxis of NK cells (Bernstein et al, 2006), and CD14 is linked to an alternative pathway of NK cell development from myeloid precursors in certain inflammatory circumstance (Grzywacz et al, 2011; Perez et al, 2003). Furthermore, this C5 subset is characterized by rare expression of the majority of the activating receptors as revealed by CyTOF, concurrent with compromised anti-Mtb function. Considering these unique phenotypes and functions, we propose this C5 subset (CD3$^-$CD56$^+$CD4$^+$CD14$^+$LILRB1$^+$ cells) as a previously unrecognized TB-associated pathologic NK cell subset, which may be used to indicate changes in host immune status during Mtb infection, and serve as a specific target for host-directed anti-TB therapy. Also, these findings expand the understanding toward the substantial plasticity of NK cells, whose phenotypic and functional properties could be shaped by both host and pathogens during chronic infections (Quatrini et al, 2021).

It is worth mentioning that exhausted NK cells in patients with chronic infectious diseases (such as TB) exhibit certain different characteristics compared to that in cancer patients. For instance, we demonstrate that TB-associated NK cells retain the expression of IFN-γ and TNF-α, distinct from NK cells in cancer patients that display general deficiency in both cytokine production and cytotoxic activity (Cong et al, 2018; Zhang et al, 2018). Similar results were observed during chronic viral infections, in which the exhausted T cells are deficient in a range of effector functions while maintaining the ability to produce IFN-γ (Appay et al, 2000; Wherry et al, 2003). Thus, our results support the notion that immunological exhaustion during chronic infection mainly refers to inefficient effector functions of immune cells that results in a host-pathogen stalemate, rather than complete absence of immune cell functions (Blank et al, 2019; McLane et al, 2019). During tumor development, host immune cells usually go through a long latency period in a non-inflammatory context (Willimsky and Blankenstein, 2005). In contrast, during chronic infections, host immune

cells are likely to have differentiated into functional effector cells in a pathogen-driven inflammatory context, before they become exhausted (Blank et al, 2019; McLane et al, 2019). In the case of Mtb infection, while individuals with LTBI are asymptomatic, ATB patients are characterized by progressive respiratory symptoms with persistent inflammation. Therefore, NK cells in ATB patients probably have gone through an initial effector phase before they become exhausted by continuous stimulation from Mtb, hence exhibiting attenuated anti-Mtb capacity while retaining the expression of some inflammatory cytokines. Thus, the disease-specific features of exhausted immune cells are related to the inflammatory environment in different disease settings, and the developmental trajectories of the exhausted immune cells could be heterogeneous (Blank et al, 2019).

Host anti-Mtb responses are complicated and multilayered due to the constellation of immunologic players including various immune cells and factors, which interact and influence each other to determine the ultimate outcome of infection (Flynn and Chan, 2022). In our efforts to explore the cellular interactions and their signaling network mediating immune cell exhaustion in TB, we reveal that the exhausted anti-Mtb function of NK cells in ATB patients is contributed by the LILRB1-HLA-G interaction between NK cells and Mtb-infected macrophages. Dysregulated expression of HLA-G has been linked to diverse microenvironmental factors such as hypoxia, oxidative stress, nutrient deprivation and cytokines, all of which could be influenced by Mtb upon infection within the host (Carosella et al, 2008; Orme et al, 2015; Qualls and Murray, 2016). Here, we demonstrate that Mtb upregulates HLA-G expression and production in macrophages, accordant with the observation that the serum level of HLA-G in ATB patients is elevated. Furthermore, we identify Mtb glycan antigen peptidoglycan to be the critical determinant that induces HLA-G production in MDMs. The increased amount of HLA-G further upregulates and activates LILRB1 on NK cells to dampen cytotoxic activity and to promote apoptosis of NK cells by inhibiting MAPK signaling. In TB granulomas, LILRB1 expression is upregulated on NK cells, implying compromised NK cell functions in the Mtb-infection microenvironments. Thus, our findings unravel the intricate interactions between innate immune cells, indicating a LILRB1-HLA-G immune checkpoint axis driving NK cell exhaustion in TB. We also notice that the expression of KIR2DL4, another receptor of HLA-G that is upregulated on NK cells to mediate the immune suppression in cancer patients (LeMaoult et al, 2005; Zheng et al, 2021), displays no significant change on NK cells in ATB patients, which finding is further supported by the co-culture experiment data showing that Mtb-infected macrophages have little effect on NK cells expressing KIR2DL4. These findings indicate that different NK cell exhaustion features and regulatory networks are involved

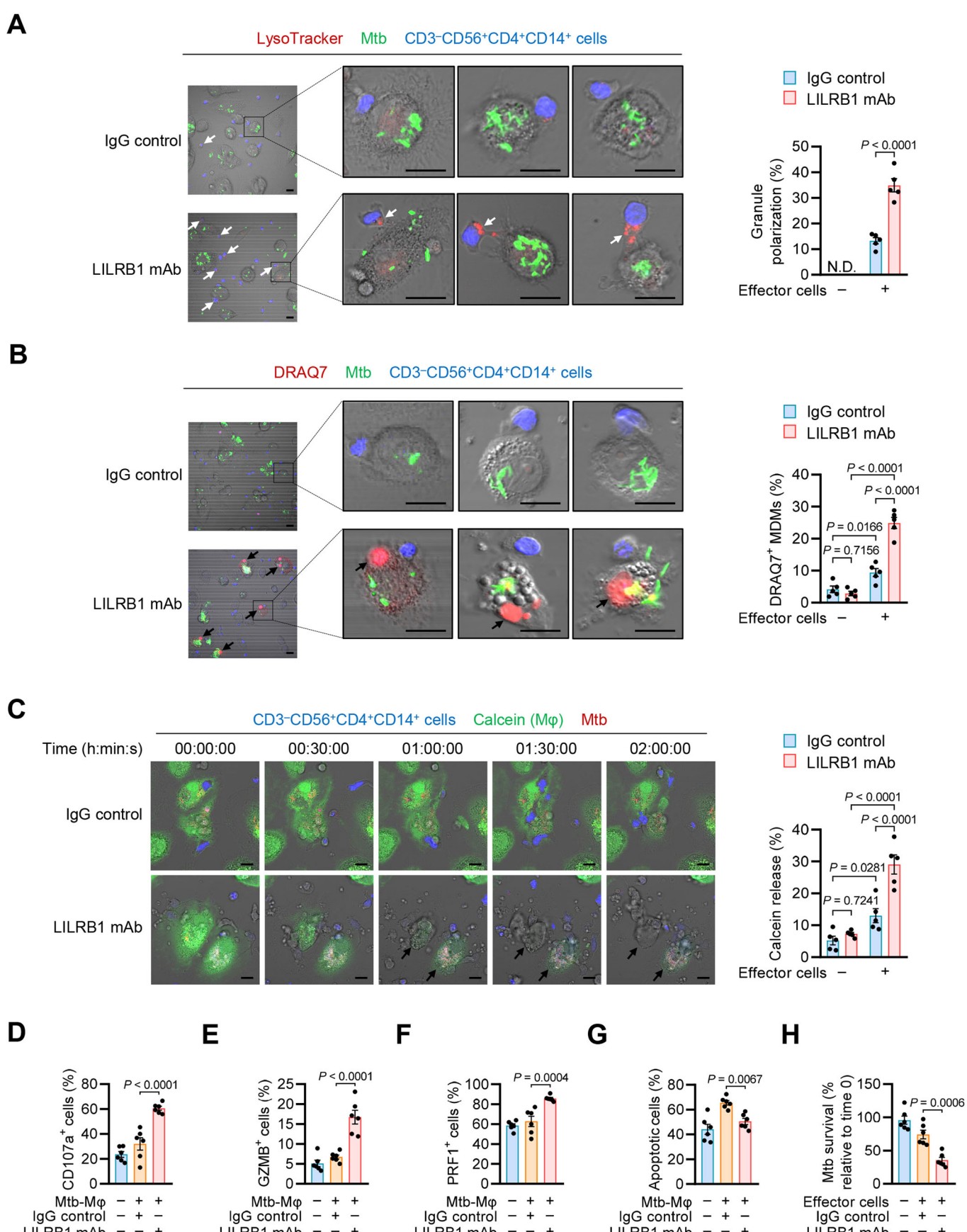

◄ **Figure 6. LILRB1 blockade enhances anti-Mtb function of CD3⁻CD56⁺CD4⁺CD14⁺ NK cells from ATB patients.**

(A, B) Representative images (left) and quantitation (right) for immunofluorescence examining the granule polarization (A) and cytotoxic activity (B) of CD3⁻CD56⁺CD4⁺CD14⁺ cells (effector cells) towards Mtb-infected MDMs (Mtb-Mφ). CD3⁻CD56⁺CD4⁺CD14⁺ cells were co-cultured with Mtb-Mφ at a ratio of 3:1 for 24 h in the presence of anti-LILRB1 (GHI/75) mAb or IgG control. Bacteria were stained with Alexa Fluor 488 succinimidyl ester (green) before infection. CD3⁻CD56⁺CD4⁺CD14⁺ cells were prestained with Hoechst (blue) for visualization, and were labeled with LysoTracker (red; white arrows) to monitor lytic granule polarization towards target cells, a characteristic of NK cells during degranulation. MDM lysis was determined by uptake of DRAQ7 (red; black arrows). Scale bars, 10 μm. Approximately 100 cells were examined. N.D., not detectable. (C) Time-lapse analysis (left) and quantitation (right) of calcein release from Mtb-Mφ. CD3⁻CD56⁺CD4⁺CD14⁺ cells were co-cultured with Mtb-Mφ as in (A, B) for 12 h and were then monitored for 2 h. Bacteria were stained with Alexa Fluor 594 succinimidyl ester (red) before infection. MDMs were preloaded with calcein (green), an indicator of cell viability that was released from dead cells lysed by effector cells (arrows). Scale bars, 10 μm. See also Movies EV1 and EV2. (D–G) Percentages of CD107a⁺ (D), GZMB⁺ (E), PRF1⁺ (F), and apoptotic (G) cells in total CD3⁻CD56⁺CD4⁺CD14⁺ cells. CD3⁻CD56⁺CD4⁺CD14⁺ cells were co-cultured with or without Mtb-Mφ at a ratio of 3:1 for 24 h in the presence of anti-LILRB1 (GHI/75) mAb or IgG control. (H) Mtb survival in MDM-effector cell co-cultures. MDMs were infected with Mtb, followed by co-culture with or without CD3⁻CD56⁺CD4⁺CD14⁺ cells as in (D–G) for 24 h. Data are mean ± SEM [$n = 5$ donors per group in (A–C) and $n = 6$ donors per group in (D–H)]. Statistical significance was determined using two-way ANOVA (A–C) or one-way ANOVA (D–H) with Tukey's post-hoc test. Results are representative of three independent experiments. Source data are available online for this figure.

in chronic infection and tumor microenvironments. It should also be mentioned that LILRB1-HLA-G axis was reported to participate in the education of decidual NK (dNK) cells to inhibit lysis of trophoblasts, thus preventing the rejection of the fetus (Ponte et al, 1999). However, in dNK cells, LILRB1-HLA-G signal mediates the activation of AKT pathway to facilitate PBX1-driven pleiotrophin and osteoglycin transcription for promoting fetal development (Zhou et al, 2020), a signaling pathway which is distinct from that of LILRB1-HLA-G axis-driven exhaustion in TB-associated NK cells. Accordingly, the phenotypic and functional properties of LILRB1⁺ dNK cells are also different from that of TB-associated LILRB1⁺ NK cell subset (Zhou et al, 2020). Thus, the LILRB1-HLA-G axis probably has multifaceted role in driving NK cell phenotype and function in a context-dependent manner.

Immune checkpoint blockade has emerged as a potential strategy to reinforce host immunity and to improve the efficiency of vaccines against infectious diseases including TB (Abers et al, 2019; Jayaraman et al, 2016; Wykes and Lewin, 2018). However, T cell-based checkpoint therapy such as PD-1 blockade is prone to aggravate the course of TB by inducing lethal immunopathology (Barber et al, 2011; Kauffman et al, 2021). Clinically, boosting T cell function by PD-1 blockade during cancer immunotherapy increases the risk of TB reactivation (Barber et al, 2019; Langan et al, 2020; Tezera et al, 2020), and may trigger autoimmunity to exacerbate TB (Postow et al, 2018). These phenomena could be caused by the fact that chronic infection induces aberrant T cell responses at late stage of infection to elicit pathogenic responses that cause lung damage, even though a robust inflammatory response protects host at the early stage of infection (Gagneux, 2018; Orme et al, 2015). This notion is consistent with recent findings showing a low degree of T cell exhaustion in progressive TB granulomas (Flynn and Chan, 2022; Gideon et al, 2022; McCaffrey et al, 2022; Wong et al, 2018). Thus, anti-TB immunotherapy based on checkpoint blockade should promote host protective immunity while avoiding severe pathogenic inflammatory responses. Given that persistent Mtb infection has been linked to dysfunction and exhaustion of host innate immune system, exploring innate immune checkpoints could pave the way to develop alternative anti-TB immunotherapy (Flynn and Chan, 2022; McCaffrey et al, 2022). Here, we identify LILRB1 as a critical checkpoint receptor on NK cells, and further demonstrate that LILRB1 blockade decreases apoptosis and increases cytotoxic activity of NK cells from ATB patients, without enhancing the production of IFN-γ and TNF-α, two cytokines

whose overproduction has been linked to the excessive immunopathology caused by PD-1 blockade in Mtb-infected hosts (Sakai et al, 2016; Tezera et al, 2020). Accordingly, LILRB1 blockade reduces mycobacterial loads and mitigates lung immunopathology in a NK cell-dependent manner in immuno-humanized mice. Together, these results indicate that blocking the checkpoint receptor LILRB1 is a potential strategy to restore host NK cell-dependent anti-Mtb immune responses without causing exaggerated inflammation, paving the way to develop TB immunotherapy that may provide therapeutic potential alone or in combination with antibiotic treatment. Also, this NK cell checkpoint blockade provides a potential strategy for TB vaccine optimization. It should be mentioned that several other pathogens such as human cytomegalovirus and *Plasmodium* species have been reported to modulate LILRB1-mediated inhibition of NK cell functions (Chapman et al, 1999; Harrison et al, 2020). Also, blocking LILRB1 displays beneficial effects on enhancing NK cell anti-tumor activity (Mandel et al, 2022). Given the fact that TB patients are often complicated by concurrent cancer or co-infections, targeting LILRB1 could be a multi-pronged strategy that meets the needs of TB patients with those comorbidities. Taken together, as a proof-of-principle demonstration, our findings suggest that NK cell checkpoint blockade is a rational strategy to unleash host anti-TB immunity.

## Methods

### Setting and participants

ATB patients were diagnosed by significant symptoms of ATB with a positive result of T-SPOT.TB test using a commercially available interferon-γ release assay (IGRA) kit (Chee et al, 2009; Wagstaff and Zellweger, 2006), and were further confirmed by Mtb culture of respiratory specimens. Individuals with LTBI were healthcare workers with positive T-SPOT.TB results but without any clinical ATB symptoms (Chee et al, 2009; Herrera et al, 2011). The individuals without known history of exposure to Mtb were included as HC subjects based on negative results of T-SPOT.TB test. All participants aged ≥16 years old, and had no significant other medical history such as human immunodeficiency virus (HIV) infection, cancers and diabetes. Individuals from HC, LTBI, or ATB groups were matched in multi-factors including sex, age,

**Reagents and tools table**

| Reagent/Resource | Reference or Source | Identifier or Catalog Number |
|---|---|---|
| **Experimental Models** | | |
| Mtb H37Rv strain | ATCC | 27294 |
| HEK293T cells | ATCC | CRL-3216 |
| NCG mice | GemPharmatech | N/A |
| **Recombinant DNA** | | |
| pEGFP-N1 | Clontech | 6085-1 |
| pGEX-6p-1 | GE Healthcare | 28-9546-48 |
| pMSCVpuro | Clontech | PT3303-5 |
| pCMV-VSV-G | Addgene | 8454 |
| pCMV-Gag-Pol | Cell Biolabs | RV-111 |
| **Antibodies** | | |
| Rabbit anti-B-raf | Abcam | ab33899 |
| Rabbit anti-phospho-B-raf | Cell Signaling Technology | 2696 |
| Rabbit anti-MEK1/2 | Cell Signaling Technology | 8727 |
| Rabbit anti-phospho-MEK1/2 | Cell Signaling Technology | 2338 |
| Rabbit anti-Erk1/2 | Cell Signaling Technology | 9102 |
| Rabbit anti-phospho-Erk1/2 | Cell Signaling Technology | 4695 |
| Rabbit anti-ETS1 | Abcam | ab220361 |
| Rabbit anti-phospho-ETS1 (Thr38) | Bioss | bs-13113R |
| Rabbit anti-LILRB1 | Abcam | ab229186 |
| Mouse anti-human CD56 (123C3) | Cell Signaling Technology | 3576 |
| Mouse anti-human CD69 (FN50) | BioLegend | 310902 |
| Mouse anti-human CD103 (B-Ly7) | Invitrogen | 14-1038-82 |
| Mouse anti-human CD68 (E-11) | Santa Cruz | sc-17832 |
| Mouse anti-human HLA-G (87G) | BioLegend | 335902 |
| Rabbit anti-SHP1 (EPR5519) | Abcam | ab124942 |
| Rabbit anti-phospho-SHP1 (Tyr564) (D11G5) | Cell Signaling Technology | 8849 |
| Rabbit anti-SHP2 (Y478) | Abcam | ab32083 |
| Rabbit anti-phospho-SHP2 (Tyr542) | Abcam | ab62322 |
| Rabbit anti-caspase-3 | Cell Signaling Technology | 9662 |
| Rabbit anti-cleaved caspase-3 | Cell Signaling Technology | 9661 |
| Rabbit anti-GAPDH | Santa Cruz Biotechnology | sc-25778 |
| Mouse anti-Tubulin | Sigma-Aldrich | T6199 |
| Pacific Blue anti-human CD3 (UCHT1) | BioLegend | 300418 |
| Brilliant Violet 421 anti-human CD3 (UCHT1) | BioLegend | 300434 |
| PE/Cyanine7 anti-human CD3 (OKT3) | BioLegend | 317333 |
| FITC anti-human CD56 (5.1H11) | BioLegend | 362545 |
| Brilliant Violet 421 anti-human CD19 (HIB19) | BioLegend | 302234 |
| PE anti-human CD19 (HIB19) | BioLegend | 302207 |
| PE anti-human CD14 (63D3) | BioLegend | 367103 |
| PerCP anti-human CD14 (HCD14) | BioLegend | 325631 |
| Brilliant Violet 570 anti-human CD4 (RPA-T4) | BioLegend | 300534 |
| PE anti-human CD107a (eBioH4A3) | Invitrogen | 12-1079-42 |
| FITC anti-human TNF-α (MAb11) | Invitrogen | 11-7349-82 |

| Reagent/Resource | Reference or Source | Identifier or Catalog Number |
|---|---|---|
| Brilliant Violet 421 anti-human IFN-γ (4S.B3) | BioLegend | 502531 |
| Alexa Fluor 647 anti-human/mouse GZMB (GB11) | BioLegend | 515405 |
| Brilliant Violet 421 anti-human PRF1 (dG9) | BioLegend | 308121 |
| APC anti-human LILRB1 (GHI/75) | BioLegend | 333719 |
| PE/Cyanine7 anti-human TIGIT (A15153G) | BioLegend | 372713 |
| APC anti-human KIR2DL4 (mAb 33) | BioLegend | 347008 |
| PE anti-human HLA-G (87G) | BioLegend | 335905 |
| Mouse anti-LILRB1 (GHI/75) mAb | BioLegend | 333702 |
| Mouse anti-HLA-G (87G) | BioLegend | 335902 |
| Goat anti-rabbit IgG | ZSGB-BIO | ZB-2301 |
| Goat anti-mouse IgG | ZSGB-BIO | ZB-2305 |
| Immobilon Western Chemiluminescent HRP Substrate | Millipore | WBKLS0500 |
| **Oligonucleotides and other sequence-based reagents** | | |
| **PCR primers** | **Forward (5′-3′)** | **Reverse (5′-3′)** |
| Extracellular domain of *LILRB1* | GAAGATCTATGGGGCACCTCCCCAAGC | CCGCTCGAGCTAAACCCCCAGGTGCCTTC |
| *LILRB1* | GAAGATCTATGACCCCCATCCTCACGGTC | CCGCTCGAGCTAGTGGATGGCCAGAGTGGCG |
| **siRNA** | **Sense (5′-3′)** | **Antisense (5′-3′)** |
| *ERK1* siRNA-1 | CAGCAGCUGAGCAAUGACCAUAUCU | AGAUAUGGUCAUUGCUCAGCUGCUG |
| *ERK1* siRNA-2 | GACCGGAUGUUAACCUUUAUU | UAAAGGUUAACAUCCGGUCUU |
| *ERK2* siRNA-1 | CAUCACAAGAAGACCUGAAUUGUAU | AUACAAUUCAGGUCUUCUUGUGAUG |
| *ERK2* siRNA-2 | CAAGAGGAUUGAAGUAGAAUU | UUCUACUUCAAUCCUCUUGUU |
| *SHP1* siRNA-1 | GCAAGAACCGCUACAAGAAUU | GCAAGAACCGCUACAAGAAUU |
| *SHP1* siRNA-2 | GAGUGUUGGAACUGAACAAUU | UUGUUCAGUUCCAACACUCUU |
| *SHP2* siRNA-1 | CAGGGACGUUCAUUGUGAUUGAUAU | AUAUCAAUCACAAUGAACGUCCCUG |
| *SHP2* siRNA-2 | GCAAUGACGGCAAGUCUAAUU | UUAGACUUGCCGUCAUUGCUU |
| **Chemicals, Enzymes, and other reagents** | | |
| Middlebrook 7H9 medium | BD Biosciences | 271310 |
| Tween-80 | Sigma-Aldrich | 9005-65-6 |
| Middlebrook 7H10 agar | BD Biosciences | 262710 |
| Dulbecco's modified Eagle's medium | Gibco | C11995500BT-1 |
| Fetal bovine serum | Gibco | 10091148 |
| Human Peripheral Blood Lymphocyte Separation Medium | DAKAWE | 7111011 |
| RPMI 1640 medium | Gibco | C11875500BT |
| Penicillin-streptomycin | Hyclone | SV30010 |
| M-CSF | Gibco | PHC9501 |
| MojoSort Human NK Cell Isolation Kit | BioLegend | 480054 |
| HLA-G ELISA Kit | ABclonal | RK01849 |
| Dabrafenib mesylate | Selleck | S5069 |
| SCH772984 | Selleck | S7101 |
| TPI-1 | Selleck | S6570 |
| SHP099 | Selleck | S8278 |
| NSC87877 | Selleck | S8182 |
| Human GZMB ELISA Kit | DAKAWE | 1118502 |
| Human PRF1 ELISA Kit | DAKAWE | 1118302 |

| Reagent/Resource | Reference or Source | Identifier or Catalog Number |
|---|---|---|
| Cytofix/Cytoperm Fixation/Permeabilization Kit | BD Biosciences | 554714 |
| Annexin V-FITC Apoptosis Detection Kit | Beyotime | C1062 |
| Hoechst 33342 | Cell signaling Technology | 4082 |
| LysoTracker™ Red DND-99 | Invitrogen | L7528 |
| DRAQ7 | Cell signaling Technology | 7406 |
| Calcein AM | Abcam | ab141420 |
| RNA Isolation Kit | Dongsheng | R1061 |
| Cell Acquisition Solution | Fluidigm | 201240 |
| EQ Four Element Calibration Beads | Fluidigm | 201078 |
| MaxPAR antibody Labeling kit | Fluidigm | 201153A |
| Cisplatin | Fluidigm | 201064 |
| FACS buffer and perm buffer | eBioscience | 00833356 |
| Hieff Trans Liposomal Transfection Reagent | Yeasen | 40802ES02 |
| HLA-G protein | Solarbio | P05199 |
| ReadiLink xtra Rapid iFluor 647 Antibody Labeling Kit | AAT Bioquest | 1963 |
| Cell Lysis Buffer | Beyotime | P0013 |
| Protein A-Agarose | Santa Cruz | sc-2001 |
| Polybrene | Santa Cruz | sc-134220 |
| Puromycin | InvivoGen | ant-pr-1 |
| ALC-0315 | MedChemExpress | HY-138170 |
| Cholesterol | MedChemExpress | HY-N0322 |
| 1,2-Distearoyl-sn-glycero-3-phosphorylcholine (DSPC) | MedChemExpress | HY-W040193 |
| DMG-PEG2000 | MedChemExpress | HY-112764 |
| Proteinase K | Yeasen | 10401ES80 |
| Protease inhibitor cocktail | Yeasen | 20124ES03 |
| NaIO$_4$ | Adamas | 013474148 |
| Mutanolysin | Sigma-Aldrich | SAE0092 |
| α-mannosidase | Sigma-Aldrich | M7257 |
| Protein A/G agarose | Santa Cruz | sc-2003 |
| Human CD56 MicroBeads | Miltenyi Biotec | 130-050-401 |
| Human IFN-γ ELISA Kit | RayBiotech | ELH-IFNg-1 |
| Human TNF-α ELISA Kit | RayBiotech | ELH-TNFa-1 |
| AR6 buffer | AKOYA Biosciences | AR600250ML |
| Goat serum blocking buffer | CWBIO | CW0130 |
| Fluorescent dye-coupled TSA reagents | FreeThinking | FS1010 |
| **Software** | | |
| FlowJo vx0.7 software | https://www.flowjo.com/ | |
| FV31S-SW 2.3.1.163 | https://www.olympus-global.com/ | |
| ImageJ 1.50e | https://imagej.net/ | |
| Molecular Signatures Database | http://software.broadinstitute.org/gsea/msigdb/ | |
| Echelon Biosciences | https://www.echelon-inc.com/ | |
| IHC Toolbox plugin | https://imagej.net/ij/plugins/ | |
| HALO v3.4.2986.170 | https://learn.indicalab.com/ | |
| Imaris 9.6 | https://imaris.oxinst.com/ | |

| Reagent/Resource | Reference or Source | Identifier or Catalog Number |
|---|---|---|
| GraphPad Prism 8.0 | https://www.graphpad.com/ | |
| **Other** | | |
| Olympus FV3000RS | Olympus | |
| Flow cytometer (Fortessa) | BD Biosciences | |
| Illumina HiSeq X Ten platform | Novogene Bioinformatics Technology | |
| Helios CyTOF Mass Cytometer | Fluidigm | |
| Mass cytometer | Fluidigm | |
| FastPrep-24 System | MP Biomedicals | |
| Aperio CS2 | Leica Biosystems | |
| Pannoramic MIDI scanner | 3D HISTECN | |

and body mass index for RNA sequencing, time-of-flight mass cytometry (CyTOF) or other independent experiments as indicated (Dataset EV1). All participants were recruited from Beijing Chest Hospital, a TB specialized hospital in China.

## Bacterial strains, mammalian cell lines, and plasmids

Mtb H37Rv strains were obtained from the American Type Culture Collection (ATCC, Cat# 27294), and were grown in Middlebrook 7H9 medium (BD Biosciences, Cat# 271310) supplemented with 10% oleic acid-albumin-dexrose-catalase (OADC) and 0.05% Tween-80 (Sigma, Cat# 9005-65-6), or on Middlebrook 7H10 agar (BD Biosciences, Cat# 262710) supplemented with 10% OADC. HEK293T cells were obtained from the ATCC (Cat# CRL-3216), and were cultured in Dulbecco's modified Eagle's medium (DMEM) (Gibco, Cat# C11995500BT-1) supplemented with 10% fetal bovine serum (FBS) (Gibco, Cat# 10091148). pEGFP-N1 vector (Clontech, Cat# 6085-1) was used for mammalian expression of LILRB1 with C-terminal green fluorescent protein (GFP) (Wang et al, 2020). pGEX-6p-1 vector (GE Healthcare, Cat# 28-9546-48) was used for prokaryotic expression of glutathione S-transferase (GST)-tagged extracellular domain of LILRB1 (Gly$^{24}$–Val$^{461}$) (forward primer: 5′-GAAGATCTATGGGGCACCTCCCCAAGC-3′; reverse primer: 5′-CCGCTCGAGCTAAACCCCCAGGTGCCT TC-3′). All plasmids were sequenced at the Beijing Genomics Institute (BGI) for verification.

## Antibodies

All of the antibodies were used according to the manufacturer's instructions. The following commercial antibodies have been used: anti-B-raf (Abcam, Cat# ab33899; RRID: AB_725762), 1:2000 for immunoblotting; anti-phospho-B-raf (Ser445) (Cell Signaling Technology, Cat# 2696; RRID: AB_390721), 1:2000 for immunoblotting; anti-MEK1/2 (Cell Signaling Technology, Cat# 8727; RRID: AB_10829473), 1:2000 for immunoblotting; anti-phospho-MEK1/2 (Ser221) (Cell Signaling Technology, Cat# 2338; RRID: AB_490903), 1:2000 for immunoblotting; anti-Erk1/2 (Cell Signaling Technology, Cat# 9102; RRID: AB_330744), 1:2000 for immunoblotting; anti-phospho-Erk1/2 (Thr202/Tyr204) (Cell Signaling Technology, Cat# 4695; RRID: AB_390779), 1:2000 for immunoblotting; anti-ETS1 (Abcam, Cat# ab220361), 1:2000 for immunoblotting; anti-phospho-ETS1 (Thr38) (Bioss, Cat# bs-13113R), 1:2000 for immunoblotting; anti-LILRB1 (EPR21007) (Abcam, Cat# ab229186), 1:2000 for immunoblotting and 1:750 for immunofluorescence; anti-human CD56 (123C3) (Cell Signaling Technology, Cat# 3576; RRID: AB_2149540), 1:100 for immunofluorescence; anti-human CD69 (FN50) (BioLegend, Cat# 310902; RRID: AB_314837), 1:200 for immunofluorescence; anti-human CD103 (B-Ly7) (Invitrogen, Cat# 14-1038-82; RRID: AB_467412), 1:100 for immunofluorescence; anti-human CD68 (E-11) (Santa Cruz, Cat# sc-17832; RRID: AB_627157), 1:100 for immunofluorescence; anti-human HLA-G (87 G) (BioLegend, Cat# 335902; RRID: AB_1227707), 1:200 for immunofluorescence; anti-SHP1 (EPR5519) (Abcam, Cat# ab124942; RRID: AB_10976224), 1:2000 for immunoblotting and 1:50 for immunoprecipitation; anti-phospho-SHP1 (Tyr564) (D11G5) (Cell Signaling Technology, Cat# 8849; RRID: AB_11141050), 1:1000 for immunoblotting; anti-SHP2 (Y478) (Abcam, Cat# ab32083; RRID: AB_777915), 1:2000 for immunoblotting and 1:50 for immunoprecipitation; anti-phospho-SHP2 (Tyr542) [EP508(2)Y] (Abcam, Cat# ab62322; RRID: AB_945452), 1:1000 for immunoblotting; anti-caspase-3 (Cell Signaling Technology, Cat# 9662; RRID: AB_331439), 1:2000 for immunoblotting; anti-cleaved caspase-3 (Cell Signaling Technology, Cat# 9661; RRID: AB_2341188), 1:1000 for immunoblotting; anti-GAPDH (Santa Cruz Biotechnology, Cat# sc-25778; RRID: AB_10167668), 1:5000 for immunoblotting; anti-Tubulin (Sigma-Aldrich, Cat# T6199; RRID: AB_477583), 1:1000 for immunoblotting; Pacific Blue anti-human CD3 Antibody (UCHT1) (BioLegend, Cat# 300418; RRID: AB_493095), 1:200 for flow cytometry; Brilliant Violet 421 anti-human CD3 (UCHT1) (BioLegend, Cat# 300434; RRID: AB_10962690), 1:200 for flow cytometry; PE/Cyanine7 anti-human CD3 Antibody (OKT3) (BioLegend, Cat# 317333; RRID: AB_2561451), 1:200 for flow cytometry; FITC anti-human CD56 (5.1H11) (BioLegend, Cat# 362545; RRID: AB_2565963), 1:200 for flow cytometry; Brilliant Violet 421 anti-human CD19 (HIB19) (BioLegend, Cat# 302234; RRID: AB_11142678), 1:200 for flow cytometry; PE anti-human CD19 (HIB19) (BioLegend, Cat# 302207; RRID: AB_314237), 1:200 for flow cytometry; PE anti-human CD14 (63D3) (BioLegend, Cat# 367103; RRID: AB_2565887), 1:200 for flow cytometry; PerCP anti-human CD14 (HCD14) (BioLegend, Cat# 325631; RRID: AB_2563327), 1:200 for flow cytometry; Brilliant Violet 570 anti-human CD4 (RPA-T4) (BioLegend, Cat# 300534; RRID: AB_2563791), 1:200 for flow cytometry; PE anti-human CD107a (eBioH4A3) (Invitrogen, Cat#

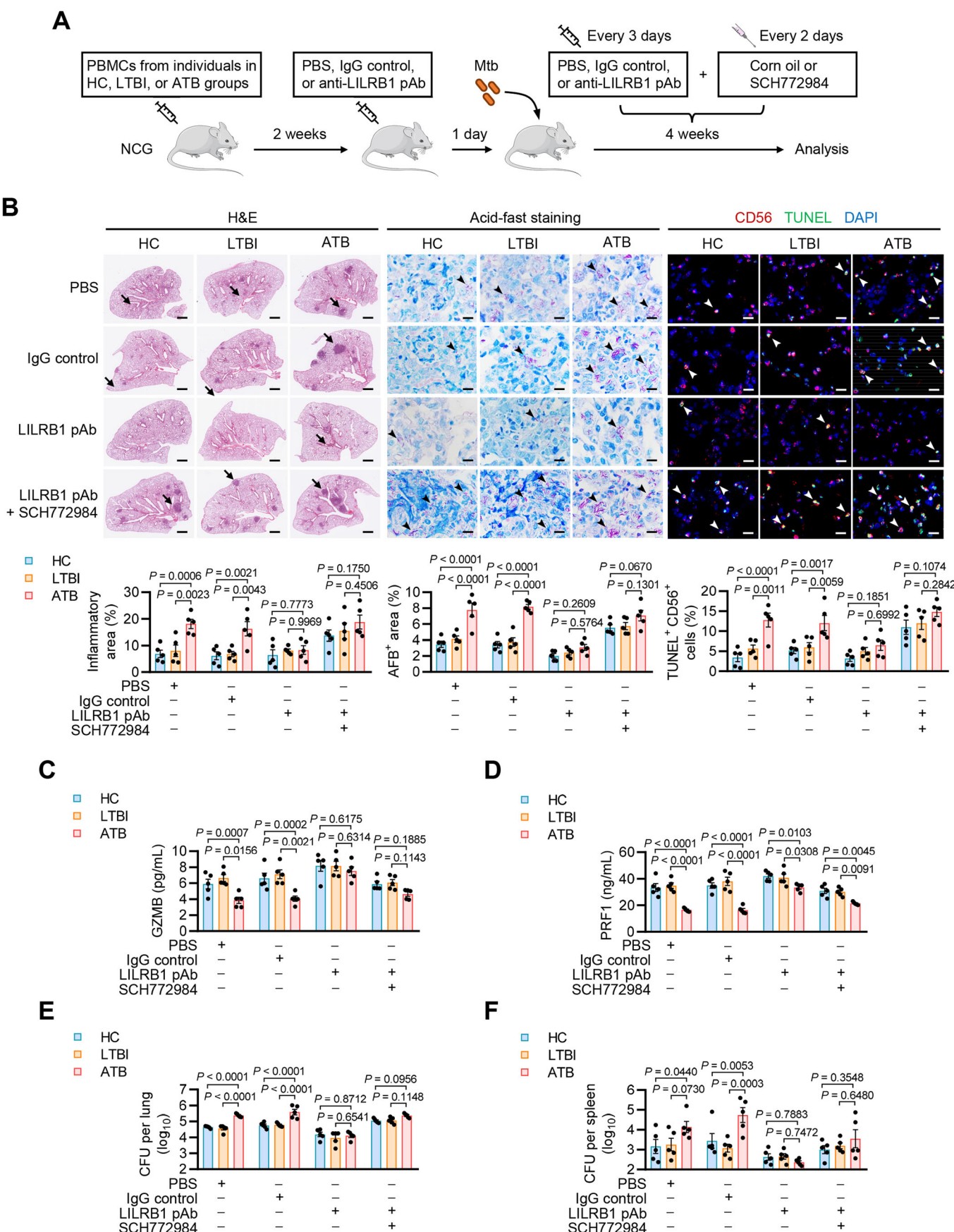

**Figure 7. Blocking LILRB1 restores host anti-Mtb immunity in immuno-humanized mice.**

(A) Schematic representation of the experimental design. PBMCs from individuals in HC, LTBI, or ATB group were transplanted to NCG mice. Two weeks later, mice were infected with Mtb by aerosol (~100 CFUs) for 4 weeks with treatment of PBS (as control), IgG control, or anti-LILRB1 blocking pAb every 3 days, together with treatment of corn oil (as control) or SCH772984 every 2 days. (B) Representative images (upper) and quantitation (lower) for H&E staining, acid-fast staining, and immunostaining of lung sections. Scale bars, 1 mm, 10 μm, and 10 μm, respectively. For immunostaining, NK cells were stained with anti-human CD56 antibody (red), apoptotic cells were stained with TUNEL (green), and nuclei were stained with DAPI (blue). Black arrows indicate granuloma-like inflammatory lesions. Black arrowheads indicate Mtb. White arrowheads indicate TUNEL-positive NK cells. (C, D) Enzyme-linked immunosorbent assay (ELISA) of GZMB (C) or PRF1 (D) in the lungs. (E, F) Bacterial CFUs in the lungs (E) or spleens (F). For (B–F), each group (HC, LTBI, or ATB) includes 5 donors, and each donor's PBMCs were transplanted to four mice, which were subjected to four different treatments as indicated (PBS, IgG control, anti-LILRB1 blocking pAb, and anti-LILRB1 blocking pAb along with SCH772984, respectively). Data are mean ± SEM ($n = 5$ mice per treatment group). Statistical significance was determined using two-way ANOVA with Tukey's post-hoc test (B–F). Results are representative of two independent experiments. Source data are available online for this figure.

12-1079-42; RRID: AB_10853326), 1:200 for flow cytometry; FITC anti-human TNF-α (MAb11) (Invitrogen, Cat# 11-7349-82; RRID: AB_465424), 1:200 for flow cytometry; Brilliant Violet 421 anti-human IFN-γ (4S.B3) (BioLegend, Cat# 502531; RRID: AB_10900083), 1:200 for flow cytometry; Alexa Fluor 647 anti-human/mouse GZMB (GB11) (BioLegend, Cat# 515405; RRID: AB_2294995), 1:200 for flow cytometry; Brilliant Violet 421 anti-human PRF1 (dG9) (BioLegend, Cat# 308121; RRID: AB_2566203), 1:200 for flow cytometry; APC anti-human LILRB1 (GHI/75) (BioLegend, Cat# 333719; RRID: AB_2728293), 1:200 for flow cytometry; PE/Cyanine7 anti-human TIGIT (A15153G) (BioLegend, Cat# 372713; RRID: AB_2632928), 1:200 for flow cytometry; APC anti-human KIR2DL4 (mAb 33) (BioLegend, Cat# 347008; RRID: AB_2130691), 1:200 for flow cytometry; PE anti-human HLA-G (87G) (BioLegend, Cat# 335905; RRID: AB_1227710), 1:200 for flow cytometry. Information for antibodies used in mass cytometry analysis is listed in Appendix Table S1.

## Isolation of PBMCs and analysis of immune cell frequency by fluorescence-activated cell sorting (FACS)

PBMCs were isolated from the ethylenediaminetetraacetic acid (EDTA)-treated peripheral blood of participants using Ficoll density gradient separation medium according to the manufacturer's instruction (DAKAWE, Cat# 7111011). For frequency analysis of each immune cell subset, the isolated PBMCs were stained with the indicated surface markers of NK cells, T cells, B cells, and monocytes (see below) for 20 min at room temperature. For frequency analysis of LILRB1$^+$, TIGIT$^+$ or KIR2DL4$^+$ cells within each indicated immune cell subset, cells were stained with APC anti-human LILRB1 antibody, PE/Cyanine7 anti-human TIGIT or APC anti-human KIR2DL4 antibody at 4 °C for 20 min in the dark. Flow cytometric analysis was performed on a flow cytometer (Fortessa, BD Biosciences) and data were analyzed using FlowJo vx0.7 software (Tree star incorporation; https://www.flowjo.com/).

## Flow cytometry gating strategy

NK cell gating: FSC, SSC, CD3$^-$, CD19$^-$, CD56$^{bright/dim}$; T cell gating: FSC, SSC, CD3$^+$, CD19$^-$, CD56$^-$; B cell gating: FSC, SSC, CD3$^-$, CD19$^+$, CD56$^-$; monocyte gating: FSC, SSC, CD3$^-$, CD19$^-$, CD14$^+$; LILRB1$^+$ NK cells: FSC, SSC, CD3$^-$, CD19$^-$, CD56$^{bright/dim}$, LILRB1$^+$; LILRB1$^+$ T cells: FSC, SSC, CD3$^+$, CD19$^-$, CD56$^-$, LILRB1$^+$; LILRB1$^+$ B cells: FSC, SSC, CD3$^-$, CD19$^+$, CD56$^-$, LILRB1$^+$; LILRB1$^+$ monocytes: FSC, SSC, CD3$^-$, CD19$^-$, CD14$^+$, LILRB1$^+$;

TIGIT$^+$ NK cells: FSC, SSC, CD3$^-$, CD19$^-$, CD56$^{bright/dim}$, TIGIT$^+$; KIR2DL4$^+$ NK cells: FSC, SSC, CD3$^-$, CD19$^-$, CD56$^{bright/dim}$, KIR2DL4$^+$.

## Quantification of immune cell counts in peripheral blood

PBMCs were isolated from a total of 2 mL whole blood, and each immune cell subset was stained with the indicated surface markers. Thereafter, the cells were added to the BD Trucount tube (BD Biosciences, Cat# 340334) containing a freeze-dried pellet of fluorescent beads, and the tube was gently rotated to dissolve the lyophilized pellet and release the fluorescent beads. Then, the cells and the fluorescent beads were incubated for 15 min in the dark at room temperature, followed by flow cytometer analysis. A bright and compact cluster representing the fluorescent beads and the indicated immune cell clusters were gated, respectively. A total of 1000 fluorescent beads and a certain number of the indicated immune cell subset were recorded simultaneously. Absolute count of each immune cell subset was calculated as follows: Cell counts/mL = (the number of the indicated immune cell subset/1000 fluorescent beads/2) * the total number of fluorescent beads per tube.

## Preparation of monocyte-derived macrophages (MDMs) and NK cells

For preparation of MDMs, PBMCs were cultured in RPMI 1640 (Gibco, Cat# C11875500BT) containing 10% FBS, 4 mM L-glutamine, 1% penicillin-streptomycin (Hyclone, Cat# SV30010), and 50 ng/mL of recombinant human macrophage colony-stimulating factor (M-CSF) (Gibco, Cat# PHC9501) for 5–7 days at 37 °C with 5% CO$_2$ to allow the differentiation of monocytes into macrophages. NK cell populations (CD3$^-$CD56$^{bright/dim}$) were enriched from PBMCs by negative selection using MojoSort Human NK Cell Isolation Kit (BioLegend, Cat# 480054). CD3$^-$CD56$^+$CD4$^+$CD14$^+$ NK cells were isolated from PBMCs by flow cytometry. The sorting purity of cell subsets of interest was >95% as validated by flow cytometry. NK cells were cultured in RPMI 1640 containing 10% FBS and 50 ng/mL interleukin-2 at 37 °C with 5% CO$_2$ for 24 h before co-culture with macrophages.

## Macrophage infection

Mtb grown to midlogarithmic phase were collected and washed twice with 1 × PBS containing 0.05% Tween-80 and were then pelleted and thoroughly resuspended using DMEM medium with

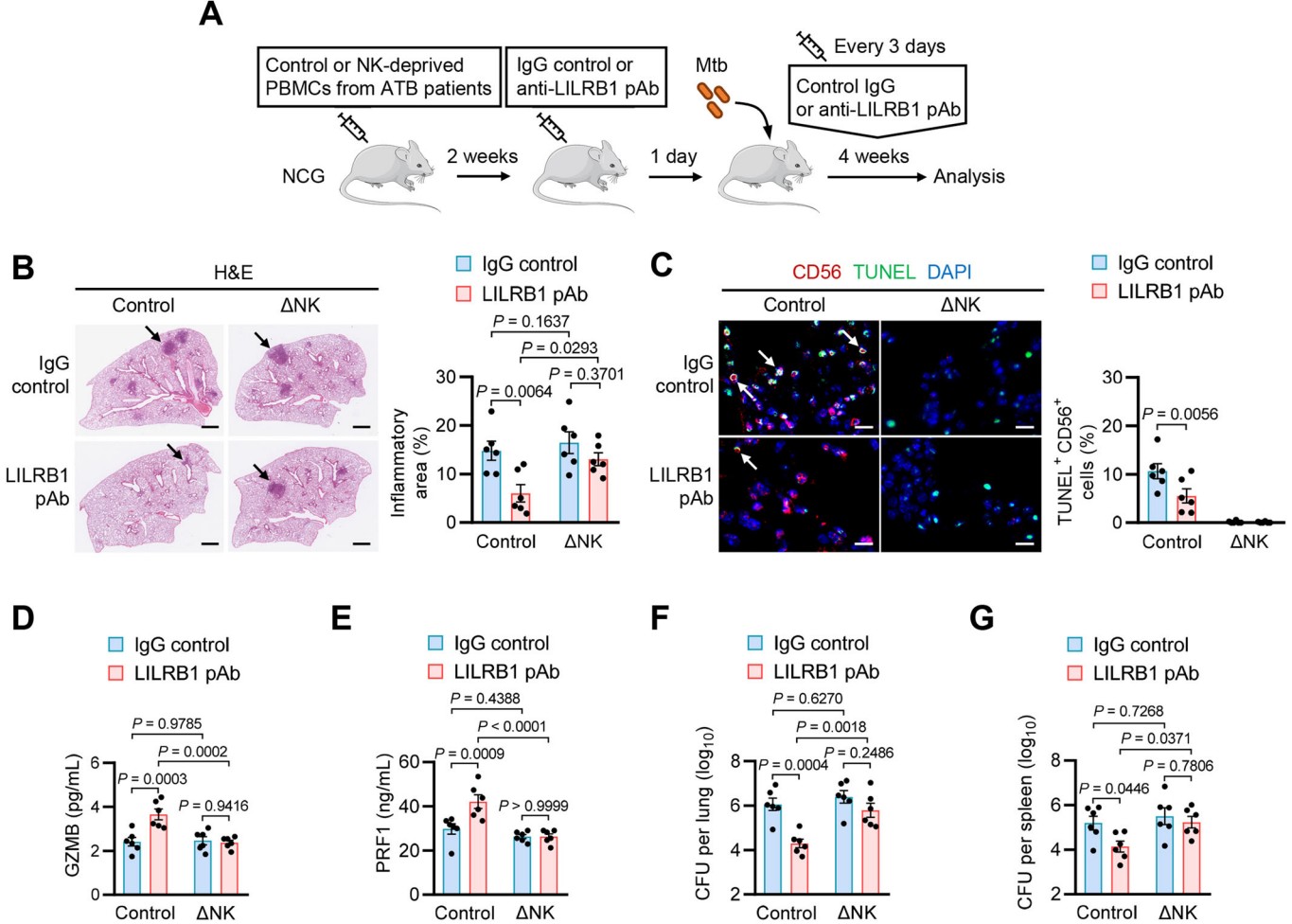

**Figure 8. LILRB1 blockade enhances host anti-Mtb immunity in a NK cell-dependent manner.**

(A) Schematic representation of the experimental design. Control PBMCs or NK cell-depleted (ΔNK) PBMCs from ATB patients were transplanted to NCG mice. Two weeks later, mice were infected with Mtb by aerosol (~100 CFUs) for 4 weeks with treatment of IgG control or anti-LILRB1 blocking pAb every 3 days. (B, C) Representative images (left) and quantitation (right) for H&E staining (B) and immunostaining (C) of lung sections. Scale bars, 1 mm and 10 μm, respectively. For immunostaining, NK cells were stained with anti-human CD56 antibody (red), apoptotic cells were stained with TUNEL (green), and nuclei were stained with DAPI (blue). Black arrows indicate granuloma-like inflammatory lesions. White arrows indicate TUNEL-positive NK cells. (D, E) ELISA of GZMB (D) or PRF1 (E) in the lungs. (F, G) Bacterial CFUs in the lungs (F) or spleens (G). For (B–G), a total of 6 ATB donors were included for collecting PBMCs, and each donor's PBMCs were divided into two treatment groups, of which one was subjected to NK cell deprivation (ΔNK PBMCs) and one was not (control PBMCs). Then, the control PBMCs or ΔNK PBMCs from each donor were transplanted to two mice, of which one was treated with anti-LILRB1 blocking pAb and one was treated with IgG control. Data are mean ± SEM ($n = 6$ mice per treatment group). Statistical significance was determined using two-way ANOVA with Tukey's post-hoc test (B–G). Results are representative of two independent experiments. Source data are available online for this figure.

0.05% Tween-80. MDMs were seeded in 12-well plates at a density of $5 \times 10^4$ cells per well. For infection, MDMs were incubated with Mtb H37Rv at a multiplicity of infection (MOI) of 1 for 2 h at 37 °C to allow bacterial entry into cells. Thereafter, the culture media were discarded, and the cells were washed three times with $1 \times$ PBS to exclude non-internalized bacteria and were then incubated again with the fresh medium. At designated time points, the supernatants were collected for quantitation of soluble HLA-G by enzyme-linked immunosorbent assay (ELISA) using the commercial HLA-G ELISA Kit (ABclonal, Cat# RK01849), and MDMs were collected for frequency analysis of HLA-G$^+$ cells by FACS.

## Macrophage-NK cell co-culture

(1) For macrophage-NK cell co-culture, MDMs were seeded at $5 \times 10^4$ cells per well in 12-well plates and were infected with Mtb H37Rv at a MOI of 1. After 2 h, the cells were washed three times to exclude non-internalized bacteria.

(2) NK cells were then co-cultured with Mtb-infected MDMs at a ratio of 5:1 in RPMI 1640 medium containing 10% FBS for 24–72 h. For blocking LILRB1, cells were treated with 10 μg/mL mouse anti-LILRB1 (GHI/75) mAb (BioLegend, Cat# 333702; RRID: AB_1089089) or 10 μg/mL purified mouse anti-LILRB1 pAb (see below). The control group was treated with 10 μg/mL

mouse IgG control. For neutralizing HLA-G, cells were treated with 10 µg/mL mouse anti-HLA-G (87G) mAb (BioLegend, Cat# 335902) or 10 µg/mL mouse IgG isotype control. For inhibition of B-raf and ERK1/2, cells were treated with 5 µM dabrafenib mesylate (Selleck, Cat# S5069) and 5 µM SCH772984 (Selleck, Cat# S7101), respectively. For inhibition of SHP1 and/or SHP2, cells were treated with 0.5 µM TPI-1 (Selleck, Cat# S6570), SHP099 (Selleck, Cat# S8278) or NSC87877 (Selleck, Cat# S8182). The control group was treated with equal volume of dimethyl sulfoxide.

(3) At designated time points, the supernatants were used to analyze the expression of GZMB or PRF1 by ELISA using the commercial ELISA Kits (Human GZMB ELISA Kit, DAKAWE, Cat# 1118502; Human PRF1 ELISA Kit, DAKAWE, Cat# 1118302).

(4) Cells were collected for isolation of NK cells using MojoSort Human NK Cell Isolation Kit (as described above) for immunoblotting or FACS. For FACS-based frequency analysis of CD107a$^+$, TNF-α$^+$, IFN-γ$^+$, GZMB$^+$ or PRF1$^+$ cells within total NK cells, BD Cytofix/Cytoperm Fixation/Permeabilization Kit (BD Biosciences, Cat# 554714) was used according to the manufacturer's specifications. Briefly, the isolated NK cells were staining with PE anti-human CD107a antibody at 4 °C for 20 min in the dark. Thereafter, cells were fixed and permeabilized with Fixation/Permeabilization solution (supplied with the kit) for 20 min at 4 °C, followed by staining with FITC anti-human TNF-α, Brilliant Violet 421 anti-human IFN-γ, Alexa Fluor 647 anti-human/mouse GZMB or Brilliant Violet 421 anti-human PRF1 antibodies at 4 °C for 20 min in the dark. For FACS-based frequency analysis of apoptotic cells within NK cells, cells were stained using Annexin V-FITC Apoptosis Detection Kit (Beyotime, Cat# C1062) according to the manufacturer's instructions.

(5) For examining absolute counts of viable MDMs in co-culture systems, nonadherent NK cells were removed along with the culture medium at each designated time point. The adherent MDMs were washed 3 times with 1 × PBS buffer, and were then collected for FACS-based living cell count analysis using BD Trucount tube as described above. The dead cells were excluded by gating based on forward scatter and side scatter (FSC/SSC).

(6) Alternatively, to determine bacterial survival within MDM-NK cell co-cultures, total cells along with the culture media were collected and centrifuged at $3600 \times g$ at 4 °C for 5 min. The pellet was resuspended in 7H9 broth containing 0.05% SDS with 5-min vortex to lyse the cells. Bacterial CFUs were determined by serial-dilution of cell lysates plating on 7H10 agar plates.

## NK cell killing assay

(1) Alexa Fluor (488 or 594) succinimidyl esters (Thermo Fisher) were used for bacterial staining before infection, as described previously (Chai et al, 2019). Briefly, the bacterial strains were washed thrice with Hanks' Balanced Salt Solution (Beyotime) containing 0.05% Tween-80 by vortexing and resuspended in the buffer with adequate dye for 30 min at 37 °C. Then the bacterial was again extensively washed and prepared in DMEM medium with 0.05% Tween-80 for infection.

(2) For live-cell imaging, MDMs were plated in 20-mm glass-bottom dishes (NEST), and were infected with Mtb H37Rv prestained with succinimidyl esters at a MOI of 5 for 2 h.

(3) CD3$^-$CD56$^+$CD4$^+$CD14$^+$ NK cells were then co-cultured with Mtb-infected MDMs at a ratio of 3:1 in RPMI 1640 medium containing 10% FBS for 24 h. For blocking LILRB1, cells were treated with 10 µg/mL mouse anti-LILRB1 (GHI/75) mAb. The control group was treated with 10 µg/mL mouse IgG control. Before co-culture, CD3$^-$CD56$^+$CD4$^+$CD14$^+$ NK cells were prestained with 1 µg/mL Hoechst 33342 (Cell signaling Technology, Cat# 4082) at 37 °C for 10 min for visualization. To monitor lytic granules, CD3$^-$CD56$^+$CD4$^+$CD14$^+$ NK cells were prestained with 50 nM LysoTracker™ Red DND-99 (Invitrogen; Cat# L7528) at 37 °C for 20 min. Alternatively, to detect the MDM lysis after culture, the cell membrane permeabilization marker DRAQ7 (Cell signaling Technology, Cat# 7406) was added into the medium at a final concentration of 3 µM. For calcein-based NK cell killing assay, Mtb-infected MDMs were prestained with 1 µM of calcein AM (Abcam, Cat# ab141420) in the medium for 10 min at 37 °C before co-culture with CD3$^-$CD56$^+$CD4$^+$CD14$^+$ NK cells.

(4) The cell dishes were placed in a humidified chamber supplemented with 5% CO$_2$ at 37 °C during imaging. Images were taken with Olympus FV3000RS confocal microscope and analyzed by FV31S-SW 2.3.1.163 (Olympus; https://www.olympus-global.com/).

(5) To quantify the MDM lysis, calcein release was calculated from the fluorescence at Ex/Em = 485/530 nm measured using a microplate fluorimeter.

## Immunoblotting analysis

Cells were washed twice with 1 × PBS and were then lysed in the Cell Lysis Buffer for Western and IP (Beyotime, Cat# P0013). Proteins were separated by 10% SDS-PAGE gel and transferred to the polyvinylidene fluoride (PVDF) membrane (Millipore, Cat# IPVH00010). The blots were blocked with 5% (w/v) nonfat dry milk in Tris-buffered saline (TBS) solution for 1 h at room temperature, and were then incubated overnight at 4 °C with specific primary antibodies in TBS with Tween-20 (TBST) and 5% (w/v) nonfat dry milk. Following three washes, the blots were incubated with horse-radish peroxidase (HRP)-labeled goat anti-rabbit IgG (ZSGB-BIO, Cat# ZB-2301) or goat anti-mouse IgG (ZSGB-BIO, Cat# ZB-2305) for 1 h at room temperature. Finally, the blots were washed three times, and then developed by Immobilon Western Chemiluminescent HRP Substrate (Millipore, Cat# WBKLS0500) and exposed to X-ray film. For quantifications, densitometry was performed using ImageJ 1.50e (National Institutes of Health; https://imagej.net/) to evaluate the intensity of the immunoblotting band signals.

## Total RNA extraction and RNA sequencing

CD3$^-$CD56$^+$ NK cells were isolated from PBMCs by flow cytometry, and were subjected to total RNA isolation using RNA Isolation Kit (Dongsheng, Cat# R1061) according to the manufacturer's instructions. RNA sequencing was conducted using the Illumina HiSeq X Ten platform (Novogene Bioinformatics Technology) with paired-end 150-bp reads. The adaptors and low-quality bases assessed using

FASTQC 0.11.3 were trimmed by Trimmomatic 0.39 using the following options: TRAILING:20, MINLEN:235 and CROP:235. Trimmed reads were then aligned to the ensembl 79 (GRCh38.p2) reference genome using STAR 2.4.2a. FeatureCounts 1.6.2 was subsequently employed to convert aligned short reads into read counts for each sample. Genes with less than ten counts in two or more samples were removed. The data were then analyzed using R 3.4.4 and DEseq2 1.18.1. Differentially expressed genes of ATB and LTBI groups were identified using Wald statistics test, with fold-change >1.2 and Benjamini–Hochberg (BH)-adjusted $P < 0.05$ as compared to HC group. Cluster analysis and visualization were performed using the pheatmap Bioconductor package 1.0.12.4. Kyoto Encyclopedia of Genes Genomes analysis was carried out using R clusterProfiler package 3.6.078, and the enriched items with BH-adjusted $P < 0.05$ was considered to be statistically significant. Gene set enrichment analysis (GSEA) was performed using online R code from Molecular Signatures Database (http://software.broadinstitute.org/gsea/msigdb/). GSEA calculates the normalized enrichment score (NES) by applying weighted Kolmogorov–Smirnov statistic to a running sum of the ranked list with 1000 permutations and normalized to account for the size of the inputted gene set. The false discovery rate (FDR) < 0.05 were assumed to be statistically significant.

## CyTOF detection and subsequent analysis

Before acquisition, PBMCs were washed twice with $1 \times$ PBS and then resuspended at a concentration of $1.1 \times 10^6$ cells/mL in Cell Acquisition Solution (Fluidigm, Cat# 201240) containing 20% of EQ Four Element Calibration Beads (Fluidigm, Cat# 201078). The PBMCs were acquired on the Helios CyTOF Mass Cytometer (Fluidigm) equipped with a SuperSampler fluidics system (Victorian Airships), and data were collected as .fcs files. CyTOF analysis was performed by PLTTech (Hangzhou, China). For mass cytometry analysis, purified antibodies were obtained from BioLegend, R&D Systems, BD Biosciences, and Fluidigm using clones as listed in Appendix Table S1. Antibody labeling with the indicated metal tag was performed using the MaxPAR antibody Labeling kit (Fluidigm, Cat# 201153A). Conjugated antibodies were titrated for optimal concentration before using. Purified PBMCs were washed once with $1 \times$ PBS and then stained with 100 mL of 250 nM cisplatin (Fluidigm, Cat# 201064) for 5 min on ice to exclude dead cells, and then incubated in Fc receptor blocking solution before staining with surface antibody cocktail for 30 min on ice. Cells were then washed twice with FACS buffer ($1 \times$ PBS buffer containing 0.5% BSA) and fixed in 200 mL of intercalation solution (Maxpar Fix and Perm Buffer containing 250 nM 191/193Ir, Fluidigm) overnight. After fixation, cells were washed once sequentially with FACS buffer and perm buffer (eBioscience, Cat# 00833356), and were then stained with intracellular antibody cocktail for 30 min on ice. Cells were washed and resuspended with deionized water, added with 20% EQ beads (Fluidigm, Cat# 201078), and were then acquired on a mass cytometer (Helios, Fluidigm).

After acquisition, PBMCs were manually gated to retain live, singlet, valid NK cells (CD3⁻CD56^bright/dim) by FlowJo software. For downstream analysis, .fcs files loaded into R. Signal intensities for each channel were arcsinh transformed with a cofactor of 5 [x_transf = asinh(x/5)] and normalized by CytoNorm package.

About 10,000 cell events in each individual sample have been pooled and included in nonlinear dimensionality reduction ($t$-SNE) analysis, with a perplexity of 30 and a theta of 0.5. The R $t$-SNE package for Barnes-Hut implementation of $t$-SNE was used. Data were displayed using the ggplot2 R package. To visualize expression analysis on $t$-SNE maps, the expression (y) was normalized between 0 and 1. Y = (value–minimum)/(maximum–minimum); minimum = 3% quantile, maximum = 97% quantile. Y is 0 for all y smaller than 0; y is 1 for all y bigger than 1. The data from all samples were divided by this y value leading to signal intensities ranging between 0 and 1 for each channel. Clustering analysis was performed using FlowSOM run on all samples simultaneously. To identify the main cell subsets in the datasets, FlowSOM was run with the parameter k = 35, defining the number of clusters. Heatmaps were displayed in R using complex-heatmap, and the expression was normalized between 0 and 1 as mentioned above.

## LILRB1 and HLA-G binding assay

HEK293T cells were transfected with vector encoding GFP-tagged LILRB1 using Hieff Trans Liposomal Transfection Reagent (Yeasen, Cat# 40802ES02) for 24 h. Cells were then collected and incubated with 1 μg/mL mouse anti-human LILRB1 monoclonal antibody (BioLegend, Cat# 333702; RRID: AB_1089089), 1 μg/mL purified mouse anti-human LILRB1 polyclonal antibody or 1 μg/mL mouse IgG control in $1 \times$ PBS buffer at room temperature for 30 min. After three washes with $1 \times$ PBS, cells were incubated with 1 μg/mL HLA-G protein (Solarbio, Cat# P05199) which was labeled with iFluor 647 using ReadiLink xtra Rapid iFluor 647 Antibody Labeling Kit (AAT Bioquest, Cat# 1963). After 1 h incubation at room temperature, cells were washed again and were then analyzed using a flow cytometer (Fortessa, BD Biosciences) with excitation at 488 nm and 647 nm.

## SHP1 and SHP2 activity assays

(1) For immunoprecipitation of intracellular SHP1 or SHP2, approximately $1 \times 10^6$ NK cells were lysed in 200 μL of the Cell Lysis Buffer for Western and IP (Beyotime, Cat# P0013) after two washes with $1 \times$ PBS.

(2) Cellular debris were pelleted by centrifugation at $10,000 \times g$ for 10 min at 4 °C, and the cell lysates were added with anti-SHP1 antibody (1:50) or anti-SHP2 antibody (1:50), followed by incubation at 4 °C for 1 h.

(3) Thereafter, 20 μL of resuspended volume of Protein A-Agarose (Santa Cruz, Cat# sc-2001) were added into the above cell lysates for precipitation of SHP1 or SHP2 by incubation at 4 °C on a rotating device for 1 h.

(4) The immunoprecipitants were then collected and washed 4 times with $1 \times$ PBS, followed by analysis of phosphatase activity towards para-nitrophenyl phosphate (pNPP) at 30 °C in a 50 μL reaction buffer (10 mM pNPP, 60 mM HEPES, pH 7.2, 75 mM NaCl, 75 mM KCl, 1 M EDTA, 5 mM DTT) for 30 min of incubation.

(5) The reaction was quenched by the addition of 100 μL of 1 M NaOH, and the pNPP hydrolysis was measured by absorbance at 405 nm.

## Retroviral transduction of NK cells

A retroviral transduction system was used for overexpression of LILRB1 in NK cells (Imai et al, 2005). Briefly, *LILRB1* gene was cloned into the retroviral plasmid pMSCVpuro (Clontech, Cat# PT3303-5) (forward primer: 5′-GAAGATCTATGACCCC-CATCCTCACGGTC-3′; reverse primer: 5′-CCGCTCGAGC-TAGTGGATGGCCAGAGTGGCG-3′). HEK293T cells were co-transfected with this retroviral construct and two packaging plasmids, pCMV-VSV-G (Addgene, Cat# 8454) and pCMV-Gag-Pol (Cell Biolabs, Cat# RV-111), for 48 h. The retroviral super-natant was then collected and filtered through a 0.45-μm filter, and was used to transduce NK cells in the presence of 4 μg/mL polybrene (Santa Cruz, Cat# sc-134220). Transduced cells were selected using 1 μg/mL puromycin (InvivoGen, Cat# ant-pr-1), and the expression of LILRB1 was verified by immunoblotting and flow cytometry with anti-LILRB1 antibody.

## Delivery of small interfering RNA (siRNA)-loaded lipid nanoparticles (LNPs)

siRNAs were synthesized by Sangon Biotech (Shanghai). For efficient knockdown of the target genes, pooled siRNAs were employed at a final concentration of 50 nM. The ERK1/2 pool oligonucleotides are CAGCAGCUGAGCAAUGACCAUAUCU (which targets *ERK1*), GACCGGAUGUUAACCUUUAUU (which targets *ERK1*), CAUCA-CAAGAAGACCUGAAUUGUAU (which targets *ERK2*), and CAA-GAGGAUUGAAGUAGAAUU (which targets *ERK2*). The SHP1/2 pool oligonucleotides are GCAAGAACCGCUACAAGAAUU (which targets *SHP1*), GAGUGUUGGAACUGAACAAUU (which targets *SHP1*), CAGGGACGUUCAUUGUGAUUGAUAU (which targets *SHP2*), and GCAAUGACGGCAAGUCUAAUU (which targets *SHP2*). siGENOME Non-targeting siRNA (Dharmacon, Cat# D-001220-01) was used as a negative control.

For NK cell delivery of siRNA-loaded LNPs, NK cells were seeded in 12-well plates at a density of $2 \times 10^5$ cells per well. siRNA-loaded LNPs were prepared according to Lipid Nanopar-ticle Protocol provided by Echelon Biosciences (https://www.echelon-inc.com/). Briefly, four lipid components purchase from MedChemExpress, including 12.5 mM ALC-0315 (Cat# HY-138170), 9.625 mM cholesterol (Cat# HY-N0322), 2.5 mM 1,2-Distearoyl-sn-glycero-3-phosphorylcholine (DSPC) (Cat# HY-W040193), and 0.375 mM DMG-PEG2000 (Cat# HY-112764) were dissolved in EtOH for preparation of a 25 mM lipid mixture. RNAs diluted in sodium acetate buffer were mixed with lipids to achieve a 5:1 RNA:lipid ratio (v:v) with a 50 nM concentration, and were then added into the medium. At 48 h after treatment, cells were harvested for immunoblotting to verify the expression of ERK1/2 and SHP1/2, or were used for culture with Mtb-infected MDMs.

## Digestion of Mtb subcellular fractions

Mtb proteins and carbohydrate residues were digested as described previously (Zhao et al, 2024). Briefly, for digestion of Mtb surface proteins, the bacilli were incubated with 100 μg/mL proteinase K (Yeasen, Cat# 10401ES80) at 37 °C for 30 min, and then protease inhibitor cocktail (Yeasen, Cat# 20124ES03) was used to terminate

the reaction. For digestion of carbohydrate residues, the bacteria were treated with 40 mM NaIO₄ (Adamas, Cat# 013474148) to degrade saccharides, with 0.5 KU/mL mutanolysin (Sigma, Cat# SAE0092) to digest peptidoglycan, or with 60 U/mL α-mannosidase (Sigma, Cat# M7257) to transform lipomannan and lipoarabino-mannan into mannoglycolipids for 24 h at 37 °C. The digestion processes were terminated by denaturing the enzymes at 100 °C for 5 min. All digestive products were incubated with MDMs to detect intracellular levels of *HLA-G* mRNA or supernatant levels of HLA-G produced by macrophages.

## Purification of mouse polyclonal antibodies against LILRB1

Purified GST-LILRB1 fusion protein (10 μg) was solubilized in 50 μL of Freund's complete adjuvant and intramuscularly injected into the hind leg of mice. Subsequently, two injections of 10 μg GST-LILRB1 fusion protein in 50 μL of Freund's incomplete adjuvant were given after an interval of 15 days. Ten days after the final injection, animals were bled and titers of anti-LILRB1 antibody were determined by ELISA. Finally, LILRB1 antibody was isolated by passaging the immunized mouse serum on protein A/G agarose (Santa Cruz, Cat# sc-2003).

## Immuno-humanized mouse infection

NCG mice (NOD/ShiLtJGpt-Prkdc^em26Cd52^Il2rg^em26Cd22^/Gpt; strain NO. T001475) were purchased from GemPharmatech (Nanjing, China), and were maintained under barrier conditions in a BSL-3 biohazard animal room (12-h light/dark cycle, 50% relative humidity, at 25–27 °C). All experiments were performed with sex-matched groups of 8-week-old mice. A total of $5 \times 10^6$ PBMCs were purified from the EDTA-treated blood of participants, including ATB, LTBI, and HC individuals. For depletion of NK cells, PBMCs were treated with human CD56 MicroBeads (Miltenyi Biotec, Cat# 130-050-401) to remove NK cells according to the manufacturer's instructions. PBMCs or NK cell-depleted PBMCs were resuspended in 1 × PBS and were then injected into the tail vein of NCG mice. Two weeks after the PBMC injection, mice were challenged by aerosol exposure with Mtb H37Rv (which were collected at the midlogarithmic phase of growth and were resuspended in 10 mL 1 × PBS at an OD₆₀₀ of 0.1) using an inhalation device (Glas-Col) calibrated to deliver ~100 CFUs of Mtb (Chai et al, 2022). For LILRB1 blockade, mice were treated with 100 μL of 2 mg/mL purified mouse anti-LILRB1 pAb, 100 μL of 2 mg/mL mouse IgG control or equal volume of PBS buffer (as control) by intraperitoneal injection once every 3 days, beginning at the day before infection. For ERK1/2 inhibition, mice were orally fed with 0.4 mg SCH772984 dissolved in 100 μL corn oil (Selleck, Cat# S88219) or 100 μL corn oil (as control) once every 2 days, beginning at the day of infection. After 4 weeks of infection, lungs were subjected to section for pathological analysis, or were homogenized with a FastPrep-24 System (MP Biomedicals) for ELISA (Human GZMB and PRF1 ELISA Kits, see above; Human IFN-γ ELISA Kit, RayBiotech, Cat# ELH-IFNg-1; Human TNF-α ELISA Kit, RayBiotech, Cat# ELH-TNFa-1) or CFU counting. Spleens were homogenized in the same way for CFU counting. Experiments or analyses were performed without blinding.

**The paper explained**

**Problem**

Persistent infection of *Mycobacterium tuberculosis* (Mtb), which causes tuberculosis (TB), has been linked to functional exhaustion of host immune cells including natural killer (NK) cells, a group of innate immune cells exerting prompt immune response with cytolytic activity against pathogen-infected cells. Immune checkpoints are inhibitory immunoreceptors that control the duration and magnitude of immune cell responses, but whether and how the checkpoint molecules mediate the functional exhaustion of NK cells during TB infection remain largely unclear.

**Results**

In this study, we demonstrated the compromised immune functions of NK cells in TB patients, and identified leukocyte immunoglobulin-like receptor B1 (LILRB1) as a critical checkpoint receptor that defines a TB-associated NK cell subset and drives NK cell exhaustion in TB patients. Mechanistically, Mtb-infected macrophages display high expression of human leukocyte antigen-G (HLA-G), which upregulates and activates LILRB1 on NK cells to impair their functions. Furthermore, LILRB1 blockade restores NK cell-dependent anti-Mtb immunity in immuno-humanized mice.

**Impact**

Our findings suggest that NK cell checkpoint blockade is a rational strategy to unleash host anti-TB immunity. Moreover, our discovery of the LILRB1-HLA-G axis, which drives exhaustion of TB-associated NK cells, provides a promising target for TB immunotherapy.

## Histopathology and tissue immunofluorescence

Lungs from Mtb-infected mice were fixed by inflating the tissues with 4% formaldehyde, sectioned, and stained with hematoxylin-eosin (H&E) or by the Ziehl-Neelsen method to visualize acid-fast mycobacteria. Slides containing histological sections were scanned with Aperio CS2 (Leica Biosystems). Quantitation of the inflammation or acid-fast bacteria-positive area in each tissue section was performed using ImageJ 1.50e with an IHC Toolbox plugin (National Institutes of Health; https://imagej.net/).

For immunofluorescence analysis of NK cell viability in mouse lungs, the engrafted human NK cells were detected using anti-human CD56 antibody (1:100) with a Two-step IHC Detection Kit (ZSGB-BIO), followed by amplification with tyramide (1:200) using Opal 690 Reagent Pack (AKOYA) according to the manufacturer's instructions. Slides were then rinsed three times with PBS buffer, and were subjected to TUNEL assay using One-step TUNEL Assay Kit (KeyGEN BioTECH) based on the manufacturer's protocol. For multiplex immunofluorescence analysis of human TB granulomas, a multiplexed tyramide signal amplification (TSA) method was performed on 4-μm lung tissue sections achieved from Beijing Chest Hospital (see Dataset EV1 for details). Slides were deparaffinized using xylene and a graded ethanol series, and were washed with distilled water for two times. For antigen retrieval, the tissue was microwaved in AR6 buffer (AKOYA Biosciences, Cat# AR600250ML) at full power for 45 s and at 20% power for 15 min, followed by washing with distilled water and TBST twice each. Next, the tissue was blocked using normal goat serum blocking buffer (CWBIO, Cat# CW0130). The primary antibody was then applied at 4 °C overnight, followed by washing with TBST three

times. The secondary HRP-bound antibody against the primary species was then applied for 10 min at room temperature, followed by washing with TBST three times. Finally, the fluorescent dye-coupled TSA reagents (FreeThinking, Cat# FS1010) were applied for 10 min at room temperature, followed by washing with TBST three times again. The steps of antigen retrieval and blocking were repeated before staining each additional marker. Single staining slides were included in each antibody cycle as controls. DAPI Staining Solution was used to visualize the nuclei and to mount the slides. Slides were scanned using a Pannoramic MIDI scanner (3D HISTECN) and images were analyzed by HALO v3.4.2986.170 (Indica Labs; https://learn.indicalab.com/). Quantitation of the indicated cells in each lung section was performed using Imaris 9.6 (Bitplane; https://imaris.oxinst.com/).

## Statistics

Data are shown as mean and standard error of mean (SEM) or median and 25–75% interquartile range. Statistical analysis was performed using GraphPad Prism 8.0 (https://www.graphpad.com/). Two-way analysis of variance (ANOVA) was used for analysis of experiments with multiple groups and multiple independent variables, and one-way ANOVA was used for analysis of multiple groups with a single independent variable. The Tukey tests were used as follow-up tests to the ANOVAs for comparing every mean with every other mean. Unpaired two-tailed Student's *t*-tests were used for single comparison of two groups. Details of statistical analysis of experiments and number of biological replicates (*n*) can be found in figures and/or figure legends. Unless otherwise indicated, results are representative of three independent experiments.

## Study approval

All experiments using human clinical samples were approved by the Ethics Committee of Beijing Chest Hospital, Capital Medical University (2018KY41). Informed consent was obtained from all human subjects and that the experiments conformed to the principles set out in the WMA Declaration of Helsinki and the Department of Health and Human Services Belmont Report. All animal studies were approved by the Biomedical Research Ethics Committee of Institute of Microbiology, Chinese Academy of Sciences (SQIMCAS2021002). Standard safety procedures for biosafety level-3 (BSL-3) work according to institutional protocols were used throughout.

## For more information

Current status of global tuberculosis: https://www.who.int/health-topics/tuberculosis#tab=tab_1.

Compendium of protein information on LILRB1: https://www.uniprot.org/uniprotkb/Q8NHL6/entry.

Cui Hua Liu Lab website: https://english.im.cas.cn/people_/facultyandstaff/KLPMI/202012/t20201204_256035.html.

## Data availability

The gene expression data from this publication have been deposited to the Gene Expression Omnibus (GEO) database (https://

www.ncbi.nlm.nih.gov/geo/) and assigned the identifier GSE222001. The CyTOF data have been deposited to Flow-Repository and assigned the identifier FR-FCM-Z6ZJ. The unprocessed immunoblots and source data of this paper were provided as source data files. The microscopy images and flow cytometry standard files in this study have been deposited at Figshare (https://doi.org/10.6084/m9.figshare.25983547).

The source data of this paper are collected in the following database record: biostudies:S-SCDT-10_1038-S44321-024-00106-1.

# Peer review information

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

## Acknowledgements

This work was supported by the National Key Research and Development Program of China (2022YFC2302900 and 2021YFA1300200 to CHL and JW), the National Natural Science Foundation of China (82330069, 81825014, and 31830003 to C.H.L., 82372653 to JW, and 82171744 to QC), the Shenzhen Medical Research Funding (B2302035 to CHL), the Strategic Priority Research Program of the Chinese Academy of Sciences (XDB29020000 to CHL), the State Key Laboratory of Proteomics (SKLP-K202001 to CHL), the Youth Innovation Promotion Association CAS (Y2022036 to JW), the CAS Project for Young Scientists in Basic Research (YSBR-010 to JW), and the Special Research Assistant Program of Chinese Academy of Sciences (2022000031 to QC). We thank T. Zhao (Institute of Microbiology, Chinese Academy of Sciences, Beijing) for helping with flow cytometry; J. Hao (Core Facility for Protein Research, Institute of Biophysics, Chinese Academy of Sciences, Beijing) for helping with histological analysis; Y. Teng, D. Duo, and C. Liu (Center for Biological Imaging, Institute of Biophysics, Chinese Academy of Sciences, Beijing) for helping with live-cell imaging.

## Author contributions

**Jing Wang**: Data curation; Formal analysis; Funding acquisition; Validation; Investigation; Methodology; Writing—original draft; Writing—review and editing. **Qiyao Chai**: Data curation; Formal analysis; Funding acquisition; Validation; Investigation; Writing—original draft; Writing—review and editing. **Zehui Lei**: Data curation; Formal analysis; Validation; Writing—original draft; Writing—review and editing. **Yiru Wang**: Data curation; Formal analysis; Validation. **Jiehua He**: Data curation; Formal analysis; Validation; Investigation. **Pupu Ge**: Data curation; Formal analysis. **Zhe Lu**: Data curation; Software; Formal analysis. **Lihua Qiang**: Data curation; Formal analysis. **Dongdong Zhao**: Data curation; Formal analysis. **Shanshan Yu**: Data curation; Formal analysis. **Changgen Qiu**: Data curation; Formal analysis. **Yanzhao Zhong**: Data curation; Formal analysis. **Bing-Xi Li**: Project administration. **Lingqiang Zhang**: Resources. **Yu Pang**: Resources. **George Fu Gao**: Conceptualization; Resources. **Cui Hua Liu**: Conceptualization; Data curation; Formal analysis; Supervision; Funding acquisition; Validation; Investigation; Methodology; Writing—original draft; Writing—review and editing.

Source data underlying figure panels in this paper may have individual authorship assigned. Where available, figure panel/source data authorship is listed in the following database record: biostudies:S-SCDT-10_1038-S44321-024-00106-1.

## Disclosure and competing interests statement

The authors declare no competing interests.

# Expanded View Figures

**Figure EV1. Frequencies of immune cell subsets in individuals from HC, LTBI, and ATB groups.**

(A) Flow diagram summarizing the participant recruitment in this study. See "Methods" and Dataset EV1 for details of inclusion criteria and demographic information of participants included, respectively. (B) Representative results of fluorescence-activated cell sorting (FACS)-based analyses for frequencies of immune cell subsets in peripheral blood of individuals from HC, LTBI, and ATB groups. Each immune cell subsets were characterized using standard cell subset definitions. NK cell gating: FSC, SSC, CD3⁻, CD19⁻, CD56$^{bright/dim}$; T cell gating: FSC, SSC, CD3⁺, CD19⁻, CD56⁻; B cell gating: FSC, SSC, CD3⁻, CD19⁺, CD56⁻; monocyte gating: FSC, SSC, CD3⁻, CD19⁻, CD14⁺. (C) The number of immune cell subsets in peripheral blood of individuals (*n*) from HC, LTBI, and ATB groups. Statistical significance was determined using one-way ANOVA with Tukey's post-hoc test (C). Source data are available online for this figure.

▶

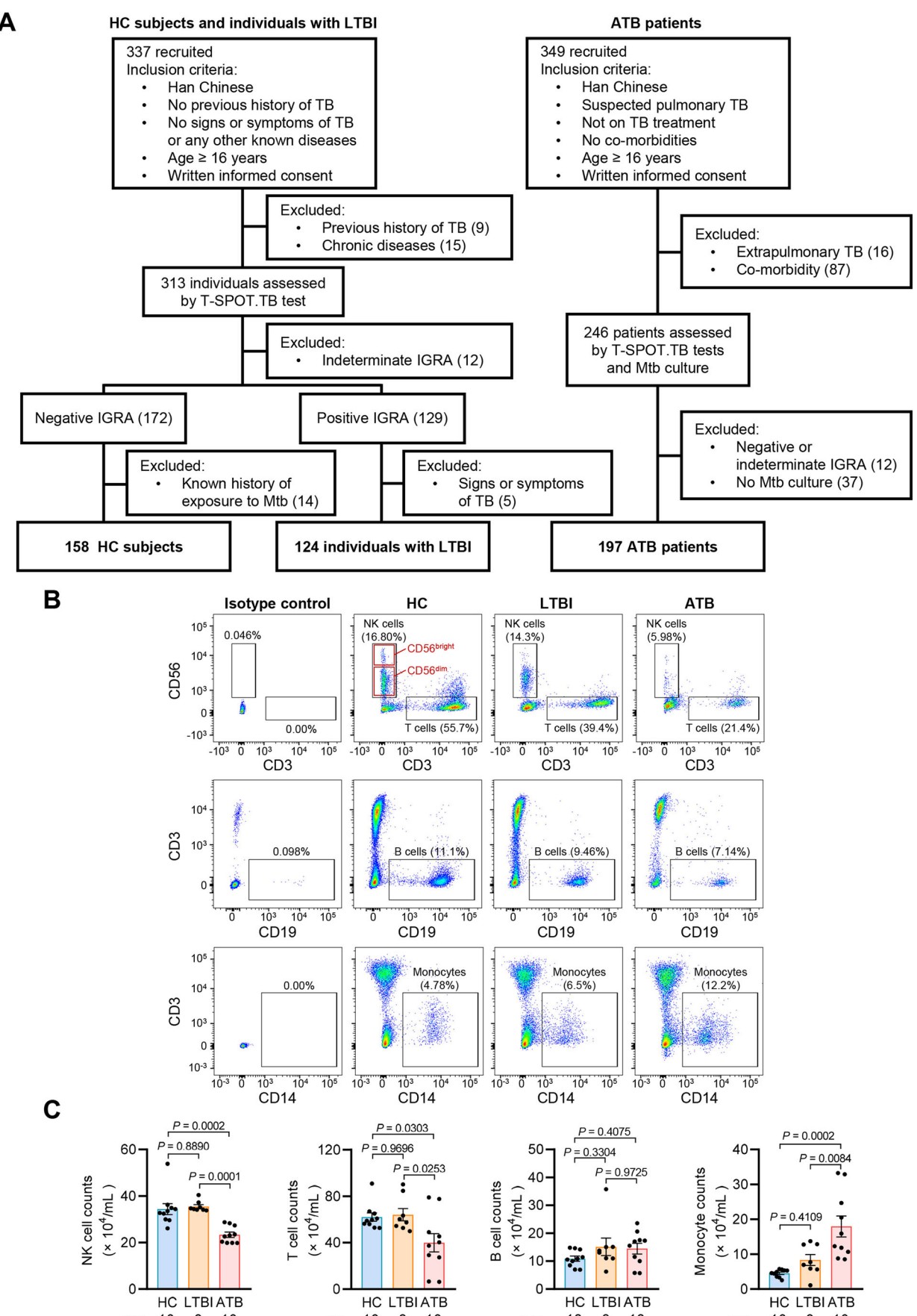

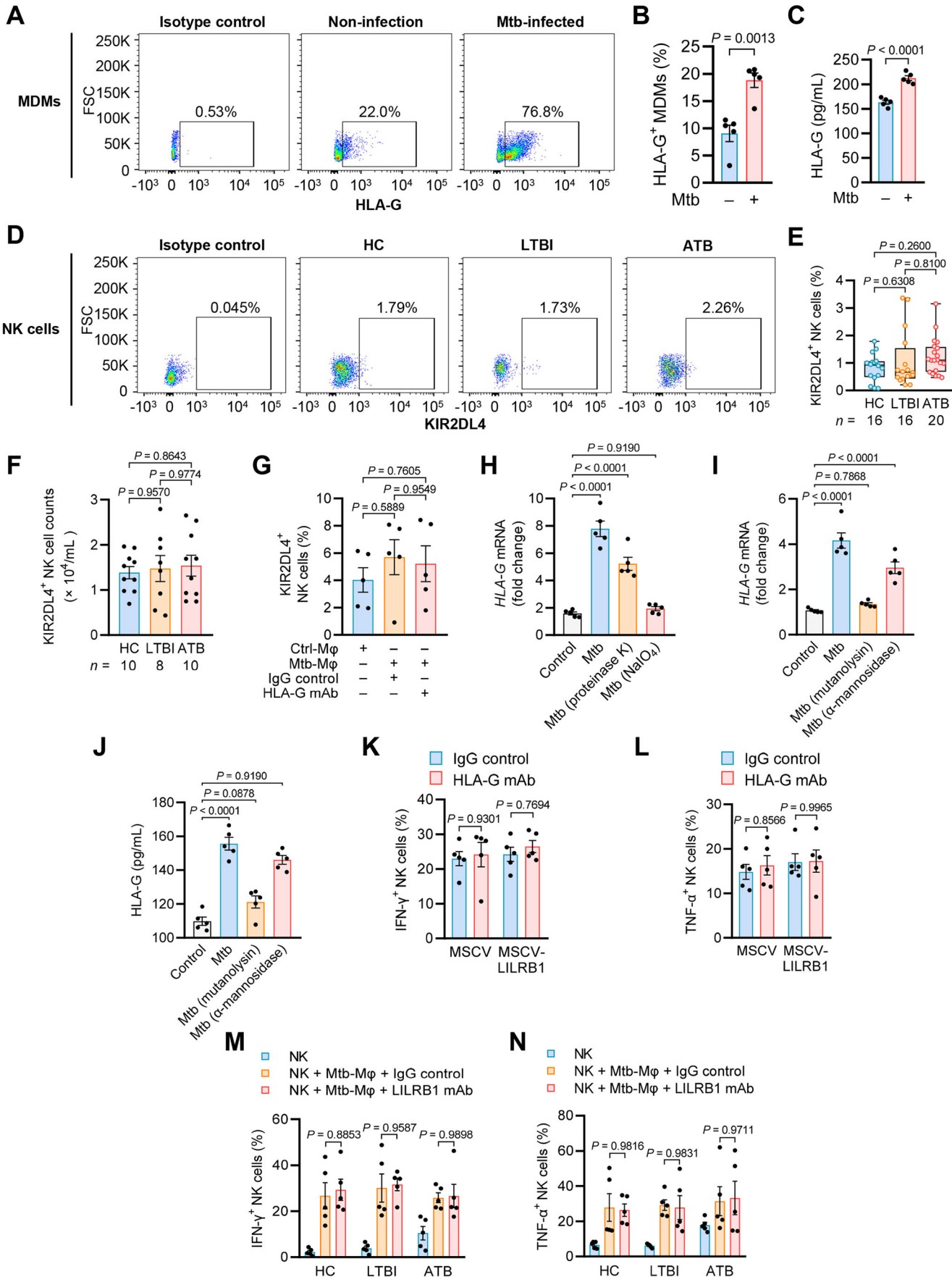

◀

**Figure EV2. Mtb infection increases HLA-G production in macrophages.**

(A) Representative results of FACS-based analysis for percentages of HLA-G$^+$ MDMs from HC group after infection with or without Mtb for 3 days. (B, C) Percentages of HLA-G$^+$ cells (B) and supernatant levels of soluble HLA-G (C) in MDMs treated as in (A). (D) Representative results of FACS-based analysis for percentages of KIR2DL4$^+$ cells within total NK cells from the peripheral blood of individuals in HC, LTBI, or ATB groups. (E) Percentages of KIR2DL4$^+$ cells within total NK cells in the peripheral blood of individuals ($n$) from HC, LTBI, or ATB groups. Box-whisker plot indicates the interquartile range (box), the median value (line within the box), and the maximum and minimum value (whiskers). (F) The number of KIR2DL4$^+$ NK cells in the peripheral blood of individuals ($n$) from HC, LTBI, or ATB groups. (G) Percentages of KIR2DL4$^+$ cells within NK cells. NK cells were co-cultured with Mtb-Mφ or control MDMs (Ctrl-Mφ) for 3 days with treatment of anti-HLA-G (87G) mAb or IgG control. (H, I) Quantitative PCR (qPCR) analysis of *HLA-G* mRNA in MDMs. MDMs were incubated with Mtb at a MOI of 5 for 3 days. Mtb was pretreated with or without proteinase K (to remove Mtb surface proteins), NaIO$_4$ (to remove Mtb surface carbohydrate residues), mutanolysin (to remove peptidoglycan), or α-mannosidase (to remove lipomannan and lipoarabinomannan). (J) Enzyme-linked immunosorbent assay (ELISA) of supernatant HLA-G produced by MDMs treated as in (I). (K, L) Percentages of IFN-γ$^+$ (K) or TNF-α$^+$ (L) cells within total NK cells. HC donor-derived NK cells with overexpression of MSCV or MSCV-LILRB1 were co-cultured with Mtb-Mφ for 24 h in the presence of anti-HLA-G (87G) antibody or IgG control. (M, N) Percentages of IFN-γ$^+$ (M) or TNF-α$^+$ (N) cells in MDM-NK cell co-cultures. NK cells from each indicated group were co-cultured with Mtb-Mφ for 24 h in the presence of anti-LILRB1 (GHI/75) mAb or IgG control. Data are mean ± SEM ($n = 5$ donors per group) in (G–N). Statistical significance was determined using two-tailed *t*-test (B, C), one-way ANOVA (E–J), and two-way ANOVA (K–N) with Tukey's post-hoc test. Results are representative of three independent experiments. Source data are available online for this figure.

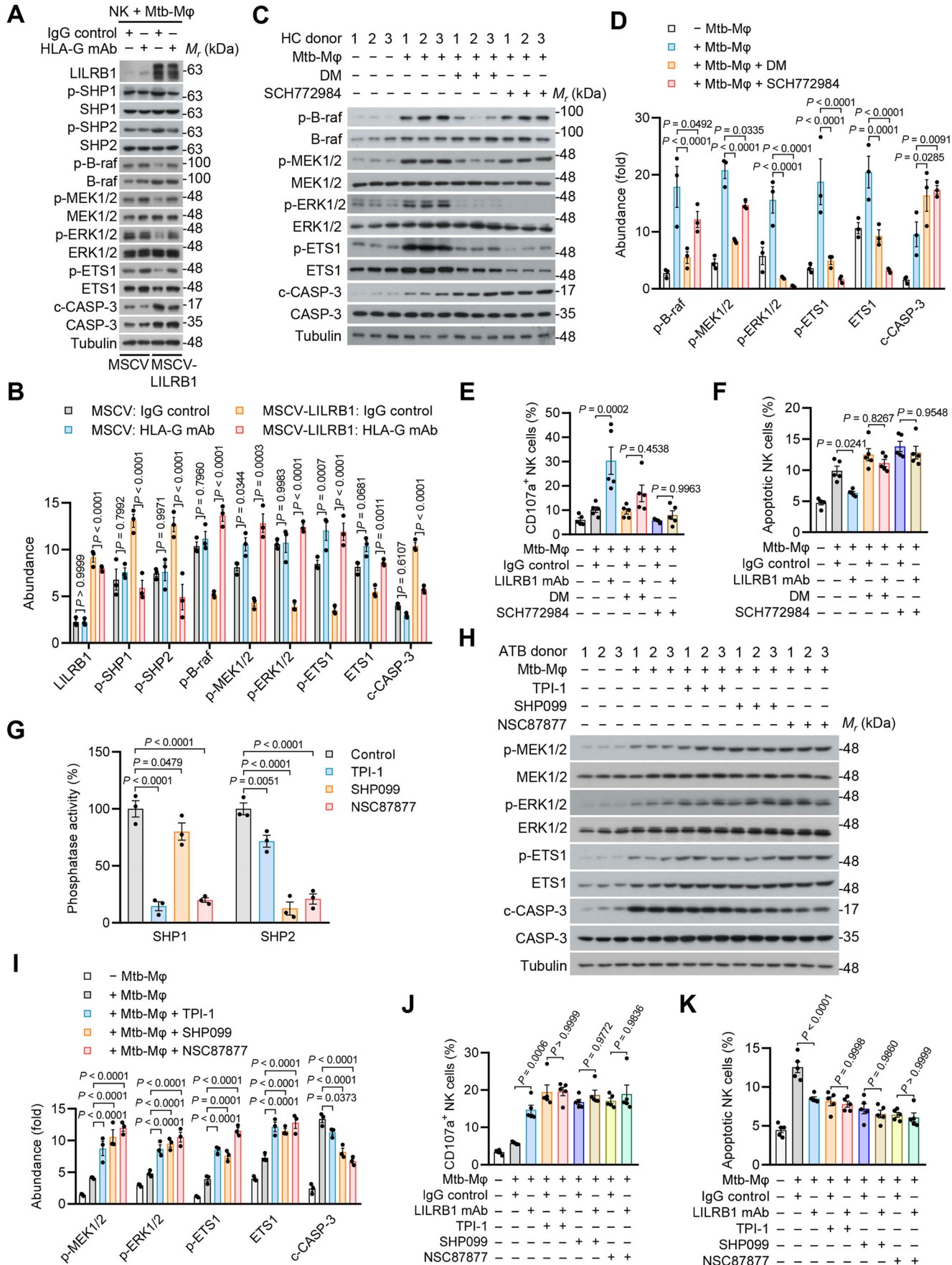

◀

**Figure EV3. Inhibition of SHP1/2 increases the activation of MEK1/2-ERK1/2-ETS1 signaling axis in NK cells.**

(A, B) Immunoblotting (A) and quantitation (B) for expression of indicated proteins in HC-donor-derived NK cells with overexpression of MSCV or MSCV-LILRB1. Cells were co-cultured with Mtb-infected MDMs for 24 h in the presence of anti-HLA-G (87G) antibody or IgG control. (C, D) Immunoblotting (C) and quantitation (D) for expression of indicated proteins in HC-donor-derived NK cells co-cultured with or without Mtb-infected MDMs (Mtb-Mφ) and treated with or without dabrafenib mesylate (DM) or SCH772984 for 24 h. (E, F) Percentages of CD107a⁺ (E) and apoptotic (F) cells within total NK cells from ATB patients. NK cells were co-cultured with Mtb-infected MDMs and treated with the indicated inhibitors in the presence of anti-LILRB1 (GHI/75) mAb or IgG control for 24 h. (G) Phosphatase activity of SHP1 and SHP2 in NK cells derived from ATB patients. Cells were treated with or without 0.5 μM TPI-1, SHP099, or NSC87877 for 24 h. (H, I) Immunoblotting (H) and quantitation (I) for expression of indicated proteins in ATB patient-derived NK cells. Cells were co-cultured with or without Mtb-Mφ in the presence or absence of indicated inhibitors (0.5 μM each) for 24 h. (J, K) Percentages of CD107a⁺ (J) and apoptotic (K) cells within total NK cells from ATB patients. NK cells were co-cultured with Mtb-infected MDMs and treated with the indicated inhibitors in the presence of anti-LILRB1 (GHI/75) mAb or IgG control for 24 h. Data are mean ± SEM [$n = 3$ donors per group in (B, D, G, I) and $n = 5$ donors per group in (E, F, J, K)]. Statistical significance was determined using two-way ANOVA (B, D, G, I) and one-way ANOVA (E, F, J, K) with Tukey's post-hoc test. Results are representative of three independent experiments. Source data are available online for this figure.

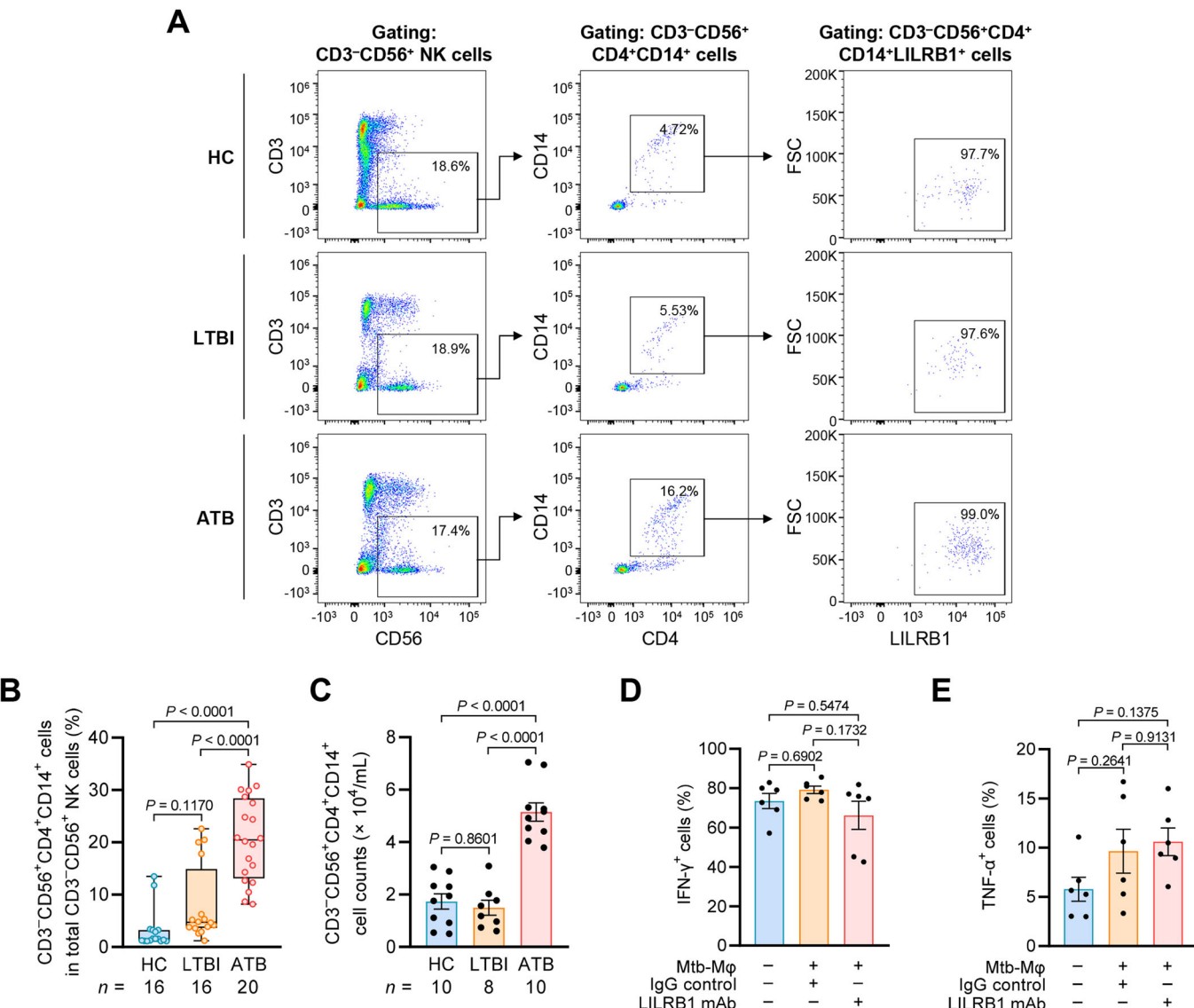

**Figure EV4. The frequency of CD3⁻CD56⁺CD4⁺CD14⁺ cell subset is increased in TB patients.**

(A) Representative results of FACS-based analysis for percentages of CD3⁻CD56⁺CD4⁺CD14⁺ cells within total CD3⁻CD56⁺ NK cells from the peripheral blood of individuals in HC, LTBI, or ATB groups. (B) Percentages of CD3⁻CD56⁺CD4⁺CD14⁺ cells within total CD3⁻CD56⁺ NK cells in the peripheral blood of individuals (n) from HC, LTBI, or ATB groups. Box-whisker plot indicates the interquartile range (box), the median value (line within the box), and the maximum and minimum value (whiskers). (C) The number of CD3⁻CD56⁺CD4⁺CD14⁺ cells in the peripheral blood of individuals (n) from HC, LTBI, or ATB groups. (D, E) Percentages of IFN-γ⁺ (D) and TNF-α⁺ (E) cells in total CD3⁻CD56⁺CD4⁺CD14⁺ cells. CD3⁻CD56⁺CD4⁺CD14⁺ cells were co-cultured with or without Mtb-Mφ at a ratio of 3:1 for 24 h in the presence of anti-LILRB1 (GHI/75) mAb or IgG control. Data are mean ± SEM (n = 6 donors per group) in (D, E). Statistical significance was determined using one-way ANOVA with Tukey's post-hoc test for (B–E). Results are representative of three independent experiments. Source data are available online for this figure.

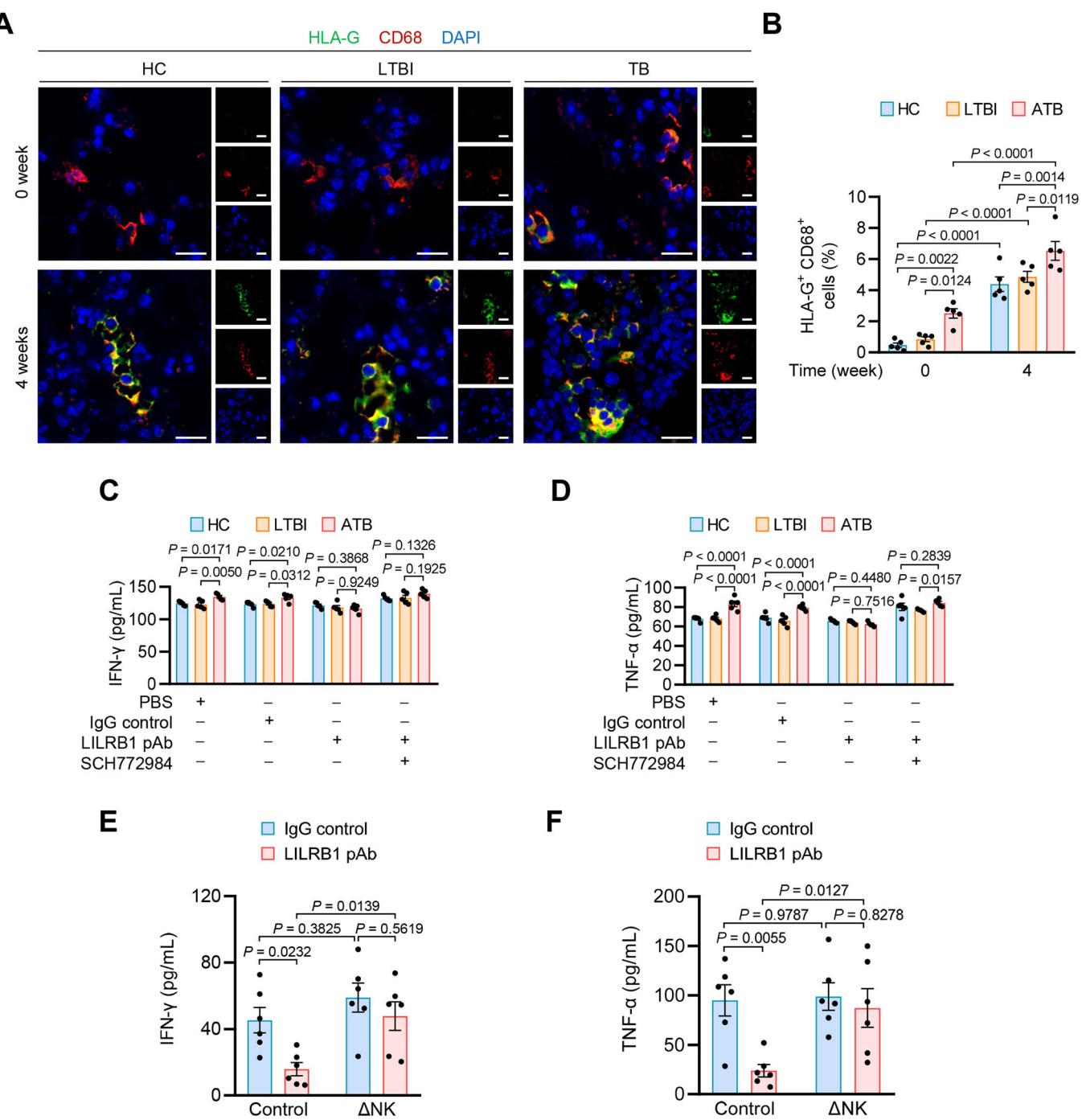

**Figure EV5. Mtb infection increases macrophage expression of HLA-G in the lung of immuno-humanized mice.**

(A) Representative images for cells stained with antibodies against HLA-G (green) and CD68 (red) in the lungs of NCG mice after infection with Mtb for 0–4 weeks. Nuclei were stained with DAPI (blue). Scale bars, 20 μm. (B) Quantitation of HLA-G expression in CD68$^+$ cells. (C–F) Quantitation of IFN-γ (C, E) and TNF-α (D, F) in the lungs of NCG mice. For (A–D), mice ($n = 5$ per group) were transplanted with PBMCs from individuals in HC, LTBI, or ATB groups. Two weeks later, mice were infected with Mtb by aerosol (~100 CFUs) for 4 weeks with treatment of PBS (as control), IgG control, or anti-LILRB1 blocking pAb every 3 days, together with the treatment of corn oil (as control) or SCH772984 every 2 days (see Fig. 7A). For (E, F), control PBMCs or NK cell-depleted (ΔNK) PBMCs from ATB patients were transplanted to NCG mice ($n = 6$ per group). Two weeks later, mice were infected with Mtb by aerosol (~100 CFUs) for 4 weeks with treatment of IgG control or anti-LILRB1 blocking pAb every 3 days (see Fig. 8A). Data are mean ± SEM [$n = 5$ mice per group in (B–D) and $n = 6$ mice per group in (E, F)]. Statistical significance was determined using two-way ANOVA with Tukey's post-hoc test for (B–F). Results are representative of two independent experiments. Source data are available online for this figure.

