## [Peer Review File · EMBO Molecular Medicine]

LILRB1-HLA-G axis defines a checkpoint driving natural killer cell exhaustion in tuberculosis

Cui Hua Liu, Jing Wang, Qiyao Chai, Zehui Lei, Yiru Wang, Jiehua He, Pupu Ge, Zhe Lu, Lihua Qiang, Dongdong Zhao, Shanshan Yu, Changgen Qiu, Yanzhao Zhong, Bingxi Li, Lingqiang Zhang, Yu Pang, and George Gao

Corresponding authors: Cui Hua Liu (liucuihua@im.ac.cn), George Gao (gaof@im.ac.cn), Yu Pang (py@bjxkyy.cn)

Review Timeline:

Submission Date:	24th Jan 24
Editorial Decision:	15th Mar 24
Revision Received:	2nd Jun 24
Editorial Decision:	1st Jul 24
Revision Received:	3rd Jul 24
Accepted:	5th Jul 24

Editor: Zeljko Durdevic

Transaction Report:

15th Mar 2024

Dear Prof. Liu,

Thank you for the submission of your manuscript to EMBO Molecular Medicine, and please accept my apologies for the delay in getting back to you, which is due to the fact that one referee needed more time to complete his/her review. We have now received feedback from the three reviewers who agreed to evaluate your manuscript. As you will see from the reports pasted below, all three referees recognize potential interest of the study but also raise important and partially overlapping concerns that should be addressed in a major revision of the current manuscript. If you would like to discuss further the points raised by the referees, I am available to do so via email or video. Let me know if you are interested in this option.

We would welcome the submission of a revised version within three months for further consideration. Please let us know if you require longer to complete the revision.

I look forward to receiving your revised manuscript.

Yours sincerely,

Zeljko Durdevic

We require:

- 1) A .docx formatted version of the manuscript text (including legends for main figures, EV figures and tables). Please make sure that the changes are highlighted to be clearly visible.
- 2) Individual production quality figure files as .eps, .tif, .jpg (one file per figure). For guidance, download the 'Figure Guide PDF': (<https://www.embopress.org/page/journal/17574684/authorguide#figureformat>).
- 3) A .docx formatted letter INCLUDING the reviewers' reports and your detailed point-by-point responses to their comments. As part of the EMBO Press transparent editorial process, the point-by-point response is part of the Review Process File (RPF), which will be published alongside your paper.
- 4) A complete author checklist, which you can download from our author guidelines (<https://www.embopress.org/page/journal/17574684/authorguide#submissionofrevisions>). Please insert information in the checklist that is also reflected in the manuscript. The completed author checklist will also be part of the RPF.
- 5) Please note that all corresponding authors are required to supply an ORCID ID for their name upon submission of a revised manuscript.

6) It is mandatory to include a 'Data Availability' section after the Materials and Methods. Before submitting your revision, primary datasets produced in this study need to be deposited in an appropriate public database, and the accession numbers and database listed under 'Data Availability'. Please remember to provide a reviewer password if the datasets are not yet public (see <https://www.embopress.org/page/journal/17574684/authorguide#dataavailability>).

13) Author contributions: You will be asked to provide CRediT (Contributor Role Taxonomy) terms in the submission system. These replace a narrative author contribution section in the manuscript.

14) A Conflict of Interest statement should be provided in the main text.

Please also suggest a striking image or visual abstract to illustrate your article as a PNG file 550 px wide x 300-800 px high.

**** Reviewer's comments ****

Referee #1 (Comments on Novelty/Model System for Author):

This is a very large body of work with a lot of investigation which appears to have been done in a reproducible and robust way. The detailed identification of a cell subset in well characterized subjects is novel and the demonstration that these cells can be involved in immunity is of medical impact. The complexity makes it difficult to fully assess the models and this is probably the most important defect in the paper. I don't have any ethical concerns.

Referee #1 (Remarks for Author):

This is a large body of work with some important observations and potential for follow through to have impact. The issues that arise for this reviewer are two fold.

1. The article is written with a view to proving a point rather than investigating a problem. This can be addressed by removing or rewriting sections as detailed below. There is no evidence of cell exhaustion provided - this element should be removed and the actual markers and functions being measured should be clearly reported.
2. The figures, models and descriptions of the work are almost impenetrable. The authors might consider removing some data sets to the supplementary and altering the figures to ensure that the key data is visible and being much clearer about what was done and what was observed in the results section. There is overinterpretation in the results section that should be moved to the discussion, as detailed in the notes below

Line 18 and 38 usually drives in an unscientific phrase - not sure that this is an accepted fact

Line 45 - has it been unexplored? Bagcchi reference appropriate?

Maybe making a bit much of potential - it works as it is

Line 58 'obtained unsatisfactory efficacy'

Line 57 to 63 shoe horning' may or may not be the case - but it is good to investigate

Line 65 to 68 I don't think the fact that most people who are exposed to TB use innate immunity to control is shown by the reference .

Line 77 to 81 are the cells exhausted or do they express specific markers - over interpretation

Line 83 to 85 - they do not approach the work with an open mind they are looking for a specific outcome - not the scientific method.

Fig 1 - okay with cohort. need details on the gating for the populations - percentage change can mean changes in other populations please clarify

Line 108 what is meant by positive correlation? What is being correlated with what?

Line 112-3 and line 116-118 over interpretation why does this suggest compromised effector function. Maybe leave for discussion.

Fig 2 D to G - its confusing having the with and without Mtb macrophages on the same graph - can you separate?

Line 126 'showed lower percentage in expressing CD107a' what does this mean? Lower frequency of CD107a expressing NK cells?

Line 132 'showed much weaker effects on suppressing Mtb survival'? I think you mean the NK cells from ATB blood was less able to reduce bacterial number in co-culture.

Line 135 - what is the evidence for 'distinct exhaustion characteristics'?

Fig 3C - I can't work out what is what here.

Line 170 to 173 is an over interpretation and needs to go to discussion for supporting rationale

Line 179 to 181 overstates the case - needs to go into discussion

Line 182 - what is the evidence for exhaustion?

Line 231-236 be careful of making causal connections when only associations are seen

Line 246 'were enriched to be potentially involved in regulating 247 NK cell anti-Mtb responses,' make this statement clearer

Figs 6 D and F too tiny needs to be rationalized.

Fig 6 overall is very dense and its almost impossible to discern whether the statements in the results section are supported by the data.

Fig 8 D - where is the lung data for CFU? Also, you cannot put a 0 for a log₁₀ scale graph.

Line 381 deprived or derived? It's confusing.

So much work but too confusing as presented.

Referee #2 (Comments on Novelty/Model System for Author):

There are some details with the humanized mouse that need to be clarified to fully understand the impact of the results

Referee #2 (Remarks for Author):

In the manuscript by Wang et al the authors explore the presence of a potential checkpoint (Inhibitory receptor-ligand combination) in NK cells, as NK cell dysfunctional is associated with active tuberculosis. Employing multiple approaches, the authors explore how LILRB1- HLA-G interactions reduce NK cell degranulation, induce apoptosis, and diminish cytotoxic activity, consequently affecting Mycobacterium tuberculosis (Mtb) killing capacity. The authors show data that a unique subset of NK cell that express LILRB1 require HLA-G interaction with macrophages to be in the inhibitory state and this interaction is prominent in active TB patients. They also observe and confirm the interactions is mediated through the MAPK signaling and via tyrosine kinase phosphatase SHP1/2. They then try and utilize an humanized mouse model to try and find the mechanistic basis for LILRB1 in NK-cell function in TB.

Major concerns:

1. The authors explore and present mostly convincing evidence using clinical patient samples that this unique NK subset via LILRB1-HLA-G axis may be driving the diminished NK cell functionality. Most data showing the presence of the C5 subset of NK cells expressing LILRB1 and the interactions with HLAG axis is strong and convincing. I have some comments about these data that need to be addressed: it is unclear how and why their studies deviate from published studies in fig 1C (Chowdury et al, Esaulova et al) which have shown increased NK cell frequencies in LTBI when compared to HC? This is not seen in this study. In fig 4M in human TB lesions- how many individual samples were included for this analysis- this is unclear? Multiple tissue samples need to be assessed.

2. The overexpression/blockade of LILRB1 in vitro is exciting and convincing.

3. My major concerns arise from the mouse experiments which need additional validation to be compelling:

In figure 7, was 1 donor to one mouse performed or was it the same donor to all mice within the group? Also, the two experiments repeated need to be shown maybe in supplementary data after clarification of the above point. It is surprising that there is very little inflammation in the lungs in PBS treated Mtb-infected mice.

In figure 8, there is no difference between the NK depleted and the control cells transferred from ATB patients into mice, so the point that this is NK cell driven is not convincing. Same point as above- was it one donor to 1 mouse or 1 donor to all mice within the group?

In fig 7/8, without measuring Mtb CFU by plating the data is not validated by standard methods. Mtb CFU needs to be included.

Referee #3 (Remarks for Author):

In their manuscript, Wang and colleagues conducted an analysis of immune cell subsets in a Tuberculosis (TB) cohort. The cohort was divided into three groups: patients diagnosed with pulmonary TB before treatment initiation (ATB) (182), patients with latent tuberculosis (LTBI) positive for interferon-gamma release assay (IGRA) (114), and healthy subjects negative for IGRA (HC) (142).

As expected, ATB patients exhibited lower T-cell counts and higher monocyte counts. The percentage of NK cells was significantly reduced in ATB patients compared to LTBI patients. However, the observation of decreased NK cells in TB patients varied across previous reports by others, raising questions about the significance of subsequent steps.

The authors proceeded with a transcriptomic analysis of NK cells isolated from peripheral blood mononuclear cells (PBMCs). Differences were observed in classical pathways: TLR, NLR, IL-17, TNF, and NF- κ B signaling were upregulated in ATB, while

PI3K-AKT signaling was downregulated. Additionally, increased inflammatory response and apoptosis were found, alongside decreased cytotoxic activity in ATB compared to HC or LTBI cases.

To analyze possible cellular dysfunction, NK cells were incubated with macrophages, and cytokine production was assessed. More IFN-gamma, TNF-alpha, increased CD107a+, and apoptotic NK cells were recorded after macrophage co-culture. Subsequently, a mass spectrometry CyTOF study identified an LILRB1-expressing subset named cluster C5 specific to ATB patients. This C5-NK cell cluster expressed low levels of granzyme B (GZMB) and perforin 1 (PRF1).

Further, the impact of LILRB1 on NK cell effect on macrophages was evaluated by overexpressing LILRB1 in NK cells from healthy donors and co-incubating with macrophages. However, the resulting data from these co-cultures were challenging to interpret due to the absence of clear counts of living phagocytes. Additionally, bacterial CFU displayed after 24h would be more convincing with a kinetic analysis over several days.

Despite the impressive study, the manuscript's figures were difficult to interpret quickly, and some findings were hard to believe. Notably, questions arose regarding the specificity of the LILRB1-HLA-G axis for Mtb and whether it depends on Mtb antigens.

Regarding the figures:

Figure 1: The number of cells after 24H of infection and the nature of CFU differences require clarification.

Figure 2: Panel H lacks informativeness. The absence of PRF1 in Figure 2C and the coloration of GZMB could be addressed.

Figure 5: A dose response of HLA-G might enhance data credibility, and the figure appears crowded.

Figure 6: The western blot in panel C could benefit from quantification, and the study with chemical inhibitors lacks robustness.

Figure 7: Providing multiple cells for each condition could improve data representation.

Figure 9: The graphical abstract is commendable.

In summary, while the manuscript is well-written, clearer figure presentation and more convincing data interpretations are warranted.

Point-by-point responses to reviewers' comments

RE: Manuscript (# EMM-2024-19348) "**LILRB1-HLA-G axis defines a checkpoint driving natural killer cell exhaustion in tuberculosis**" by Wang *et al.*

Reviewers' comments:

Referee #1:

Comments on Novelty/Model System for Author:

This is a very large body of work with a lot of investigation which appears to have been done in a reproducible and robust way. The detailed identification of a cell subset in well characterized subjects is novel and the demonstration that these cells can be involved in immunity is of medical impact. The complexity makes it difficult to fully assess the models and this is probably the most important defect in the paper. I don't have any ethical concerns.

R: We thank the reviewer for the encouraging comments on our manuscript and for the constructive suggestions to improve this manuscript.

Remarks for Author:

This is a large body of work with some important observations and potential for follow through to have impact. The issues that arise for this reviewer are two fold.

1. The article is written with a view to proving a point rather than investigating a problem. This can be addressed by removing or rewriting sections as detailed below. There is no evidence of cell exhaustion provided - this element should be removed and the actual markers and functions being measured should be clearly reported.

R: We thank the reviewer for these suggestions. We have rewritten some paragraphs in the main text to avoid overinterpretation, and we have also rearranged the figures after

performing additional experiments to clearly report the phenotype and function of TB-associated NK cells (e.g., revised **Fig. 1D–G**, **Fig. 6A,B**, and **Appendix Figure S1**). Based on our observations, we suggest that TB-associated NK cells display an exhausted phenotype with compromised anti-Mtb capacity, as explained as follows:

Immune cell exhaustion is a dysfunctional state characterized by decreased effector functions that was originally described in T cells in the setting of chronic lymphocytic choriomeningitis virus (LCMV) infection (Gallimore, A. *et al.*, *J Exp Med*, 1998). Recently, some studies have demonstrated that exposure to tumors or chronic infections also drives NK cells into an exhausted status (Zhang, Q. *et al.*, *Nat Immunol*, 2018; Zhang, C. *et al.*, *Hepatology*, 2021). Despite not being fully understood, NK cell exhaustion has been characterized by altered transcriptional programs, downregulated expression of certain activating receptors (such as NKG2C, NKG2D, and NCRs), upregulated expression of inhibitory receptors (such as TIGIT and KIRs), and decreased effector functions (Bi, J. & Tian, Z., *Front Immunol*, 2017; Merino, A. M. *et al.*, *J Leukoc Biol*, 2020). In our study, we revealed that ATB patient-derived NK cells displayed altered transcriptional programs suggestive of altered signaling pathways associated with NK cell effector functions (revised **Fig. 1B,C** and **Appendix Figure S1**). By performing CyTOF analysis, we further identified a TB-associated NK cell subset (i.e., C5 subset), whose frequency was significantly increased in ATB patients. Of note, this C5 NK cell subset was featured with high expression of inhibitory receptor LILRB1 but rare expression of activating markers such as NKG2C, NKG2D, and NCRs including NKp30, NKp44, and NKp46 (revised **Fig. 2C**). Further experimental evidence also indicated that this TB-associated NK cell subset exhibited decreased cytotoxic activity, increased apoptosis, and compromised anti-Mtb capacity (revised **Fig. 6**). Taken together, these evidences indicate an exhausted phenotype and function of TB-associated NK cells in the context of Mtb infection.

It should be mentioned that exhausted NK cells derived from cancer patients and those with chronic infectious diseases (such as TB) exhibit certain different characteristics. For instance, we demonstrated that NK cells in ATB patients were characterized by high expression of inhibitory marker LILRB1, rather than TIGIT or KIR2DL4, which are upregulated on NK cells to dampen their effector function in tumors (revised **Fig. 4A,B**, **Fig. EV2D–G**, and **Appendix Fig. S5A–C**) (Zhang, Q. *et al.*, *Nat Immunol*, 2018; Zheng, G. *et al.*, *Signal Transduct Target Ther*, 2021).

Furthermore, NK cells in ATB patients retain the expression of IFN- γ and TNF- α , distinct from that in cancer patients displaying general deficiency in both cytokine production and cytotoxic activity (Zhang, Q. *et al.*, *Nat Immunol*, 2018; Cong, J. J. *et al.*, *Cell Metab*, 2018). Actually, similar results were also observed during chronic infection of virus (such as MSCV and HIV), in which the exhausted T cells were deficient in a range of effector functions while maintaining the ability to produce IFN- γ (Appay, V. *et al.*, *J Exp Med*, 2000; Wherry, E. J. *et al.*, *J Virol*, 2003). Thus, our results support the notion that immunological exhaustion during chronic infection mainly refers to inefficient effector functions of immune cells that results in a host-pathogen stalemate, rather than complete absence of immune cell functions (Blank, C. U. *et al.*, *Nat Rev Immunol*, 2019; McLane, L. M. *et al.*, *Annu Rev Immunol*, 2019). During tumor development, host immune cells usually go through a long latency period in a non-inflammatory context (Willimsky, G. & Blankenstein, T., *Nature*, 2005; Blank, C. U. *et al.*, *Nat Rev Immunol*, 2019). In contrast, during chronic infections, host immune cells are likely to have differentiated into functional effector cells, in response to a pathogen-driven inflammatory context, before they become exhausted (Blank, C. U. *et al.*, *Nat Rev Immunol*, 2019; McLane, L. M. *et al.*, *Annu Rev Immunol*, 2019). In the case of Mtb infection, while individuals with LTBI are asymptomatic, ATB patients are characterized by progressive respiratory symptoms with persistent inflammation. Therefore, NK cells in ATB patients probably have gone through an initial effector phase before they become exhausted in response to continuous stimulation from Mtb, hence exhibiting attenuated anti-Mtb capacity while retaining the expression of some inflammatory cytokines. Thus, the disease-specific features of exhausted immune cells are related to the inflammatory environment in different disease settings, and the developmental trajectories of the exhausted immune cells could be heterogeneous (Blank, C. U. *et al.*, *Nat Rev Immunol*, 2019).

Accordingly, we have made a comprehensive discussion in the “Discussion” section (line 423–498).

2. The figures, models, and descriptions of the work are almost impenetrable. The authors might consider removing some data sets to the supplementary and altering the figures to ensure that the key data is visible and being much clearer about what was

done and what was observed in the results section. There is over interpretation in the results section that should be moved to the discussion, as detailed in the notes below.

R: We thank the reviewer for these helpful suggestions. We have rearranged the figures by moving some data to the supplementary files (including Expanded View Figures and Appendix Figures) to ensure that the key data is more visible in the main figures. Specifically, previous Figs. 1A, 1B, 1D, 5K, 5L, 5S, 5T, 6E–J, 7A–C, 7K, 7L, 9G, 9H have been moved to the revised Expanded View Figures, and previous Figs. 2C, 4B–E, 9B, 9K have been moved to the revised Appendix Figures. Accordingly, we have revised the main text to make it clearer. The followings are the point-by-point responses.

1) Line 18 and 38 usually drives in an unscientific phrase - not sure that this is an accepted fact.

R: We have rephrased the relevant sentences accordingly (**line 18–19** and **38–39**).

2) Line 45 - has it been unexplored? Bagcchi reference appropriate? Maybe making a bit much of potential - it works as it is.

R: We thank the reviewer for pointing this out. Immunotherapy based on checkpoint blockade has initially been explored for treating chronic infectious diseases, mainly chronic viral infections (Kubli, SP. et al., Nat Rev Drug Discov, 2021), while its potential for treating chronic bacterial infections remain largely unexplored. We have revised the sentence (**line 43–47**) to avoid overinterpretation. Moreover, we have replaced the reference of “Bagcchi, 2023” by “World Health Organization, 2023” (**line 50**) to clarify the serious epidemic of tuberculosis according to the Global Tuberculosis Report published by World Health Organization.

3) Line 58 'obtained unsatisfactory efficacy'.

R: We have rephrased the sentence (**line 58–62**) to make it more precise and clearer.

4) Line 57 to 63 'shoe horning' may or may not be the case - but it is good to investigate.

R: We have rephrased the sentence (**line 58–64**) to make it more precise and clearer.

5) Line 65 to 68 I don't think the fact that most people who are exposed to TB use innate immunity to control is shown by the reference.

R: We thank the reviewer for pointing this out, and we have removed the inappropriate description and rephrased the sentence (**line 64–67**).

6) Line 77 to 81 are the cells exhausted or do they express specific markers - over interpretation.

R: We thank the reviewer for pointing this out, and we have rephrased the sentence accordingly (**line 77–81**): ATB patients display decreased frequency of NK cells as compared to asymptomatic individuals with LTBI (Roy Chowdhury, R. *et al.*, *Nature*, 2018). Furthermore, NK cells from ATB patients are featured with decreased expression of activating receptors (e.g., NKp30 and NKp46) (Bozzano, F. *et al.*, *Int Immunol*, 2009), indicating the functional change of NK cells during Mtb infection.

7) Line 83 to 85 - they do not approach the work with an open mind they are looking for a specific outcome - not the scientific method.

R: We thank the reviewer for helping us improve our manuscript. We have revised the relevant description to make unbiased statements regarding the main purpose and methodology of our study (**lines 77–88**). As described above (in response to point 6), previous studies have indicated an altered frequency of circulating NK cells and altered expression of immune receptors on NK cells in individuals with Mtb infection. However, the phenotypic and functional properties of NK cells in TB patients remain largely unclear. We thus in this study combined transcriptional profiling with CyTOF to systematically analyze the functional characteristics and immune receptor repertoire of NK cells in individuals at different status of Mtb infection, with an aim to identify the critical NK checkpoint molecules and their signaling network that may serve as new targets for improving host anti-Mtb immunity.

8) Fig 1 - okay with cohort. need details on the gating for the populations - percentage change can mean changes in other populations please clarify.

R: As suggested by the reviewer, we have provided details on the gating for each population in the figure legend of revised **Fig. EV1B** (e.g., previous Fig. 1B). In addition, we have also determined the absolute counts of these peripheral blood cell populations to confirm their variations among tested groups (revised **Fig. EV1C**) by FACS analysis using BD Trucount Absolute Counting Tubes (please see the “**Methods**” section for details; **line 699–711**).

9) Line 108 what is meant by positive correlation? What is being correlated with what?

R: We have rephrased the sentence accordingly to make it easier to understand (**lines 109–115**). This sentence means that genes involved in multiple inflammatory pathways [such as Toll-like receptor (TLR), NOD-like receptor (NLR), and TB-related signaling pathways] and apoptosis were upregulated, while genes related to phosphoinositide-3 kinase (PI3K)-AKT and RAP1 signaling pathways were downregulated, in NK cells from ATB group as compared to that from HC or LTBI group.

10) Line 112-3 and line 116-118 over interpretation why does this suggest compromised effector function. Maybe leave for discussion.

R: We have removed or rephrased these descriptions (**line 109–115** and **118–120**) and added a brief discussion accordingly in the “**Discussion**” section (**line 444–449**).

11) Fig 2 D to G - its confusing having the with and without Mtb macrophages on the same graph - can you separate?

R: As suggested by the reviewer, we have rearranged these graphs (revised **Fig. 1D–G**) to make them easier to understand.

12) Line 126 'showed lower percentage in expressing CD107a' what does this mean? Lower frequency of CD107a expressing NK cells?

R: We have revised the sentence accordingly (**line 128–129**). As indicated by the reviewer, this sentence means that ATB group exhibited lower frequency of NK cells expressing CD107a in the MDM-NK cell co-culture system, as compared to HC or LTBI group.

13) Line 132 'showed much weaker effects on suppressing Mtb survival'? I think you mean the NK cells from ATB blood was less able to reduce bacterial number in co-culture.

R: We have revised the sentence accordingly (**line 134–137**). As mentioned by the reviewer, this sentence means that NK cells from ATB group were less able to reduce bacterial survival in the MDM-NK cell co-culture system, as compared to that from HC or LTBI group.

14) Line 135 - what is the evidence for 'distinct exhaustion characteristics'?

R: As mentioned above (in response to point 1), under the settings of tumors and chronic infections, NK cells exhibit an exhausted status characterized by altered transcriptional programs, downregulated expression of activating receptors, upregulated expression of inhibitory receptors, and decreased effector functions (Bi, J. & Tian, Z., *Front Immunol*, 2017; Merino, A. M. *et al.*, *J Leukoc Biol*, 2020). Previous Fig. 2 (revised **Fig. 1** and **Appendix Figure S1**) revealed that ATB patient-derived NK cells displayed altered transcriptional programs suggestive of altered signaling pathways associated with NK cell effector functions, and exhibited compromised anti-Mtb capacity with increased apoptosis and decreased cytotoxic activity, which collectively indicate an exhausted phenotype of NK cells. Accordingly, we have rephrased the relevant sentence to better explain this point (**line 137–140**).

15) Fig 3C - I can't work out what is what here.

R: We have provided a more detailed description in the legend of previous Fig. 3C (revised **Fig. 2C**) to make it easier to understand.

16) Line 170 to 173 is an over interpretation and needs to go to discussion for supporting rationale.

R: We have removed this overinterpreted description. Furthermore, we have added a brief discussion on the effects of LILRB1 upregulation on NK cell function in the “**Discussion**” section (line 509–517).

17) Line 179 to 181 overstates the case - needs to go into discussion.

R: We have moved this interpretation to the “**Discussion**” section (line 509–517).

18) Line 182 - what is the evidence for exhaustion?

R: As mentioned above (in response to point 1), under the settings of tumors and chronic infections, NK cells exhibit an exhausted status characterized by altered transcriptional programs, downregulated expression of certain activating receptors and upregulated expression of inhibitory receptors, and decreased effector functions (Bi, J. & Tian, Z., *Front Immunol*, 2017; Merino, A. M. *et al.*, *J Leukoc Biol*, 2020). Our data showed that ATB patients exhibited increased frequency of inhibitory receptor LILRB1-positive NK cell subset (i.e., C5 subset), which was also featured with rare expression of activating markers such as NKG2C, NKG2D, NKp30, NKp44, and NKp46 (revised **Fig. 2**). Then, we further demonstrated that the increased expression of LILRB1 led to decreased expression of CD107a, reduced production of GZMB and PRF1, decreased cytotoxic activity, increased apoptosis, and compromised anti-Mtb capacity of NK cells (revised **Fig. 3**). Therefore, these results collectively suggest that during Mtb infection, LILRB1 is a potential key checkpoint receptor whose upregulation is responsible for the exhaustion phenotype of NK cells including decreased expression of CD107a, reduced production of GZMB and PRF1, decreased cytotoxic activity, increased apoptosis, and attenuated anti-Mtb capacity. We have rephrased the relevant sentence to better explain this point (line 182–186).

19) Line 231-236 be careful of making causal connections when only associations are seen.

R: We have rephrased the sentence for a more precise description (**line 243–248**).

20) Line 246 'were enriched to be potentially involved in regulating NK cell anti-Mtb responses,' make this statement clearer.

R: We have rephrased the sentence for a clearer statement (**lines 257–260**).

21) Figs 6 D and F too tiny needs to be rationalized.

R: We have enlarged the graphs and displayed them in a more rational way (revised **Fig. 5D** and **Fig. EV3B**).

22) Fig 6 overall is very dense and its almost impossible to discern whether the statements in the results section are supported by the data.

R: We have rearranged the panels to make them present in a more rational way (revised **Fig. 5** and **Fig. EV3**).

23) Fig 8 D - where is the lung data for CFU? Also, you cannot put a 0 for a log10 scale graph.

R: In previous Fig. 8, the Mtb burden in mouse lungs was indicated by the results of acid-fast staining. We have now added the results of lung CFUs (revised **Fig. 7E**). The result showed that mice transferred with ATB patient-derived PBMCs displayed higher Mtb loads in the lungs as compared to that in mice transferred with PBMCs from HC or LTBI group, while LILRB1 blockade markedly reduced bacterial counts in the lungs of mice transferred with ATB patient-derived PBMCs, reaching levels similar to that in the other two groups of mice (revised **Fig. 7E**). Furthermore, inhibition of ERK1/2 by SCH772984 abolished the effect from LILRB1 blockade (revised **Fig. 7E**), supporting an indispensable role of MAPK signaling in LILRB1-blockade-mediated restoration of host anti-Mtb immunity.

Furthermore, we have corrected the CFU graphs to avoid putting zero on a logarithmic axis (revised **Figs. 7E,F** and **8F,G**).

24) Line 381 deprived or derived? It's confusing.

R: Here we mean to say that we established immuno-humanized mice transferred with ATB patient-derived PBMCs in which NK cells were depleted. We have revised the sentence accordingly (**line 406–407**).

25) So much work but too confusing as presented.

R: We thank the reviewer for the above constructive suggestions. We have rearranged the figures carefully to make them present in a more rational way, and rewritten some paragraphs in the main text accordingly to avoid overinterpretation (as in response to the above points).

Referee #2:

Comments on Novelty/Model System for Author:

There are some details with the humanized mouse that need to be clarified to fully understand the impact of the results.

R: We thank the reviewer for providing the constructive suggestions to improve our manuscript.

Remarks for Author:

In the manuscript by Wang et al the authors explore the presence of a potential checkpoint (Inhibitory receptor-ligand combination) in NK cells, as NK cell dysfunctional is associated with active tuberculosis. Employing multiple approaches, the authors explore how LILRB1- HLA-G interactions reduce NK cell degranulation, induce apoptosis, and diminish cytotoxic activity, consequently affecting *Mycobacterium tuberculosis* (Mtb) killing capacity. The authors show data that a unique subset of NK cell that express LILRB1 require HLA-G interaction with macrophages to be in the inhibitory state and this interaction is prominent in active TB patients. They

also observe and confirm the interactions is mediated through the MAPK signaling and via tyrosine kinase phosphatase SHP1/2. They then try and utilize a humanized mouse model to try and find the mechanistic basis for LILRB1 in NK-cell function in TB.

R: We thank the reviewer for this precise interpretation of our findings.

Major concerns:

1. The authors explore and present mostly convincing evidence using clinical patient samples that this unique NK subset via LILRB1-HLA-G axis may be driving the diminished NK cell functionality. Most data showing the presence of the C5 subset of NK cells expressing LILRB1 and the interactions with HLAG axis is strong and convincing. I have some comments about these data that need to be addressed: it is unclear how and why their studies deviate from published studies in fig 1C (Chowdhury et al, Esaulova et al) which have shown increased NK cell frequencies in LTBI when compared to HC? This is not seen in this study. In fig 4M in human TB lesions- how many individual samples were included for this analysis- this is unclear? Multiple tissue samples need to be assessed.

R: We thank the reviewer for raising these important questions. Regarding NK cell frequencies in individuals at different status of Mtb infection, our study showed higher NK cell frequencies in LTBI as compared to that in ATB, a result which is consistent with studies from Chowdhury et al and Esaulova et al (Roy Chowdhury, R. *et al.*, *Nature*, 2018; Esaulova, E. *et al.*, *Cell Host Microbe*, 2021). However, when compared NK cell frequencies between LTBI and HC, our study showed no significant difference between these two groups, while Chowdhury's study showed increased NK cell frequencies in LTBI than that in HC, and Esaulova's study did not compare the NK cell frequencies between LTBI and HC in humans. The different observation between studies from us and Chowdhury et al probably due to the difference in regions and ages of study participants between the two studies. Specifically, our study recruited participants aged ≥ 16 years old (with an average of ~ 40 years old) from China, while Chowdhury's study recruited much younger participants aged 13–18 years old (i.e., adolescent participants) from South Africa. We have made a brief discussion accordingly (**line 440–444**).

Moreover, for previous Fig. 4M–O (revised **Fig. 3I–K**), we collected lung sections

from 5 ATB patients to examine the expression of LILRB1 in NK cells. Previous Fig. 4M (revised **Fig. 3I**) showed representative images of LILRB1 expression in lung resident or circulating NK cells from 1 ATB patient, and those of the other 4 ATB patients are shown below (these replicate data were provided as source data that have been deposited at Figshare, according to the EMBO Press guidelines). Moreover, previous Fig. 4N,O (revised **Fig. 3J,K**) showed the quantitative results of the total 5 ATB patients. We have provided the relevant information in the revised figure legends.

Histological and immunofluorescence analysis of human lung sections containing non-necrotic or necrotic TB granulomas or healthy control tissues adjacent to granulomatous lesions. For each panel, the left shows representative images (scale bars, 200 μm) for hematoxylin and eosin (H&E) staining, and the right shows representative images (scale bars, 10 μm) for cells stained with antibodies against LILRB1 (blue), CD56 (green) and CD69 or CD103 (red). Nuclei were stained with DAPI (gray).

2. The overexpression/blockade of LILRB1 in vitro is exciting and convincing.

R: We thank the reviewer for these comments.

3. My major concerns arise from the mouse experiments which need additional validation to be compelling:

1) In figure 7, was 1 donor to one mouse performed or was it the same donor to all mice within the group? Also, the two experiments repeated need to be shown maybe in supplementary data after clarification of the above point. It is surprising that there is very little inflammation in the lungs in PBS treated Mtb-infected mice.

R: For mouse experiments in the revised **Fig. 7**, each group (HC, LTBI, and ATB) included 5 donors, and each donor's PBMCs were transplanted to four mice, which were subjected to four different treatments as indicated (PBS, IgG control, anti-LILRB1 blocking pAb, and anti-LILRB1 blocking pAb along with SCH772984, respectively). We have provided relevant information in the legend.

Moreover, the replicate data of revised **Fig. 7** are showed as below (these replicate data have been provided as source data according to the EMBO Press guidelines), which are consistent with results shown in revised **Fig. 7**. It should be pointed out that PBS-treated Mtb-infected mice showed much smaller granuloma-like lesions (black arrows) and much lower inflammation in HC and LTBI groups compared to that in ATB group (revised **Fig. 7B** and panel A in the figure below). This is probably because mice transferred with PBMCs from HC or LTBI group have stronger ability to clear Mtb and thus exhibit lower pathogenic inflammation at 4 weeks after Mtb infection, as compared to that transferred with PBMCs from ATB group. Furthermore, LILRB1 blockade by anti-LILRB1 pAb, but not IgG control, markedly reduced inflammatory infiltration, bacterial burden, and TUNEL⁺ CD56⁺ cells with increased production of GZMB and PRF1 in the lungs of mice transferred with ATB patient-derived PBMCs, reaching levels similar to mice transferred with PBMCs from HC or LTBI individuals (revised **Figs. 7B–F** and panel B–E in the figure below). Thus, these data suggest that LILRB1 blockade restores host anti-Mtb immunity in immuno-humanized mice transplanted with PBMCs from ATB patients to reduce bacteria survival and mitigate host lung inflammation.

Blocking LILRB1 restores host anti-Mtb immunity in immuno-humanized mice. (A) Representative images (upper) and quantitation (lower) for H&E staining, acid-fast staining, and immunostaining of lung sections. Scale bars, 1 mm, 10 μ m, and 10 μ m, respectively. For immunostaining, NK cells were stained with anti-human CD56 antibody (red), apoptotic cells were stained with TUNEL (green), and nuclei were stained with DAPI (blue). Black arrows indicate granuloma-like inflammatory lesions. Black arrowheads indicate Mtb. White arrowheads indicate TUNEL-positive NK cells. (B and C) Enzyme-linked immunosorbent assay (ELISA) of GZMB (B) or PRF1 (C) in the lungs. (D and E) Bacterial CFUs in the lungs (D) or spleens (E). For (A–E), each group (HC, LTBI, or ATB) includes 5 donors, and each donor's PBMCs were transplanted to four mice, which were subjected to four different treatments as indicated (PBS, IgG control, anti-LILRB1 blocking pAb, and anti-LILRB1 blocking pAb along with SCH772984, respectively). Data are mean \pm SEM ($n = 5$ mice per treatment group). $P > 0.05$, not significant (ns); $*P < 0.05$; $**P < 0.01$; $***P < 0.001$; $****P < 0.0001$ (two-way ANOVA with Tukey's post-hoc test).

2) In figure 8, there is no difference between the NK depleted and the control cells transferred from ATB patients into mice, so the point that this is NK cell driven is not convincing. Same point as above- was it one donor to 1 mouse or 1 donor to all mice within the group?

R: We thank the reviewer for raising these questions. It should be pointed out that in revised **Fig. 8**, immuno-humanized NCG mice were established by transplanting PBMCs from ATB patients, whose NK cells were compromised in effector functions against Mtb, as supported by our in vitro evidence. Thus, when compared to Δ NK PBMC-transferred group, the control PBMC-transferred group did not show significant advantage in controlling Mtb infection, unless they were treated by LILRB1 pAb, which blocks HLA-G and LILRB1 interaction to restore NK cell anti-Mtb effector functions (revised **Fig. 8**). Thus, these results support our notion that NK cell-dependent anti-Mtb immunity is critical for host defense against Mtb infection, but it is compromised by the LILRB1-HLA-G signaling in ATB patients.

For mouse experiments in revised **Fig. 8**, a total of 6 ATB donors were included for collecting PBMCs, and each donor's PBMCs were divided into two treatment groups, of which one was subjected to NK cell depletion (Δ NK PBMCs) and one was not (control PBMCs). Then, the control PBMCs or Δ NK PBMCs from each donor were transplanted to two mice, of which one was treated with anti-LILRB1 blocking pAb and one was treated with IgG control. We have provided this information in the legend.

3) In fig 7/8, without measuring Mtb CFU by plating the data is not validated by standard methods. Mtb CFU needs to be included.

R: We assume that the reviewer refers to previous Fig. 8/9 (rather than previous Fig. 7/8). For previous Fig. 8, only spleen CFUs were shown, and the Mtb burden in mouse lungs was indicated by the results of acid-fast staining. We have now rearranged this figure and added the results of lung CFUs (revised **Fig. 7E**). This result showed that mice transferred with ATB patient-derived PBMCs displayed higher Mtb loads in the lungs as compared to that in the mice transferred with PBMCs from HC or LTBI group, while LILRB1 blockade markedly reduced bacterial counts in the lungs of mice

transferred with ATB patient-derived PBMCs, reaching levels similar to that in the other two groups of mice (revised **Fig. 7E**). Furthermore, inhibition of ERK1/2 by SCH772984 abolished the effect of LILRB1 blockade (revised **Fig. 7E**), supporting an indispensable role of MAPK signaling in LILRB1-blockade-mediated restoration of host anti-Mtb immunity.

For previous Fig. 9, both lung and spleen CFUs were shown (in previous **Fig. 9I,J**). Probably, the reviewer missed these data due to the overcrowded graphs in this figure. We have now rearranged the graphs and displayed them in a more rational way (revised **Fig. 8F,G**).

Referee #3:

Remarks for Author:

In their manuscript, Wang and colleagues conducted an analysis of immune cell subsets in a Tuberculosis (TB) cohort. The cohort was divided into three groups: patients diagnosed with pulmonary TB before treatment initiation (ATB) (182), patients with latent tuberculosis (LTBI) positive for interferon-gamma release assay (IGRA) (114), and healthy subjects negative for IGRA (HC) (142).

As expected, ATB patients exhibited lower T-cell counts and higher monocyte counts. The percentage of NK cells was significantly reduced in ATB patients compared to LTBI patients. However, the observation of decreased NK cells in TB patients varied across previous reports by others, raising questions about the significance of subsequent steps.

The authors proceeded with a transcriptomic analysis of NK cells isolated from peripheral blood mononuclear cells (PBMCs). Differences were observed in classical pathways: TLR, NLR, IL-17, TNF, and NF- κ B signaling were upregulated in ATB, while PI3K-AKT signaling was downregulated. Additionally, increased inflammatory response and apoptosis were found, alongside decreased cytotoxic activity in ATB compared to HC or LTBI cases.

To analyze possible cellular dysfunction, NK cells were incubated with macrophages, and cytokine production was assessed. More IFN-gamma, TNF-alpha, increased

CD107a⁺, and apoptotic NK cells were recorded after macrophage co-culture. Subsequently, a mass spectrometry CyTOF study identified an LILRB1-expressing subset named cluster C5 specific to ATB patients. This C5-NK cell cluster expressed low levels of granzyme B (GZMB) and perforin 1 (PRF1).

Further, the impact of LILRB1 on NK cell effect on macrophages was evaluated by overexpressing LILRB1 in NK cells from healthy donors and co-incubating with macrophages. However, the resulting data from these co-cultures were challenging to interpret due to the absence of clear counts of living phagocytes. Additionally, bacterial CFU displayed after 24 h would be more convincing with a kinetic analysis over several days.

Despite the impressive study, the manuscript's figures were difficult to interpret quickly, and some findings were hard to believe. Notably, questions arose regarding the specificity of the LILRB1-HLA-G axis for Mtb and whether it depends on Mtb antigens.

R: We thank the reviewer for the insightful comments and the valuable suggestions to improve our manuscript. We have now rearranged the main figures by moving some data to the supplementary files (including Expanded View Figures and Appendix Figures) to make our key data easier to interpret. Specifically, the previous Figs. 1A, 1B, 1D, 5K, 5L, 5S, 5T, 6E–J, 7A–C, 7K, 7L, 9G, 9H were moved to the revised Expanded View Figures, and the previous Figs. 2C, 4B–E, 9B, 9K were moved to the revised Appendix Figures.

As suggested by the reviewer, we have also performed additional experiments for kinetic analysis of bacterial CFUs and macrophage viability, and our new data consistently indicate that NK cells from ATB group have weaker activity on cytolysis of Mtb-infected MDMs with attenuated ability to reduce bacterial survival in the co-culture system, as compared to that from HC or LTBI group (revised **Appendix Fig. S2E,F**).

To address the reviewer's question on whether the LILRB1-HLA-G axis depends on Mtb antigens, we have performed additional experiments to identify that Mtb peptidoglycan is the critical determinant that increases HLA-G expression in MDMs during Mtb infection (revised **Fig. EV2H–J; line 225–234**). These new data, together with our findings demonstrating that HLA-G upregulates and activates LILRB1 on NK

cells to impair their effector functions against Mtb, indicate that Mtb employs peptidoglycan to drive the LILRB1-HLA-G axis-dependent NK cell exhaustion in ATB patients.

Moreover, we agree with the reviewer's comment that the LILRB1-HLA-G axis is probably not specific for Mtb infection, since human cytomegalovirus infection and tumors have been reported to modulate LILRB1-HLA-G axis-mediated disruption of NK cell effector functions (Chapman, TL. *et al.*, *Immunity*, 1999; Mandel, I. *et al.*, *J Immunother Cancer*, 2022) (line 569–573). Thus, it is more appropriate to consider the LILRB1⁺ NK cells as a “TB-associated” NK cell subset, rather than a “TB-specific” NK cell subset. We have revised the manuscript accordingly for a more precise description (lines 25, 142, 336, 459, 468, 532, and 534).

Regarding the figures:

1. Figure 1: The number of cells after 24 H of infection and the nature of CFU differences require clarification.

R: As suggested by the reviewer, we have repeated the CFU experiments and concurrently quantified the viable number of macrophages at several time points after Mtb infection (revised **Appendix Fig. S2E,F**). The results consistently showed that NK cells from ATB group were less able to reduce macrophage viability and Mtb survival in the MDM-NK cell co-culture system, as compared to that from HC or LTBI group. These additional data further confirm the attenuated capacity of ATB patient-derived NK cells to kill Mtb-infected MDMs and to control Mtb infection.

It should be pointed out that we collected total cells along with the culture media of MDM-NK cell co-cultures to determine Mtb CFUs (as described in the “**Method**” section; line 774–778). By this way, the surviving Mtb either within MDMs or released to the culture media due to cytolysis of MDMs by NK cells can be totally detected. Therefore, the nature of CFU differences among HC, LTBI, and ATB groups are mainly caused by different anti-Mtb capacity of NK cells (which kill Mtb directly by releasing perforin and granulysin) rather than living macrophage numbers.

2. Figure 2: Panel H lacks informativeness. The absence of PRF1 in Figure 2C and the

coloration of GZMB could be addressed.

R: We thank the reviewer for pointing these out. We have added the information of treatment details for previous Fig. 2H (revised **Fig. 1H**) in the figure legend. Detailed description for the macrophage-NK cell co-culture can be found in the “**Method**” section (**line 738–778**). Furthermore, we have revised previous Fig. 2C (revised **Appendix Fig. S1**) by highlighting PRF1 and GZMB in the volcano maps.

3. Figure 5: A dose response of HLA-G might enhance data credibility, and the figure appears crowded.

R: We thank the reviewer for these helpful suggestions. We have further tested the effect of different doses of HLA-G on LILRB1 expression on NK cells, and our result confirmed that treatment with HLA-G increased expression of LILRB1 on NK cells in a dose-dependent manner (revised **Fig. 4D**). Moreover, we have rearranged this figure (revised **Fig. 4**) to make it clearer and more orderly.

4. Figure 6: The western blot in panel C could benefit from quantification, and the study with chemical inhibitors lacks robustness.

R: The quantitative results of previous Fig. 6C (revised **Fig. 5C**) is shown in previous Fig. 6D (revised **Fig. 5D**). Probably, the reviewer missed this information due to the overcrowded panels in this figure. We have now rearranged the graphs and displayed them in a more rational way (revised **Fig. 5C,D**).

Moreover, to avoid the potential off-target effect of inhibitors, we have further employed siRNA to knock down *ERK1/2* or *SHP1/2* genes in NK cells to confirm the role of SHP1/2-MAPK signaling in LILRB1-HLA-G interaction-mediated suppression of NK cell anti-Mtb function (revised **Appendix Fig. S6A**). We found that ERK1/2 knockdown eliminated the effects of LILRB1 blockade on ATB patient-derived NK cells including the increase of CD107a expression, decrease of apoptosis, and enhancement of anti-Mtb capacity, while *SHP1/2* knockdown by itself or by combined treatment with LILRB1 blockade markedly increased CD107a expression, decreased apoptosis, and enhanced anti-Mtb capacity of NK cells from ATB patients (revised **Appendix Fig. S6B–D**). These new data, together with the results from chemical

inhibitor-based investigations (revised **Fig. 5** and **Fig. EV3**), suggest that the LILRB1-HLA-G interaction suppresses MEK1/2-ERK1/2-ETS1 MAPK signaling via SHP1/2, thus leading to reduced cytotoxic activity and increased apoptosis of NK cells.

5. Figure 7: Providing multiple cells for each condition could improve data representation.

R: As suggested by the reviewer, we have provided representative images in which multiple cells are shown for each condition (revised **Fig. 6A–C**).

6. Figure 9: The graphical abstract is commendable.

R: We thank the reviewer for this comment on our schematic model in previous Fig. 9K (which has been moved to revised **Appendix Fig. S9**). Based on this schematic model, we have also prepared a “Synopsis” (i.e., “graphical abstract”, which is also requested by the editorial office) in which a visual abstract along with a brief summary of our findings were included.

7. In summary, while the manuscript is well-written, clearer figure presentation and more convincing data interpretations are warranted.

R: We thank the reviewer for the above constructive suggestions. We have revised our manuscript accordingly, as in response to the above points.

Once again, we greatly appreciate the reviewers for having helped us improve this manuscript tremendously.

1st Jul 2024

Dear Prof. Liu,

Thank you for the submission of your revised manuscript to EMBO Molecular Medicine. I am pleased to inform you that we will be able to accept your manuscript pending the following final amendments:

1) Authors: Please provide an institutional e-mail address for the co-corresponding author Yu Pang.

2) Source data:

- During our standard source data analysis we note that values in several source data files are duplicated (see attached excel files). We would like to clarify these issues before we proceed with publication of your manuscript. We kindly invite you to check attached source data excel file with identified duplicated values that are color labeled and clarify the cause of these duplications.
- Please upload source data as one (zipped) file per figure for the main figures and group all source data for EV and Appendix figures in one zipped file.

3) In the main manuscript file, please do the following:

- Please address all comments suggested by our data editors listed below:

o Figure legends:

1. Please note that the legends for figures 6d-g is not provided in the sequential manner (legend for figure 6g is provided before legend of figure 6d-f). This needs to be rectified.

2. Please note that the exact p values are not provided in the legends of figures 1a, d-h; 2b; 5a; 3b-h, j-k; 4a-p; 5d-g; 6a-h; 7b-f; 8b-g; EV 1c; EV 2b-c, h-j, m-n; EV 3b, d-g, i-k; EV 4b-c; EV 5a-f.

3. Please indicate the statistical test used for data analysis in the legend of figure 5a.

4. Please note that in figures 1a, d-h; 5d-g; EV 2b-c, h-j, m-n; there is a mismatch between the annotated p values in the figure legend and the annotated p values in the figure file that should be corrected.

5. Please note that the box plots need to be defined in terms of minima, maxima, centre, bounds of box and whiskers, and percentile in the legends of figures 2b; 4c; EV 2e; EV 4b.

- Please correct callouts for the Table S1 to Appendix Table S1.

- Author contributions: Please remove it from the manuscript and specify author contributions in our submission system. CRediT has replaced the traditional author contributions section because it offers a systematic machine-readable author contributions format that allows for more effective research assessment. You are encouraged to use the free text boxes beneath each contributing author's name to add specific details on the author's contribution. More information is available in our guide to authors:

<https://www.embopress.org/page/journal/17574684/authorguide#authorshipguidelines>

- All Materials and Methods need to be described in the main text. We would encourage you to use 'Structured Methods', our new Methods format. According to this format, the Methods section should include a Reagents and Tools Table (listing key reagents, experimental models, software and relevant equipment and including their sources and relevant identifiers) followed by a Methods and Protocols section in which we encourage the authors to describe their methods using a step-by-step protocol format with bullet points, to facilitate the adoption of the methodologies across labs. More information on how to adhere to this format as well as downloadable templates (.docx) for the Reagents and Tools Table can be found in our author guidelines:

<https://www.embopress.org/page/journal/17574684/authorguide#structuredmethods>

An example of a paper with Structured Methods can be found here:

<https://www.embopress.org/doi/full/10.1038/s44320-024-00037-6#sec-4>

- Indicate in legends number and nature of replicates and exact p= values, not a range, along with the statistical test used. To keep the figures "clear" some authors found providing an Appendix table Sx with all exact p-values preferable. You are welcome to do this if you want to.

- In Methods, provide the statement that the informed consent was obtained from all human subjects and that the experiments conformed to the principles set out in the WMA Declaration of Helsinki and the Department of Health and Human Services Belmont Report.

-

- Data availability: Please be aware that all deposited datasets should be freely accessible upon publication.

4) Appendix: Please remove dataset and movie legends.

5) Movies: Please provided movie legends in a readme.txt file and zip them with their corresponding movie file.

6) The Paper Explained: Please add it to the main manuscript text.

7) Synopsis:

- Synopsis image: Please resize the image 550 px-wide x (250-400)-px high.

8) As part of the EMBO Publications transparent editorial process initiative (see our Editorial at

<http://embomolmed.embopress.org/content/2/9/329>), EMBO Molecular Medicine will publish online a Review Process File (RPF) to accompany accepted manuscripts. This file will be published in conjunction with your paper and will include the anonymous referee reports, your point-by-point response and all pertinent correspondence relating to the manuscript. Let us know whether you agree with the publication of the RPF and as here, if you want to remove or not any figures from it prior to publication.

9) Please provide a point-by-point letter INCLUDING my comments as well as the reviewer's reports and your detailed responses (as Word file).

I look forward to reading a new revised version of your manuscript as soon as possible.

Yours sincerely,

Zeljko Durdevic

*** Instructions to submit your revised manuscript ***

1) a .docx formatted version of the manuscript text (including Figure legends and tables)

2) Separate figure files*

3) supplemental information as Expanded View and/or Appendix. Please carefully check the authors guidelines for formatting Expanded view and Appendix figures and tables at <https://www.embopress.org/page/journal/17574684/authorguide#expandedview>

4) a letter INCLUDING the reviewer's reports and your detailed responses to their comments (as Word file).

5) The paper explained: EMBO Molecular Medicine articles are accompanied by a summary of the articles to emphasize the major findings in the paper and their medical implications for the non-specialist reader. Please provide a draft summary of your article highlighting

This may be edited to ensure that readers understand the significance and context of the research.

Please refer to any of our published articles for an example.

6) For more information: There is space at the end of each article to list relevant web links for further consultation by our readers. Could you identify some relevant ones and provide such information as well? Some examples are patient associations, relevant databases, OMIM/proteins/genes links, author's websites, etc...

7) Author contributions: the contribution of every author must be detailed in a separate section.

8) EMBO Molecular Medicine now requires a complete author checklist (<https://www.embopress.org/page/journal/17574684/authorguide>) to be submitted with all revised manuscripts. Please use the

checklist as guideline for the sort of information we need WITHIN the manuscript. The checklist should only be filled with page numbers where the information can be found. This is particularly important for animal reporting, antibody dilutions (missing) and exact values and n that should be indicated instead of a range.

9) Every published paper now includes a 'Synopsis' to further enhance discoverability. Synopses are displayed on the journal webpage and are freely accessible to all readers. They include a short stand first (maximum of 300 characters, including space) as well as 2-5 one sentence bullet points that summarise the paper. Please write the bullet points to summarise the key NEW findings. They should be designed to be complementary to the abstract - i.e. not repeat the same text. We encourage inclusion of key acronyms and quantitative information (maximum of 30 words / bullet point). Please use the passive voice. Please attach these in a separate file or send them by email, we will incorporate them accordingly.

You are also welcome to suggest a striking image or visual abstract to illustrate your article. If you do please provide a jpeg file 550 px-wide x 300-600px high.

10) A Conflict of Interest statement should be provided in the main text

11) Please note that we now mandate that all corresponding authors list an ORCID digital identifier. This takes <90 seconds to complete. We encourage all authors to supply an ORCID identifier, which will be linked to their name for unambiguous name identification.

Currently, our records indicate that the ORCID for your account is 0000-0002-8035-7792.

Link Not Available

12) Include a Reagents and Tools Table as part of the Methods section, which can be downloaded from our author guidelines (<https://www.embopress.org/page/journal/17574684/authorguide#structuredmethods>)

Photos 400-800 DPI

*Additional important information regarding figures and illustrations can be found at <https://bit.ly/EMBOPressFigurePreparationGuideline>. See also figure legend preparation guidelines: <https://www.embopress.org/page/journal/17574684/authorguide#figureformat>

***** Reviewer's comments *****

Referee #1 (Comments on Novelty/Model System for Author):

All okay now - much more accessible

Referee #1 (Remarks for Author):

The authors have responded well to my concerns and have improved the clarity of text and figure presentation.

Referee #3 (Comments on Novelty/Model System for Author):

The authors answered my first set of remarks. I endorse the publication. I still believe that the figures are too dense. I acknowledge the amount of experimental work carried out and the thorough analysis. I look forward to a confirmation study by other labs.

Point-by-point responses to editor's and reviewers' comments

RE: Manuscript (# EMM-2024-19348-V2) “**LILRB1-HLA-G axis defines a checkpoint driving natural killer cell exhaustion in tuberculosis**” by Wang *et al.*

Editor's comments:

2) Source data:

- During our standard source data analysis we note that values in several source data files are duplicated (see attached excel files). We would like to clarify these issues before we proceed with publication of your manuscript. We kindly invite you to check attached source data excel file with identified duplicated values that are color labeled and clarify the cause of these duplications.

R: We thank the editor for careful examination of our datasets. We explain the duplications as follows:

In Figs. 4M, 7C, 7D, and 8D, the concentrations of GZMB and PRF1

were determined using ELISA kits (as described in the “**Materials and Methods**” section). The absolute concentration value of each sample was calculated from a standard curve based on absorbance at 450 nm (OD_{450}) measured with a microplate reader. Therefore, it is reasonable that some samples showed the same OD_{450} as detected by the microplate reader, and accordingly their concentration values calculated from the standard curve were the same as well.

In Fig. 7E, to measure the bacterial CFUs, lung homogenates were added to 7H10 agar plates by serially diluting the original samples as it was too concentrated to count (as described in the “**Materials and Methods**” section). The numbers of visible colonies present on an agar plate can be multiplied by the dilution factor to provide the CFU values. Therefore, when the counted numbers of colonies on the plate are identical between two samples, the calculated CFU (as well as \log_{10} CFU) values will be the same between these two samples.

Reviewer's comments:

Referee #1 (Comments on Novelty/Model System for Author):

All okay now - much more accessible

R: We are delighted to know that the reviewer is satisfied with our revisions.

Referee #2 (Remarks for Author):

The authors have responded well to my concerns and have improved the clarity of text and figure presentation.

R: We are glad to know that the reviewer's concerns have been addressed well.

Referee #3 (Comments on Novelty/Model System for Author):

The authors answered my first set of remarks. I endorse the publication. I still believe that the figures are too dense. I acknowledge the amount of experimental work carried out and the thorough analysis. I look forward to a confirmation study by other labs.

R: We thank the reviewer for endorsing the publication of our manuscript.

Once again, we greatly appreciate the editor and reviewers for having helped us improve this manuscript tremendously.

5th Jul 2024

Dear Prof. Liu,

We are pleased to inform you that your manuscript is accepted for publication and is now being sent to our publisher to be included in the next available issue of EMBO Molecular Medicine.
